# A Random Matrix Analysis of In-context Memorization for Nonlinear Attention

## Abstract

Attention mechanisms have revolutionized machine learning (ML) by enabling efficient modeling of global dependencies across inputs. Their inherently parallelizable structures allow for efficient scaling with the exponentially increasing size of both pretrained data and model parameters. Yet, despite their central role as the computational backbone of modern large language models (LLMs), the theoretical understanding of Attentions, especially in the nonlinear setting, remains limited.

In this paper, we provide a precise characterization of the *in-context memorization error* for *nonlinear Attention*, in the high-dimensional regime where the number of input tokens $n$ and their embedding dimension $p$ are both large and comparable. Leveraging recent advances in the theory of large kernel random matrices, we show that nonlinear Attention typically incurs higher memorization error than linear regression on random inputs. However, this gap vanishes, and can even be reversed, when the input exhibits statistical structure, particularly when the Attention weights align with the input signal direction. Our theoretical insights are supported by numerical experiments.

## 1 Introduction

Since its introduction, the Transformer architecture has become a cornerstone of modern machine learning (ML) and artificial intelligence (AI) (Vaswani et al., 2017), powering large language models (LLMs) such as BERT (Devlin et al., 2019), LLaMA (Touvron et al., 2023), and the GPT series (OpenAI et al., 2024). Originally developed for sequence modeling tasks such as machine translation and language modeling, Transformers have demonstrated remarkable versatility and now achieve state-of-the-art performance across a wide range of applications, including those that are not inherently sequential (Dosovitskiy et al., 2020). At the heart of this empirical success lies the Attention mechanism, which enables flexible integration of information across positions and scales efficiently with both data and model size. Despite its success, our theoretical understanding of Attention, especially in the nonlinear setting, remains limited, particularly in terms of how it learns statistical patterns from high-dimensional input tokens.

Recent years have seen an increasing use of high-dimensional statistics (Vershynin, 2018), statistical physics (Carleo et al., 2019), and random matrix theory (RMT) (Couillet & Liao, 2022) to derive insights in the design and optimization of large-scale ML models. In contrast to worst-case generalization bounds that can sometimes be loose, high-dimensional analysis offers *precise* characterizations and is able to explain phenomena such as the neural tangent kernel (Jacot et al., 2018), double descent in generalization (Mei & Montanari, 2021; Liao et al., 2021; Nakkiran et al., 2020; Hastie et al., 2022), and benign overfitting (Bartlett et al., 2020; 2021), which now inform core ML design principles. More recently, theoretical studies have investigated the connection between self-attention and max-margin–type model (Tarzanagh et al., 2023), as well as convergence properties of Transformer-based models under gradient-based methods (Deora et al., 2023; Vasudeva et al., 2024). In (Cui et al., 2024), the so-called dot-product Attention model has been considered for uncorrelated inputs and low-rank Attention weights, using a Generalized Approximate Message Passing (GAMP) approach. A brief review of related work is provided in Section 1.2, with a more detailed discussion in Appendix A.

Yet, a precise characterization of *nonlinear Attention*, particularly on *structural inputs*, remains largely elusive. The main technical challenges stem from the nonlinearity of the Attention operator and the complex interactions between input tokens and Attention weights via queries, keys, and

values. Prior theoretical efforts often rely on restrictive assumptions: focusing on in-context learning by reducing Attention to gradient descent on (generalized) linear model, which only holds under particular weight configuration (Bai et al., 2023; Lu et al., 2025); assuming simplified Attention matrices (e.g., all-ones (Noci et al., 2022) or random Markov matrices (Naderi et al., 2024)), or drawing connections to Bayesian learning (Tiberi et al., 2024), sequence multi-index models (Troiani et al., 2025), or generalized Potts model (Rende et al., 2024) in statistical mechanics.

In this paper, we take a different perspective by viewing the pretrained nonlinear Attention as an non-symmetric kernel matrix, and present a precise analysis of the *in-context memorization* for nonlinear Attention on structural random inputs (Definition 2). Here, "in-context" refers to the inference-time behavior of a frozen pretrained Attention block, and we study how a fixed Attention memorizes and retrieves information from its current prompt, without any parameter updates. In the high-dimensional regime where the input length $n$ and embedding dimension $p$ are both large and comparable. Building upon recent advances in the eigenspectral analysis of nonlinear random kernel matrices, we derive precise expressions for the in-context memorization error (see Definition 3) of *nonlinear* Attention with pretrained weights admitting a full-plus-low-rank decomposition (Assumption 1).

Our result shows that the memorization error of nonlinear Attention is determined by a system of nonlinear equations involving the dimension ratio $p/n$, the alignment between input signal and pretrained Attention weights, and the nonlinearity (via its Hermite coefficients). By focusing on this canonical setting, our analysis takes a step forward to unveil the theoretical origin of many visually striking features emerging in modern large-scale ML.

## 1.1 Our Contribution

The main novelty of this paper lies in viewing nonlinear Attention as a non-symmetric kernel matrix. This perspective enables a Hermite polynomial expansion to "linearize" the Attention matrix, which in turn allows for a through analysis of its memorization behavior using random matrix techniques. As a result, we develop a systematic framework for studying a new class of random matrix models arising from nonlinear Attention. The main contribution of this paper are summarized as follows.

1. In Theorem 1, we derive a *precise* characterization of the in-context memorization error (Definition 3) for *nonlinear* Attention, under a high-dimensional signal-plus-noise model (Definition 2) for the input tokens. We show that the Attention memorization error is governed by a system of nonlinear equations involving the dimension ratio $p/n$, the interaction between input signal and the Attention weights, and the nonlinearity via its Hermite coefficients.

2. In Section 4, we compare the memorization error of nonlinear Attention to that of linear regression (see Proposition 2). While nonlinear Attention generally incurs higher error than linear regression for random inputs, this disadvantage disappears—and can even be reversed—for structured inputs, particularly when the Attentions weights are well-aligned to the input signal. We further show that Attention lacking a linear component (i.e., with its first-order Hermite coefficient being zero) are *unable* to effectively memorize random and/or structural inputs.

3. From a technical perspective, we establish in Proposition 1 a novel Deterministic Equivalent (see Definition 4 for a formal definition) for the resolvent of a *generalized* sample covariance matrix (SCM) of the form $\mathbf{C}\mathbf{X}\mathbf{X}^\top\mathbf{C}^\top$. This extends classical SCM that has been extensively studied in the literature, by considering a population covariance $\mathbf{C} = \mathbf{C}(\mathbf{X})$ that depends on the input $\mathbf{X}$, and may be of independent interest beyond this paper.

## 1.2 Related Work

Here we briefly review related work. A more detailed discussion is provided in Appendix A.

**Theoretical understanding of Transformer and Attention.** Theoretical studies of Transformers have sought to characterize their expressive power and in-context learning (ICL) capabilities. For example, it has been established that Transformers are universal sequence-to-sequence function approximators (Yun et al., 2019). A growing body of work has focused on understanding the ICL behavior of Transformers and Attention, that is, their ability to adapt to new downstream tasks from a few example (Dong et al., 2024; Xie et al., 2021; Garg et al., 2022b; Li et al., 2023; Bai et al., 2023; Oswald et al., 2023; Wu et al., 2024; Zhang et al., 2024; Chen et al., 2024b; Li et al., 2025). Recent

works further provide training dynamics and generalization analyses of ICL for Attention-based Transformers (Li et al., 2024; Huang et al., 2023; He et al., 2025; Chen et al., 2024a). However, these analyses often rely on restrictive assumptions or idealized weight configurations. In contrast, our work provides a random matrix analysis of nonlinear Attention that explicitly captures the *structured interaction* between structured input signals and Attention weights, offering a more flexible and data-dependent understanding of in-context memorization.

**Memorization of neural networks.** Classical results have characterized the memorization capacity of shallow neural networks under various settings (Baum, 1988; Bubeck et al., 2020), with recent extensions to deep nets (Park et al., 2021; Vardi et al., 2021) as well as single-layer Attention (Mahdavi et al., 2023; Chen & Zou, 2024). These studies often focus on worst-case bounds, e.g., on the number of distinct samples that can be memorized by a network. In contrast, here we focus on the *statistical* (so average-case) in-context memorization of nonlinear Attention, by considering a signal-plus-noise model for the inputs. In particular, our analysis quantifies how the memorization performance depends on the alignment between the Attention weights and the input signal.

**Random matrix analyses of ML methods.** Random matrix theory (RMT) has emerged as a powerful and flexible tool to understand the dynamics and generalization properties of large-scale ML models. It has been successfully applied to shallow (Pennington & Worah, 2017; Liao & Couillet, 2018b;a; Louart et al., 2018) and deep neural networks (Benigni & Péché, 2019; Fan & Wang, 2020; Pastur, 2020), and more recently to *linear* Attention (Lu et al., 2025). These analyses encompass both homogeneous (e.g., standard normal) (Pennington & Worah, 2017; Mei & Montanari, 2019) and structured (e.g., mixture-type) input data (Liao & Couillet, 2018b; Ali et al., 2022; MAI & Liao, 2025). To the best of our knowledge, the present work provides the first precise characterization of the *statistical memorization error* of nonlinear Attention on structured input, extending RMT analysis to a broader and more realistic class of Attention-based models.

*Notations.* Scalars are denoted by lowercase letters, vectors by bold lowercase, and matrices by bold uppercase. For a matrix $\mathbf{X} \in \mathbb{R}^{p \times n}$, we write $\mathbf{X}^\top$ for its transpose, $\mathbf{x}_i \in \mathbb{R}^p$ for $i$th column, and $\|\mathbf{X}\|$ for its spectral norm. We use $\mathbf{I}_p$ for the identity matrix of size $p$. For a vector $\mathbf{x} \in \mathbb{R}^p$, its Euclidean norm is given by $\|\mathbf{x}\| = \sqrt{\mathbf{x}^\top \mathbf{x}}$. For a random variable $x$, we denote its expectation by $\mathbb{E}[x]$.

## 2 PROBLEM SETTING AND PRELIMINARIES

We consider the following form of entry-wise *nonlinear* Attention.

**Definition 1 (Nonlinear Attention).** *Let* $\mathbf{X} = [\mathbf{x}_1, \dots, \mathbf{x}_n] \in \mathbb{R}^{p \times n}$ *be the embedding of an input sequence of tokens* $\mathbf{x}_1, \dots, \mathbf{x}_n \in \mathbb{R}^p$ *of length* $n$. *A (single-head) nonlinear Attention output* $\mathbf{A}_\mathbf{X} \in \mathbb{R}^{p \times n}$ *with key, query, and value matrices* $\mathbf{W}_K \in \mathbb{R}^{d \times p}, \mathbf{W}_Q \in \mathbb{R}^{d \times p}, \mathbf{W}_V \in \mathbb{R}^{p \times p}$ *and entry-wise nonlinearity* $f : \mathbb{R} \to \mathbb{R}$, *is defined as:*

$$\mathbf{A}_\mathbf{X} = \mathbf{W}_V \mathbf{X} f(\mathbf{X}^\top \mathbf{W}_K^\top \mathbf{W}_Q \mathbf{X} / \sqrt{p}) / \sqrt{p} \equiv \mathbf{W}_V \mathbf{X} \mathbf{K}_\mathbf{X}. \tag{1}$$

The Attention in Definition 1 is practically compelling due to its computational advantage over classical Softmax Attention (Wortsman et al., 2023; Ramapuram et al., 2024). Remarkably, under Assumption 1 and for input tokens drawn from the signal-plus-noise model in Definition 2, taking $f$ to be truncated exponential function leads to approximately the same output $\mathbf{A}_\mathbf{X}$ as that using Softmax nonlinearity; see Remark 5 in Appendix B for a detailed discussion. Intuitively, the matrix $\mathbf{K}_\mathbf{X} \equiv f(\mathbf{X}^\top \mathbf{W}_K^\top \mathbf{W}_Q \mathbf{X} / \sqrt{p}) / \sqrt{p} \in \mathbb{R}^{n \times n}$ defines an asymmetric kernel parameterized by $\mathbf{W}_Q, \mathbf{W}_K$, and captures the pairwise similarly of input tokens. The output $\mathbf{A}_\mathbf{X}$ is then obtained by "mixing" the values $\mathbf{W}_V \mathbf{X}$ according to the obtained similarities in $\mathbf{K}_\mathbf{X}$.

We consider that the product of key and query matrices $\mathbf{W}_K^\top \mathbf{W}_Q$ in Definition 1 writes as the sum of a full rank identity matrix and an asymmetric low-rank (in fact rank-one) matrix as follow.

**Assumption 1 (Full-plus-low-rank decomposition of pretrained Attention weights).** *The key and query matrices* $\mathbf{W}_K, \mathbf{W}_Q \in \mathbb{R}^{d \times p}$ *in the pretrained Attention model in Definition 1 satisfy, for some given* $\mathbf{w}_Q, \mathbf{w}_K \in \mathbb{R}^p$,

$$\mathbf{W}_K^\top \mathbf{W}_Q = \mathbf{I}_p + \mathbf{w}_K \mathbf{w}_Q^\top \in \mathbb{R}^{p \times p}. \tag{2}$$

The full-plus-low-rank decomposition for $\mathbf{W}_K^\top \mathbf{W}_Q$ in Assumption 1 is largely inspired by the empirical success of Low-Rank Adaption (LoRA) in fine-tuning Transformer-based LLMs (Hu et al., 2021). Note that Assumption 1 implies that $d \geq p$, though this condition is not essential and can be relaxed by considering block decomposition of $\mathbf{W}_K^\top \mathbf{W}_Q$ with one full-rank sub-block. Also, while here we focus on the rank-one setting in Assumption 1 for clarity, our analysis extends to arbitrary but fixed (compared to $n, p, d$) rank structure; see Remark 6 in Appendix B for further discussion on this point. We would like to emphasize here that by considering, in Assumption 1, the pretrained Attention weights to be *deterministic*, allows us to assess scenarios where the pretrained weights are aligned with, orthogonal to the in-context signal direction $\boldsymbol{\mu}$ (in Definition 2 below), or anywhere in between. This is, in particular, formally different from most existing analyses, which typically incorporate pre-training *explicitly* by considering a particular learning algorithm (e.g., gradient descent).

For the sake of our theoretical analysis, we assume the following for the nonlinearity $f$ in Definition 1.

**Assumption 2** (Nonlinear function $f$). *The function $f : \mathbb{R} \to \mathbb{R}$ in Definition 1 satisfies: (1)* $\lim_{t \to \infty} |f(t)| < \infty$, $|f(x)| \leq C_1 \exp(C_2|x|)$ *for some constants* $C_1, C_2 > 0$; *and (2)* $f$ *is centered with respect to standard Gaussian measure, that is,* $\mathbb{E}[f(\xi)] = 0$ *and* $a_1 \equiv \mathbb{E}[\xi f(\xi)] \neq 0$, $\sqrt{2} a_2 \equiv \mathbb{E}[\xi^2 f(\xi)] = 0$, *and* $\nu \equiv \mathbb{E}[f^2(\xi)]$ *for* $\xi \sim \mathcal{N}(0, 1)$.

The first item of Assumption 2 holds for bounded nonlinearity such as sigmoid, truncated exponential, or ReLU variants. For the second item, note that under Assumption 1 and for tokens $\mathbf{x}_i$ drawn from the signal-plus-noise model in Definition 2 below, it follows from the Central Limit Theorem that the non-diagonal entry of $[\mathbf{X}^\top \mathbf{W}_K^\top \mathbf{W}_Q \mathbf{X}]_{ij} / \sqrt{p} \to \mathcal{N}(0, 1)$ in law as $p \to \infty$ and for $i \neq j$, so that $f$ is applied on a random matrix with asymptotically Gaussian (but strongly correlated) entries, justifying the Gaussian-centric Hermite expansion. We consider in Assumption 2 that the zeroth-order Hermite coefficient $\mathbb{E}[f(\xi)]$ of $f$ is zero: This can be achieved by subtracting the *same* constant from all non-diagonal entries of $\mathbf{K}_X$ in (1) and should *not* alter the Attention memorization behavior.

We consider input tokens independently drawn from the following signal-plus-noise model, in the high-dimensional regime where $n, p, d$ are all large and comparable.

**Definition 2** (**Signal-plus-noise model**). *Each token-target pair* $(\mathbf{x}_i, y_i) \in \mathbb{R}^p \times \{\pm 1\}, i \in \{1, \ldots, n\}$ *is independently drawn from the following binary Gaussian signal-plus-noise model:*

$$\mathbf{x}_i = y_i \boldsymbol{\mu} + \mathbf{z}_i \in \mathbb{R}^p, \quad y_i \in \{\pm 1\}, \quad \mathbf{z}_i \sim \mathcal{N}(\mathbf{0}, \mathbf{I}_p), \tag{3}$$

*where* $\boldsymbol{\mu} \in \mathbb{R}^p$ *is a deterministic signal vector.*

The model in Definition 2 considers that the inputs are as the sum of highly correlated signal ($\boldsymbol{\mu}$) and i.i.d. random noise, and is widely used in the study of statistical learning under structured inputs.

**Assumption 3** (High-dimensional asymptotics). *As* $n \to \infty$, *we have that (1)* $p/n \to c \in (0, \infty)$, $d/n \in (0, \infty)$; *and (2) the mean vector* $\boldsymbol{\mu} \in \mathbb{R}^p$ *in Definition 2, the weight vectors in Assumption 1* $\mathbf{w}_Q, \mathbf{w}_K \in \mathbb{R}^p$ *satisfy* $\limsup_n \max\{\|\boldsymbol{\mu}\|, \|\mathbf{w}_Q\|, \|\mathbf{w}_K\|\} < \infty$.

Under Definition 2 and Assumption 3, the matrix of input tokens writes $\mathbf{X} = \boldsymbol{\mu} \mathbf{y}^\top + \mathbf{Z}$, for $\mathbf{y} = [y_1, \ldots, y_n]^\top \in \mathbb{R}^n$, and random noise matrix $\mathbf{Z} \in \mathbb{R}^{p \times n}$ having i.i.d. standard Gaussian entries. Note that both the rank-one signal $\boldsymbol{\mu} \mathbf{y}^\top$ and the noise matrix $\mathbf{Z}$ have spectral norms of order $O(\sqrt{n})$. so that they are set on even ground in the high-dimensional regime as $n, p \to \infty$ under Assumption 3.

**Remark 1** (Beyond the signal-plus-noise model in Definition 2). Our analysis can be extended beyond the binary Gaussian signal-plus-noise model in Definition 2 in a few aspects. Such extensions include, e.g., sub-exponential mixture model for $\mathbf{z}_i \in \mathbb{R}^p$ having i.i.d. sub-gaussian entries of zero mean and unit variance; and multi-class settings where the number of classes is larger than two but remains finite as $n, p \to \infty$. See Remark 7 in Appendix B for further discussion. Also, it would be of future interest to to consider non i.i.d. input tokens, e.g., those having a temporally correlated structure of the form $\mathbf{X} = \mathbf{Z}\mathbf{C}$ for Toeplitz $\mathbf{C} \in \mathbb{R}^{n \times n}$ to model an auto-regressive process.

In this paper, we aim to quantify the *statistical memorization error* of the nonlinear Attention defined in Definition 1, under the signal-plus-noise input model in Definition 2. To this end, we evaluate the performance of Attention as a feature extractor in downstream tasks via *linear probing*. Let $\mathbf{A}_X \in \mathbb{R}^{p \times n}$ be the nonlinear Attention output defined in (1) of Definition 1 for input matrix $\mathbf{X} = [\mathbf{x}_1, \ldots, \mathbf{x}_n] \in \mathbb{R}^{p \times n}$, and let $\mathbf{y} = [y_1, \ldots, y_n]^\top \in \mathbb{R}^n$ denote the associated labels. We define a

ridge-regularized *linear probing* vector $\mathbf{w} \in \mathbb{R}^p$ that minimizes the following mean squared error (MSE) on the pair $(\mathbf{X}, \mathbf{y})$:

$$L(\mathbf{w}) = \frac{1}{n} \left\| \mathbf{y}^\top - \mathbf{w}^\top \mathbf{A_X} \right\|^2 + \gamma \| \mathbf{W}_V^\top \mathbf{w} \|^2 = \frac{1}{n} \left\| \mathbf{y}^\top - \mathbf{w}_V^\top \mathbf{X} \mathbf{K_X} \right\|^2 + \gamma \| \mathbf{w}_V \|^2 \equiv L(\mathbf{w}_V), \qquad (4)$$

where $\mathbf{w}_V = \mathbf{W}_V^\top \mathbf{w} \in \mathbb{R}^p$, and $\gamma \geq 0$ is the regularization penalty. For any $\gamma > 0$, the unique minimizer of (4) admits the following closed-form expression:

$$\mathbf{w}_V^* = \left( \mathbf{X} \mathbf{K_X} \mathbf{K_X}^\top \mathbf{X}^\top + n\gamma \mathbf{I}_p \right)^{-1} \mathbf{X} \mathbf{K_X} \mathbf{y} = \mathbf{X} \mathbf{K_X} \left( \mathbf{K_X}^\top \mathbf{X}^\top \mathbf{X} \mathbf{K_X} + n\gamma \mathbf{I}_n \right)^{-1} \mathbf{y}. \qquad (5)$$

With the explicit linear probing solution $\mathbf{w}_V^*$ given in (5), we now define the *in-context memorization error* of the nonlinear Attention in Definition 1 as follow.

**Definition 3** (**In-context memorization error of nonlinear Attention**). *For $(\mathbf{X}, \mathbf{y}) \in \mathbb{R}^{p \times n} \times \{\pm 1\}^n$ drawn from the signal-plus-noise model in Definition 2, the* in-context memorization error *of nonlinear Attention in Definition 1 is defined as the mean squared error of the optimal linear probe $\mathbf{w}_V^*$ in (5):*

$$E_{\mathrm{A}} = \frac{1}{n} \left\| \mathbf{y}^\top - (\mathbf{w}_V^*)^\top \mathbf{X} \mathbf{K_X} \right\|^2 = -\frac{\gamma^2}{n} \frac{\partial \mathbf{y}^\top \mathbf{Q}(\gamma) \mathbf{y}}{\partial \gamma}, \qquad (6)$$

*where we denote the* nonlinear resolvent matrix

$$\mathbf{Q}(\gamma) = \left( \mathbf{K_X}^\top \mathbf{X}^\top \mathbf{X} \mathbf{K_X} / n + \gamma \mathbf{I}_n \right)^{-1}. \qquad (7)$$

By Equation (6), assessing the in-context memorization error of nonlinear Attention reduces to the analysis of the quadratic form $\mathbf{y}^\top \mathbf{Q}(\gamma) \mathbf{y}$ of the random nonlinear resolvent $\mathbf{Q}(\gamma)$ defined in (7). When the random input tokens $\mathbf{X}$ are drawn from the signal-plus-noise model in Definition 2, this analysis presents the following technical challenges.

1. The resolvent matrix $\mathbf{Q}$ depends on the input $\mathbf{X}$ in a highly *nonlinear* fashion: both through the entry-wise nonlinearity $f$ (see Definition 1) and through the matrix inverse in (7).

2. The structure of $\mathbf{Q}$ is more complex than classical random matrix models (e.g., Wigner or Wishart matrices) studied in RMT (Bai & Silverstein, 2010) or high-dimensional statistics (Vershynin, 2018). Specially, the matrix $\mathbf{K_X}^\top \mathbf{X}^\top \mathbf{X} \mathbf{K_X} / n$ can be viewed as a nonlinear extension of the standard sample covariance (or Gram) matrix (Marcenko & Pastur, 1967; Baik & Silverstein, 2006), but with the key distinction that the population covariance taking the form of an Attention kernel matrix that is itself *dependent* of $\mathbf{X}$.

These challenges must be addressed to characterize the memorization error of nonlinear Attention. To this end, we introduce the notion of *Deterministic Equivalent*, which provides a tractable surrogate for analyzing the high-dimensional behavior of the random resolvent $\mathbf{Q}(\gamma)$ defined in (7).

**Definition 4** (**Deterministic Equivalent**, (Couillet & Liao, 2022, Definition 4)). *Let $\mathbf{Q} \in \mathbb{R}^{n \times n}$ be a sequence of random matrices. A sequence of deterministic matrices $\bar{\mathbf{Q}}$ (of the same size) is called a* Deterministic Equivalent *for $\mathbf{Q}$, denoted $\mathbf{Q} \leftrightarrow \bar{\mathbf{Q}}$, if for all (sequences of) deterministic matrices $\mathbf{A} \in \mathbb{R}^{n \times n}$ and vectors $\mathbf{a}, \mathbf{b} \in \mathbb{R}^n$ of unit spectral and Euclidean norm, we have,*

$$\mathbf{Q} \leftrightarrow \bar{\mathbf{Q}} \ : \ \frac{1}{n} \operatorname{tr} \left( \mathbf{A}(\mathbf{Q} - \bar{\mathbf{Q}}) \right) \to 0, \quad \mathbf{a}^\top (\mathbf{Q} - \bar{\mathbf{Q}}) \mathbf{b} \to 0, \qquad (8)$$

*in probability as $n \to \infty$.*

We aim to deriving a Deterministic Equivalent for the nonlinear resolvent $\mathbf{Q}(\gamma)$ defined in (7), which in turn enables high-dimensional characterization of the quadratic form $\mathbf{y}^\top \mathbf{Q}(\gamma) \mathbf{y} / n$ and the in-context memorization error $E_{\mathrm{A}}$ in (6) of Definition 3. This is the focus of the next section.

## 3 MAIN TECHNICAL RESULTS

This section presents our main technical contributions. We begin with Lemma 1, which establishes a high-dimensional linearization of the Attention kernel matrix $\mathbf{K_X}$ defined in (1). Next, Proposition 1 provides a Deterministic Equivalent for the noise-only nonlinear Attention resolvent. Together, these results enables a precise characterization of the in-context memorization error $E_{\mathrm{A}}$ defined in Definition 3, which we present in Theorem 1 at the end of this section.

To start with, note that under the full-plus-low-rank decomposition of the Attention weights in Assumption 1, the Attention kernel matrix $\mathbf{K_X}$ in (1) admits a more tractable approximation via a Hermite polynomial expansion in the high-dimensional regime of Assumption 3. This is given in the following result and proven in Appendix C.2.

**Lemma 1** (**High-dimensional linearization of Attention kernel matrix**). *Let Assumptions 1–3 hold. Then, the Attention kernel matrix* $\mathbf{K_X} = f(\mathbf{X}^\top \mathbf{W}_K^\top \mathbf{W}_Q \mathbf{X}/\sqrt{p})/\sqrt{p}$ *defined in* (1) *satisfies*

$$\|\mathbf{K_X} - \tilde{\mathbf{K}}_\mathbf{X}\| = O(n^{-1/2}) \quad \textit{with} \quad \tilde{\mathbf{K}}_\mathbf{X} = \mathbf{K}_N + \mathbf{U}_K \Sigma_\mathbf{K} \mathbf{V}_Q^\top, \quad \Sigma_\mathbf{K} = a_1 \begin{bmatrix} \|\boldsymbol{\mu}\|^2 + \boldsymbol{\mu}^\top \mathbf{w}_K \mathbf{w}_Q^\top \boldsymbol{\mu} & 1 & \boldsymbol{\mu}^\top \mathbf{w}_K \\ 1 & 0 & 0 \\ \boldsymbol{\mu}^\top \mathbf{w}_Q & 0 & 1 \end{bmatrix} \in \mathbb{R}^{3\times3}, \quad (9)$$

*with probability approaching one as* $n, p \to \infty$. *Here,* $a_1$ *is the first Hermite coefficient of* $f$ *(see Assumption 2),* $\mathbf{K}_N \equiv f(\mathbf{Z}^\top \mathbf{Z}/\sqrt{p})/\sqrt{p} - \mathrm{diag}(\cdot)$ *is a* symmetric *noise-only kernel matrix and*

$$\mathbf{U}_K = [\mathbf{y}, \mathbf{Z}^\top \boldsymbol{\mu}, \mathbf{Z}^\top \mathbf{w}_K]/\sqrt{p} \in \mathbb{R}^{n\times3}, \quad \mathbf{V}_Q = [\mathbf{y}, \mathbf{Z}^\top \boldsymbol{\mu}, \mathbf{Z}^\top \mathbf{w}_Q]/\sqrt{p} \in \mathbb{R}^{n\times3}. \quad (10)$$

*Moreover, we have that* $\max\{\|\mathbf{K}_N\|, \|\mathbf{U}_K\|, \|\Sigma_\mathbf{K}\|, \|\mathbf{V}_Q\|\} = O(1)$ *with high probability as* $n, p \to \infty$.

Lemma 1 shows that the nonlinear kernel matrix $\mathbf{K_X}$ can be decomposed as the sum of: (1) a symmetric noisy-only random kernel matrix $\mathbf{K}_N$;[1] and (2) a low-rank, asymmetric informative matrix (rank at most three), whose structure depends on the interaction between the signal $\boldsymbol{\mu}$ and Attention weights $\mathbf{w}_K, \mathbf{w}_Q$, and on the nonlinearity $f$ only via its first Hermite coefficient $a_1 = \mathbb{E}[\xi f(\xi)], \xi \sim \mathcal{N}(0,1)$. Note that under Definition 2, the input matrix $\mathbf{X} = \boldsymbol{\mu}\mathbf{y}^\top + \mathbf{Z}$ also admits a rank-one signal-plus-noise decomposition. As such, the matrix of interest $\mathbf{K_X}^\top \mathbf{X}^\top \mathbf{X} \mathbf{K_X}/n$ can be approximated, per Lemma 1, as the sum of some full-rank and low-rank matrices.

In the following result, we focus on the full-rank (and noise-only) part of the Attention matrix and derive a Deterministic Equivalent for its resolvent, the proof of which is given in Appendix C.3.

**Proposition 1** (**Deterministic Equivalent for noise-only nonlinear Attention**). *Let* $\mathbf{Z} \in \mathbb{R}^{p\times n}$ *be a random matrix having i.i.d. standard Gaussian entries, and define the symmetric noise-only kernel matrix* $\mathbf{K}_N = f(\mathbf{Z}^\top \mathbf{Z}/\sqrt{p})/\sqrt{p} - \mathrm{diag}(\cdot)$ *as in Lemma 1. Then, as* $n, p \to \infty$ *with* $p/n \to c \in (0, \infty)$ *and* $\gamma > 0$, *the following Deterministic Equivalent (see Definition 4) holds*

$$\left(\mathbf{K}_N \mathbf{Z}^\top \mathbf{Z} \mathbf{K}_N/n + \gamma \mathbf{I}_n\right)^{-1} \leftrightarrow m(\gamma)/c \cdot \mathbf{I}_n,$$

*where* $m(\gamma)$ *is the unique Stieltjes transform solution to the fixed-point equation*

$$m(\gamma) = \left(\gamma/c + v/c + a_1^2/c^2 - \mathbf{v}^\top \mathbf{T}(\gamma)\mathbf{v}\right)^{-1},$$

*with* $\mathbf{v} = \begin{bmatrix} \frac{a_1^2}{c^2}(1+c) & \frac{a_1}{c} & \frac{a_1}{c} & 0 & 0 & 1 \end{bmatrix}^\top \in \mathbb{R}^6$ *and* $\mathbf{T}(\gamma) \in \mathbb{R}^{6\times6}$ *is a symmetric matrix whose entries are polynomial involving* $m(\gamma), \delta_1(\gamma), \delta_2(\gamma), \delta_3(\gamma), \delta_4(\gamma)$ *defined in* (28) *of Appendix C.3. Notably, the system of equations depends on the regularization penalty* $\gamma$, *the dimension ratio* $c$, *and the nonlinearity* $f$ *via its Hermite coefficients* $a_1$ *and* $v$ *in Assumption 2.*

Using Lemma 1 and Proposition 1, we obtain the following precise characterization of the in-context memorization error $E_A$ for nonlinear Attention. The proof is given in Appendix C.4.

**Theorem 1** (**High-dimensional characterization of in-context memorization error**). *Let Assumptions 1–3 hold. Then, the in-context memorization error* $E_A$ *defined in* (6) *satisfies* $E_A - \bar{E}_A \to 0$ *in probability as* $n, p \to \infty$ *with* $p/n \to c \in (0, \infty)$, *where*

$$\bar{E}_A = -\gamma^2 c^2 \cdot \mathbf{e}_7^\top (c\mathbf{I}_9 + \Delta(\gamma)\Lambda)^{-1} \Delta'(\gamma) (c\mathbf{I}_9 + \Lambda\Delta(\gamma))^{-1} \mathbf{e}_7. \quad (11)$$

*Here,* $\mathbf{e}_7 \in \mathbb{R}^9$ *is the canonical basis vector with* $[\mathbf{e}_i]_j = \delta_{ij}$, $\Lambda, \Delta(\gamma) \in \mathbb{R}^{9\times9}$ *are symmetric matrices defined in Lemma 8 of Appendix C.4, and* $\Delta'(\gamma)$ *is the derivative of* $\Delta(\gamma)$ *with respect to* $\gamma$.

---

[1]The noise-only kernel matrix $\mathbf{K}_N$ is known in the literature as a *random inner-product kernel matrix* (Cheng, 2013; Fan & Montanari, 2019; Kammoun & Couillet, 2023), with connections to single-hidden-layer (random) neural networks (Pennington & Worah, 2017; Benigni & Péché, 2019).

# 4 MEMORIZATION OF NONLINEAR ATTENTION VERSUS LINEAR REGRESSION

In this section, we discuss the implications of our technical results in Theorem 1, by contrasting the in-context memorization behavior of nonlinear Attention with that of linear regression.

## 4.1 IN-CONTEXT MEMORIZATION OF LINEAR REGRESSION

We begin by considering a classical baseline where the input embedding matrix $\mathbf{X}$ is directly used for linear probing, instead of the nonlinear Attention output $\mathbf{A_X}$ defined in (1) of Definition 1. In this case, the probing vector $\mathbf{w}_{\mathrm{LR}} \in \mathbb{R}^p$ is obtained by minimizing the following ridge-regularized MSE:

$$L_{\mathrm{LR}}(\mathbf{w}) = \frac{1}{n} \left\| \mathbf{y}^\top - \mathbf{w}^\top \mathbf{X} \right\|^2 + \gamma \|\mathbf{w}\|^2. \tag{12}$$

This leads to the linear regression model defined below.

**Definition 5 (Linear regression and its in-context memorization error).** *For* $(\mathbf{X}, \mathbf{y}) \in \mathbb{R}^{p \times n} \times \{\pm 1\}^n$ *drawn from the signal-plus-noise model in Definition 2, the linear regression solution* $\mathbf{w}_{\mathrm{LR}}$ *is given by*

$$\mathbf{w}_{\mathrm{LR}} = \left( \mathbf{X}\mathbf{X}^\top + n\gamma \mathbf{I}_p \right)^{-1} \mathbf{X}\mathbf{y} = \mathbf{X} \left( \mathbf{X}^\top \mathbf{X} + n\gamma \mathbf{I}_n \right)^{-1} \mathbf{y}, \quad \gamma > 0. \tag{13}$$

*Its associated* in-context memorization error *is given by*

$$E_{\mathrm{LR}} = \frac{1}{n} \left\| \mathbf{y}^\top - \mathbf{w}_{\mathrm{LR}}^\top \mathbf{X} \right\|^2 = -\frac{\gamma^2}{n} \frac{\partial \mathbf{y}^\top \left( \mathbf{X}^\top \mathbf{X}/n + \gamma \mathbf{I}_n \right)^{-1} \mathbf{y}}{\partial \gamma}, \tag{14}$$

*which is also the derivative of the quadratic form of the* linear resolvent $(\mathbf{X}^\top \mathbf{X}/n + \gamma \mathbf{I}_n)^{-1}$.

We now characterize the linear regression memorization error $E_{\mathrm{LR}}$ in (14), in the high-dimensional regime of Assumption 3. The proof is standard and included in Appendix C.5 for completeness.

**Proposition 2 (High-dimensional characterization of in-context memorization for linear regression).** *Let Assumption 3 hold. Then, the in-context memorization error* $E_{\mathrm{LR}}$ *defined in (14) of the linear regression model in Definition 5 satisfies* $E_{\mathrm{LR}} - \bar{E}_{\mathrm{LR}} \to 0$ *in probability as* $n, p \to \infty$, *with*

$$\bar{E}_{\mathrm{LR}} = -\frac{c\gamma^2 m'(\gamma) + c - 1 + \|\boldsymbol{\mu}\|^2 \left( \gamma^2 m'(\gamma) + (1 - c - \gamma)(\gamma m(\gamma) - 1) \right)}{(1 + \|\boldsymbol{\mu}\|^2 - \|\boldsymbol{\mu}\|^2 \gamma m_{\mathrm{LR}}(\gamma))^2}, \tag{15}$$

*where* $m_{\mathrm{LR}}(\gamma)$ *is the Stieltjes transform solution to the following Marčenko-Pastur equation (Marcenko & Pastur, 1967):*

$$c\gamma m_{\mathrm{LR}}^2(\gamma) + (1 - c + \gamma) m_{\mathrm{LR}}(\gamma) - 1 = 0, \tag{16}$$

*and* $m'_{\mathrm{LR}}(\gamma) = -\frac{c m_{\mathrm{LR}}^2(\gamma) + m_{\mathrm{LR}}(\gamma)}{2c\gamma m_{\mathrm{LR}}(\gamma) + 1 - c + \gamma}$ *is its derivative with respect to* $\gamma$.

In what follows, we leverage Proposition 2 to assess how the in-context memorization error $E_{\mathrm{LR}}$ of linear regression is influenced by: the regularization strength $\gamma$, the dimension ratio $c = \lim p/n$, and the signal-to-noise ratio (SNR) $\|\boldsymbol{\mu}\|^2$.

**Remark 2 (Effect of regularization strength for linear regression).** Under the settings and notations of Proposition 2, the in-context memorization error $E_{\mathrm{LR}}$ is an *increasing* function of the regularization strength $\gamma$. In the "ridgeless" limit $\gamma \to 0$, the memorization error vanishes $E_{\mathrm{LR}} \to 0$ for $p > n$; whereas in the strongly regularized limit $\gamma \to \infty$ we have $E_{\mathrm{LR}} \to 1$. Interestingly, when $\gamma \to 0$ and $c = \lim p/n \to 0$, the Stieltjes transform $m_{\mathrm{LR}}(\gamma)$ becomes singular, which is connected to the now well-known "double descent" phenomenon in test error curves (Bartlett et al., 2020; Mei & Montanari, 2021; Liao et al., 2020; Hastie et al., 2022).

**Remark 3 (Effect of embedding dimension for linear regression).** The in-context memorization error $E_{\mathrm{LR}}$ of linear regression is a *decreasing* function of the dimension ratio $c = \lim p/n$. For fixed $n$, increasing the embedding dimension $p$ thus improves memorization. In the limit $c \to 0$ and for $\gamma = 0$, the memorization error converges to $E_{\mathrm{LR}} \to 1/(1 + \|\boldsymbol{\mu}\|^2)$. Moreover, in the under-parametrized setting with $p < n$ and $\gamma = 0$, the memorization error $E_{\mathrm{LR}}$ scales approximately with the embedding dimension $p$ as $1 - c = 1 - p/n$, in line with classical statistical learning theory (Bach, 2024).

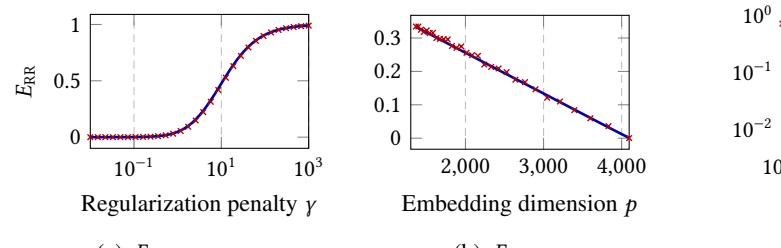

(a) $E_{RR}$ versus $\gamma$  (b) $E_{RR}$ versus $p$  (c) $E_{RR}$ versus SNR

Figure 1: Empirical memorization error $E_{LR}$ (**red**) of linear regression versus its high-dimensional equivalent $\bar{E}_{LR}$ (**blue**) in Proposition 2. **Figure 1a**: As a function of regularization strength $\gamma$, with $p = 2\,048, n = 512$, and $\|\boldsymbol{\mu}\|^2 = 1$. **Figure 1b**: As a function of embedding dimension $p$, with $n = 4\,096, \gamma = 10^{-5}$, and $\|\boldsymbol{\mu}\|^2 = 1$. **Figure 1c**: As a function of signal-to-noise ratio (SNR) $\|\boldsymbol{\mu}\|^2$, with $p = 512, n = 2\,048$, and $\gamma = 10^{-5}$.

**Remark 4** (Effect of SNR for linear regression). The in-context memorization error $E_{LR}$ *decreases* with the SNR $\|\boldsymbol{\mu}\|^2$. In the limit $\|\boldsymbol{\mu}\| \to \infty$, one has $E_{LR} \to 0$. In particular, for $\gamma = 0$ and $p < n$, the error scales as $E_{LR} \propto 1/(1 + \|\boldsymbol{\mu}\|^2)$, a trend clearly illustrated in Figure 1c.

Remarks 2, 3, and 4 are confirmed empirically in Figure 1, where we compare the theoretical $\bar{E}_{LR}$ to the empirical $E_{LR}$ over varying regularization strength, embedding dimension, and SNR.

## 4.2 In-context Memorization of Nonlinear Attention versus Linear Regression

Similar to the discussions of linear regression and the empirical trends shown in Figure 1, we compare in Figure 2 the empirical memorization error $E_A$ of nonlinear Attention with its theoretical counterpart $\bar{E}_A$ in Theorem 1, as well as with linear regression *under the same setting*.

In Figure 2a and Figure 2b, we consider the null model with no statistical signal ($\boldsymbol{\mu} = \mathbf{0}$) and for identity Attention weights ($\mathbf{w}_K = \mathbf{w}_Q = \mathbf{0}$). We observe that the in-context memorization error of nonlinear Attention exhibits the same qualitative trends as linear regression: increasing with the regularization strength $\gamma$ and decreasing with the embedding dimension $p$. Quantitatively, however, nonlinear Attention (with tanh nonlinearity at least in Figure 2) incurs a *higher* memorization error than linear regression, but *only in the absence signal*.

In contrast, in the presence of structured input signals ($\boldsymbol{\mu} \neq \mathbf{0}$) and when the Attention weights $\mathbf{w}_K, \mathbf{w}_Q$ are *aligned* with the signal, we find in Figure 2c that the memorization error of Attention are visually indistinguishable from linear regression as the SNR $\|\boldsymbol{\mu}\|^2$ increase. This illustrates that the disadvantage of nonlinear Attention in memorization vanishes when it is tuned to the input structure.

We provide further numerical results in Appendix D showing that this disadvantage can even be reversed and Attention has a significantly better statistical pattern memorization than linear regression, particularly in the *high SNR and/or limited sample regime*, see, e.g., Figure 5 in Appendix D.

Figure 2 only concerns with tanh Attention. In the following, we show that the (scaling laws of) in-context memorization error of nonlinear Attention strongly depend on the nonlinearity.

## 4.3 Importance of Linear Component for Nonlinear Attention

Figure 3 illustrates the role played by the *linear component* of the Attention nonlinearity $f$, quantified by its first Hermite coefficient $a_1 = \mathbb{E}_{\xi \sim \mathcal{N}(0,1)}[\xi f(\xi)]$, in improving memorization performance.

In Figure 3a, we consider a one-parameter family of nonlinearities parameterized by $r > 0$, $f_r(t) = \max\left(-5, \min(5, r\mathrm{He}_1(t) + \sqrt{1 - r^2}\mathrm{He}_3(t))\right)$, where $\mathrm{He}_1(t) = t$ and $\mathrm{He}_3(t) = (t^3 - 3t)/\sqrt{6}$ is the first and third normalized Hermite polynomial, respectively. Fixing the "total energy" of $f$ to $\nu = \mathbb{E}_{\xi \sim \mathcal{N}(0,1)}[f^2(\xi)] \approx 1$, we observe that memorization error decreases with increasing $a_1$, highlighting the crucial role of the linear component in $f$.

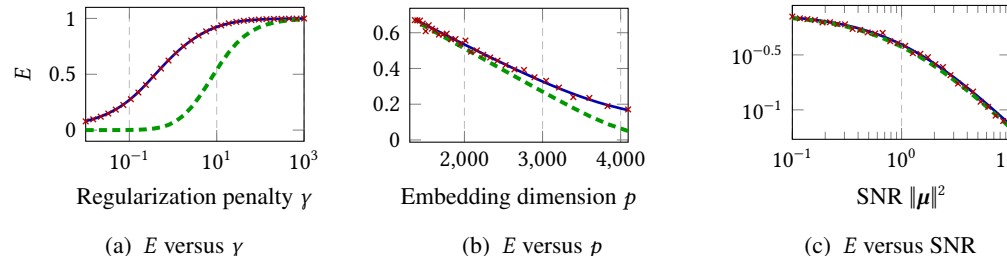

(a) $E$ versus $\gamma$      (b) $E$ versus $p$      (c) $E$ versus SNR

Figure 2: Empirical memorization error $E$ (**red**) of nonlinear Attention versus its high-dimensional equivalent $\bar{E}$ (**blue**) from Theorem 1, and the theoretical memorization error of linear regression (**green**) from Proposition 2, with $f(t) = \tanh(t)$. **Figure 2a**: As a function of regularization strength $\gamma$, under null model with $\boldsymbol{\mu} = \mathbf{w}_K = \mathbf{w}_Q = \mathbf{0}$, $p = 4\,096$, and $n = 1\,024$. **Figure 2b**: As a function of embedding dimension $p$, under null model with $n = 4\,096$, $\gamma = 10^{-2}$. **Figure 2c**: As a function of SNR $\|\boldsymbol{\mu}\|^2$, with $p = 512$, $n = 2\,048$, $\gamma = 10^{-2}$, and $\mathbf{w}_K = \mathbf{w}_Q = \boldsymbol{\mu}$.

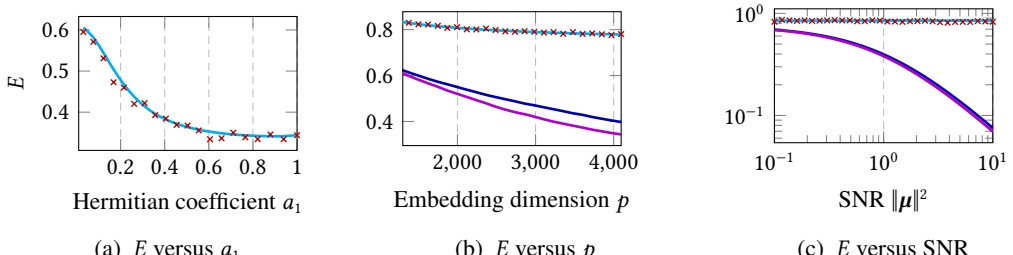

(a) $E$ versus $a_1$      (b) $E$ versus $p$      (c) $E$ versus SNR

Figure 3: Effect of linear component in Attention memorization. **Figure 3a**: Empirical (**red**) and theoretical (**cyan**) memorization error for $f(t) = \max(-5, \min(5, rt + \sqrt{1 - r^2}(t^3 - 3t)/\sqrt{6}))$ as a function of the Hermite coefficient $a_1 \approx r$ for $p = n = 4\,096$, $\gamma = 1$, and $\|\boldsymbol{\mu}\|^2 = 1$. **Figure 3b**: Empirical (**red**) and theoretical (**cyan**) for $f(t) = \cos(t)$, versus the theoretical error of $f(t) = \tanh(t)$ (**blue**) and the theoretical error of $f(t) = \max(-5, \min(5, t))$ (**purple**), as a function of the embedding dimension $p$, for in-context sample size $n = 4\,096$, $\gamma = 1$, and $\|\boldsymbol{\mu}\|^2 = 1$. **Figure 3c**: Empirical (**red**) and theoretical (**cyan**) for $f(t) = \cos(t)$, versus the theoretical error of $f(t) = \tanh(t)$ (**blue**) and the theoretical error of $f(t) = \max(-5, \min(5, t))$ (**purple**), as a function of the SNR $\|\boldsymbol{\mu}\|^2$, for $p = 512$, $n = 2\,048$, $\gamma = 10^{-2}$, and $\mathbf{w}_K = \mathbf{w}_Q = \boldsymbol{\mu}$.

To further support this, we compare, in Figure 3b and Figure 3c respectively, the trends of in-context memorization errors as a function of the embedding dimension $p$ and SNR, for three nonlinearities $f(t) = \tanh(t)$ (with $a_1 = 0.6057$), bounded linear $f(t) = \max(-5, \min(5, t))$ (with $a_1 \approx 1$), and $f(t) = \cos(t)$ (with $a_1 \approx 0$). As shown in Figure 3b, when $p$ increases, only Attentions having a linear component ($a_1 \neq 0$) exhibit a meaningful gain in memorization performance. Similarly, in Figure 3c, cosine-based Attention shows almost no improvement as SNR increases, whereas Attentions having a linear component consistently improve.

These findings suggest that retaining a sufficient linear component in the Attention nonlinearity is not merely beneficial but *essential* for efficient information integration and memorization in Transformer-based architectures.

Further experiments are presented in Appendix D to illustrate in greater detail the impact of the dimension ratio $p/n$, the SNR, the Attention nonlinearity, and the alignment between Attention weights and input data signal $\boldsymbol{\mu}$ on in-context memorization performance. Additionally, numerical results based on pretrained GPT-2 weights are included, showing trends that closely align with the theoretical predictions derived in Theorem 1.

## 5    CONCLUSION AND PERSPECTIVES

In this paper, we provide a precise high-dimensional characterization of the in-context memorization error for nonlinear Attention on structured inputs. We show that, although nonlinear Attention typically incurs slighter higher memorization error than linear regression for random inputs, this disadvantage vanishes—and can even be reversed—when the input possesses structure, particularly when the Attentions weights are aligned with the underlying input signal.

A natural extension of this work is to incorporate more realistic architectural components used in practical Transformers, such as skip connections or multi-head Attention. Another interesting direction is to go beyond the i.i.d. signal-plus-noise model in Definition 2. In real-world scenarios such as natural language processing or time series analysis, the input (tokenized) sequences typically exhibit strong temporal correlations. For instance, the case of linear temporal correlation has been recently studied in (Moniri & Hassani, 2024), though limited to linear regression model. It would be of interest to extend our nonlinear random matrix analysis to such structured input settings.

## ETHICS STATEMENT

This submission focuses on the theoretical analysis of nonlinear Attention using random matrix theory. We do not feel that this submission raises any ethical concerns regarding, e.g., human subjects or potentially harmful insights.

## REPRODUCIBILITY STATEMENT

The numerical experiments are obtained using synthetic data drawn from the binary Gaussian signal-plus-noise model defined in Definition 2. The experimental settings, hyperparameter choices, and results are fully described in the main text and in the appendix. Consequently, the numerical results reported in this paper can be independently reproduced without relying on any proprietary datasets or external resources.

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

## Supplementary Material of
### A Random Matrix Analysis of In-context Memorization for Nonlinear Attention

The technical appendices of this paper are organized as follows. An extended discussion of related work is given in Appendix A. Some auxiliary results and discussions are placed in Appendix B. The detailed proofs of our technical results are given in Appendix C. Additional numerical results are provided in Appendix D.

## A    FURTHER DISCUSSIONS OF PRIOR EFFORTS

**Transformers and empirical scaling laws.**    A growing body of work has established empirical scaling laws for Transformer models with respect to data size, model size, and computational budget. Early studies demonstrated power-law curves between generalization performance and model size for Transformer-based LLMs (Kaplan et al., 2020; Henighan et al., 2020), with subsequent extensions to transfer and multitask learning (Hernandez et al., 2021; Wei et al., 2022). Notably, it has been shown in (Hoffmann et al., 2022) that smaller models trained on more data can outperform larger "undertrained" models under fixed compute budget. Other studies have explored the effects of overparameterization, initialization, and depth-width trade-offs in the scaling laws of Transformer-based models (Bahri et al., 2024; Zhai et al., 2022; Xiao et al., 2021). Emergent phenomena and scaling transitions such as double descent (Nakkiran et al., 2020), in-context induction (Olsson et al., 2022b), and phase shifts in predictability (Ganguli et al., 2022) have also been empirically observed. Investigations on Vision Transformers and instruction-tuned models (Dosovitskiy et al., 2020; Chowdhery et al., 2023) further support the universality of scaling behaviors across different modalities.

Our work complements these empirical findings by providing a *precise* theoretical characterization on the scaling law of in-context memorization error of nonlinear Attention as a function of the sample-to-dimension ratio ($n/p$) and the SNR of the input data.

**Efficient Transformer variants and low-rank adaptation.**    The quadratic complexity of vanilla Attention with respect to sequence length has motivated a wide range of approximation-based methods to improve computational efficiency. Performer has replaced the Softmax nonlinearity with kernel-based random projections to achieve near-linear complexity (Krzysztof et al., 2021); Linformer has projected keys and values into a low-dimensional subspace (Wang et al., 2020); Nyströmformer approximates the Attention matrix using the Nyström method (Xiong et al., 2021); and Reformer has combined locality-sensitive hashing with reversible layers for memory savings (Kitaev et al., 2020). In parallel, a series of works have proposed low-rank adaptation techniques for efficient fine-tuning of Transformer-based LLMs. LoRA has introduced trainable low-rank updates to frozen weights (Hu et al., 2022); QLoRA has extended this idea to quantized models with minimal performance degradation (Dettmers et al., 2023); LoRA-FA has improved memory efficiency via factorized updates (Zhang et al., 2023); and UniPELT has unified multiple parameter-efficient tuning strategies into a single framework (Mao et al., 2021).

Motivated by these low-rank structures in computing and/or fine-tuning Transformer-based models, we assume in Assumption 1 a full-plus-low-rank decomposition of the Attention weights, and characterizes how such structure affects the memorization capacity of nonlinear Attention.

**Theoretical understanding of DNN, LLMs, and in-context learning.**    Recent theoretical advances in the optimization and generalization of over-parameterized deep neural networks (DNNs) have laid the groundwork for understanding the training behavior of modern large language models (LLMs). Despite the fact that LLMs typically operate in a regime where the number of model parameters far exceeds the number of training samples, first-order methods such as stochastic gradient descent have been shown to converge reliably and generalize effectively under specific conditions (Li & Liang, 2018; Allen-Zhu et al., 2019) for DNNs. Notably, the "edge of stability" phenomenon has emerged as a key concept, capturing the peculiar yet effective optimization dynamics commonly observed during the training of DNNs and LLMs (Cohen et al., 2021; Arora et al., 2022; Wang et al., 2022). Building on these insights, a growing body of work has investigated the mechanisms underlying in-context learning (ICL). Transformers have been shown to approximate gradient descent steps via Attention blocks (Von Oswald et al., 2023; Mahankali et al., 2023), and even to implement general-

purpose learning algorithms directly from contextual input (Akyürek et al., 2022; Garg et al., 2022a). Connection has also been drawn on Transformer computation and functional gradient descent (Cheng et al., 2023). Alternative viewpoints interpret ICL as a form of implicit Bayesian inference (Xie et al., 2021; Falck et al., 2024), offering probabilistic frameworks to explain generalization from prompts. At the mechanistic level, recent work has identified "induction heads" within Transformer architectures than enable token-level pattern recognition and generalization (Olsson et al., 2022a). Beyond training dynamics, LayerNorm has also been shown to influence the expressivity of Attention-based models (Brody et al., 2023). From a model design perspective, entropy-guided variants of the Attention have been proposed to balance stability, computational efficiency, and privacy in large-scale models (Jha & Reagen, 2025).

## B   AUXILIARY RESULTS AND FURTHER DISCUSSIONS

In this section, we provide further discussions on possible extensions of our theoretical results. We discuss in Remark 5 the connection between the entry-wise Attention in Definition 1 to the standard Softmax Attention, in Remark 6 the extension of Assumption 1 beyond the rank-one setting, and in Remark 7 the possibility to relax the binary Gaussian mixture model in Definition 2 to, e.g., multi-class sub-gaussian mixture model.

**Remark 5** (On Softmax Attention). As already mentioned in the discussion after Definition 1, while Definition 1 corresponds to entry-wise Attention (such the sigmoid or ReLU Attention (Wortsman et al., 2023; Ramapuram et al., 2024)) instead of commonly used Softmax Attention, under the setting of Assumptions 1 and 3 and for input token drawn from the signal-plus-noise model in Definition 2 taking a truncated exponential function $f(t) = \min(\exp(t), C)$ for some $C > 0$ large, leads to approximately the same Attention matrix $\mathbf{A_X}$ as that of Softmax nonlinearity, up to a scaling factor.

Precisely, note from the proof of Lemma 1 below in Appendix C.2 that the $(i, j)$ entry of $\mathbf{X}^\top \mathbf{W}_K^\top \mathbf{W}_Q \mathbf{X}$ is given, for $i \neq j$, by

$$\mathbf{x}_i^\top \mathbf{W}_K^\top \mathbf{W}_Q \mathbf{x}_j = \mathbf{x}_i^\top \mathbf{x}_j + \mathbf{x}_i^\top \mathbf{w}_K \mathbf{w}_Q^\top \mathbf{x}_j$$
$$= \underbrace{\mathbf{z}_i^\top \mathbf{z}_j}_{O(\sqrt{p})} + \underbrace{y_i y_j \|\boldsymbol{\mu}\|^2 + (y_j \mathbf{z}_i + y_i \mathbf{z}_j)^\top \boldsymbol{\mu} + y_j \mathbf{z}_i^\top \mathbf{w}_K \mathbf{w}_Q^\top \boldsymbol{\mu} + y_i \boldsymbol{\mu}^\top \mathbf{w}_K \mathbf{w}_Q^\top \mathbf{z}_j + \mathbf{z}_i^\top \mathbf{w}_K \mathbf{w}_Q^\top \mathbf{z}_j + y_i y_j \boldsymbol{\mu}^\top \mathbf{w}_K \mathbf{w}_Q^\top \boldsymbol{\mu}}_{O(1)},$$

and for $i = j$, by

$$\mathbf{x}_i^\top \mathbf{W}_K^\top \mathbf{W}_Q \mathbf{x}_i = \|\mathbf{x}_i\|^2 + \mathbf{x}_i^\top \mathbf{w}_K \mathbf{w}_Q^\top \mathbf{x}_i$$
$$= \underbrace{\|\mathbf{z}_i\|^2}_{O(p)} + \underbrace{\|\boldsymbol{\mu}\|^2 + 2y_i \mathbf{z}_i^\top \boldsymbol{\mu} + y_i \mathbf{z}_i^\top \mathbf{w}_K \mathbf{w}_Q^\top \boldsymbol{\mu} + y_i \boldsymbol{\mu}^\top \mathbf{w}_K \mathbf{w}_Q^\top \mathbf{z}_i + \mathbf{z}_i^\top \mathbf{w}_K \mathbf{w}_Q^\top \mathbf{z}_i + \boldsymbol{\mu}^\top \mathbf{w}_K \mathbf{w}_Q^\top \boldsymbol{\mu}}_{O(1)}.$$

As such, the $i$th column of $\mathbf{X}^\top \mathbf{W}_K^\top \mathbf{W}_Q \mathbf{X} / \sqrt{p}$ (on which the Softmax function is applied) writes

$$\mathbf{X}^\top \mathbf{W}_K^\top \mathbf{W}_Q \mathbf{x}_i / \sqrt{p} = \frac{1}{\sqrt{p}} \begin{bmatrix} \mathbf{z}_1^\top \mathbf{z}_i \\ \vdots \\ \|\mathbf{z}_i\|^2 = p + O(\sqrt{p}) \\ \vdots \\ \mathbf{z}_n^\top \mathbf{z}_i \end{bmatrix} + O_{\|\cdot\|_\infty}(p^{-1/2}), \tag{17}$$

for $\mathbf{z}_i \sim \mathcal{N}(\mathbf{0}, \mathbf{I}_p)$, where we denote $O_{\|\cdot\|_\infty}(p^{-1/2})$ for random vector of infinity norm $O(p^{-1/2})$ with high probability. As such, for $j \neq i$, we have, conditioned on $\mathbf{z}_i$ that $\mathbf{z}_j^\top \mathbf{z}_i / \sqrt{p} \sim \mathcal{N}(0, \|\mathbf{z}_i\|^2/p)$ which is approximately $\mathcal{N}(0, 1)$ for $p$ large. Thus, by Taylor-expansion of $f(t) = \min\{\exp(t), C\}$, we have, for large enough $C$ that with high probability

$$\min \left\{ \exp \left( \mathbf{x}_j^\top \mathbf{W}_K^\top \mathbf{W}_Q \mathbf{x}_i / \sqrt{p} \right), C \right\} = \exp \left( \mathbf{z}_j^\top \mathbf{z}_i / \sqrt{p} \right) + O(p^{-1/2}), \tag{18}$$

for $j \neq i$ and similarly

$$\min \left\{ \exp \left( \mathbf{x}_i^\top \mathbf{W}_K^\top \mathbf{W}_Q \mathbf{x}_i / \sqrt{p} \right), C \right\} = C, \tag{19}$$

as a consequence of the fact that $\|\mathbf{z}_i\|^2 / \sqrt{p} \simeq \mathbb{E}[\|\mathbf{z}_i\|^2] / \sqrt{p} = \sqrt{p}$.

Also, we have

$$\sum_{j=1}^{n} \exp\left(\mathbf{x}_j^\top \mathbf{W}_K^\top \mathbf{W}_Q \mathbf{x}_i / \sqrt{p}\right) = \sum_{j \neq i} \exp\left(\mathbf{z}_j^\top \mathbf{z}_i / \sqrt{p}\right) + O(\sqrt{p}) + C = n\mathbb{E}[\exp(\mathcal{N}(0,1))] + O(\sqrt{p}), \quad (20)$$

so that for truncated Softmax function Softmax : $\mathbb{R}^n \to \mathbb{R}^n$ defined as

$$\text{Softmax}(\mathbf{z}) = \frac{1}{\sum_{j=1}^{n} \min\{\exp(z_j), C\}} \begin{bmatrix} \min\{\exp(z_1), C\} \\ \vdots \\ \min\{\exp(z_n), C\} \end{bmatrix}, \quad (21)$$

we have that the normalization factor for each column of $\mathbf{X}^\top \mathbf{W}_K^\top \mathbf{W}_Q \mathbf{X} / \sqrt{p}$ is asymptotically the same, and that Assumptions 1 and 3, the entry-wise truncated exponential function $f(t) = \min(\exp(t), C)$ leads to approximately the same Attention matrix $\mathbf{A_X}$ as the Softmax Attention, up to a scaling factor. Nonetheless, it remains unclear how this entry-wise approximation of Attention matrix could translate into, e.g., the approximation of Softmax using truncated exponential function in terms of the in-context memorization error in Definition 3. See Figure 7 below for numerical results showing such examples and counterexamples.

**Remark 6** (Extension beyond rank-one setting). While we consider in Assumption 1 that the Attention weights admits a full-plus-low-rank decomposition of the form $\mathbf{W}_K^\top \mathbf{W}_Q = \mathbf{I}_p + \mathbf{w}_K \mathbf{w}_Q^\top$, with $\mathbf{w}_K \mathbf{w}_Q^\top$ being of rank one, it is possible to extend the analysis beyond the rank-one setting and consider a low-rank part of rank $K$ (with $K$ fixed as $n, p \to \infty$). Notably, in that setting, the linearization result in Lemma 1 must be modified so that the term $\mathbf{U}_K \mathbf{\Sigma_K} \mathbf{V}_Q^\top$ takes account of the rank-$K$ structure in the product $\mathbf{W}_K^\top \mathbf{W}_Q$.

**Remark 7** (Extension beyond binary Gaussian signal-plus-noise model). The Gaussian signal-plus-noise model in Definition 2 can be extended (at least) in the following two ways.

1. By considering more sophisticated structure in the statistical signal, for instance with $\mathbf{X} = \mathbf{M}\mathbf{J}^\top$ where $\mathbf{M} = [\boldsymbol{\mu}_1, ..., \boldsymbol{\mu}_K] \in \mathbb{R}^{p \times K}$ is the matrix containing the means of the $K > 2$ classes, and $\mathbf{J} = [\mathbf{j}_1, ..., \mathbf{j}_K] \in \mathbb{R}^{n \times K}$ is the canonical vector of class $C_k \in \{1, ... K\}$, with $[\mathbf{j}_k]_i = 1$ if $\mathbf{x}_i$ belongs to class $C_k$ and zero otherwise.

2. By considering not necessarily Gaussian noise in the input tokens. An example is the sub-gaussian distribution that is symmetric in law. It has been long known in RMT that eigenspectra of large random matrices enjoy universal properties for Gaussian and non-Gaussian entries (Tao et al., 2010; Pastur & Shcherbina, 2011), and we expect that most of our technical results also hold for sub-gaussian distributions, see for example Lemma 2 below.

**Remark 8** (Extension to multi-head Attention). Compared to the single-head Attention module studied here in Definition 1, the multi-head Attention architecture is more popularly used. As long as the number of heads remains constant and does not scale with the dimensions $n, p$, the concatenation and linear combination in multi-head Attention (with possibly different weights) do not introduce any fundamentally novel technical difficulty in the analysis, and we expect the main qualitative behavior and scaling laws characterized in this paper to remain valid in the multi-head setting. In particular, we believe that in the case of multi-task context (i.e., more that one directions to be learned from the in-context data, that are correlated with different targets, as discussed in Remark 7), multi-head Attention should be considered for the in-context memorization to be efficient. A complete treatment of the multi-head extension is left for future work.

## C    MATHEMATICAL PROOFS

In this section, we present the proofs of the technical results in this paper. We first recall in Appendix C.1 a few lemmas that will be used in the proofs. The proof of Lemma 1 is given in Appendix C.2, the proof of Proposition 1 is given in Appendix C.3, the proof of Theorem 1 is given in Appendix C.4, and finally the proof of Proposition 2 in Appendix C.5.

### C.1    USEFUL LEMMAS

**Lemma 2** (Spectral norm of kernel random matrix, (Fan & Montanari, 2019)). *For a random matrix* $\mathbf{Z} \in \mathbb{R}^{p \times n}$ *having i.i.d. sub-gaussian entries that are symmetric in law, of zero mean and unit variance,*

*and function $f : \mathbb{R} \to \mathbb{R}$ such that $|f(x)| \leq C_1 \exp(C_2|x|)$ for some constants $C_1, C_2 > 0$, the random kernel matrix*

$$\mathbf{K} = f(\mathbf{Z}^\top \mathbf{Z} / \sqrt{p}) / \sqrt{p} - \mathrm{diag}(\cdot) \in \mathbb{R}^{n \times n}, \tag{22}$$

*satisfies, with high probability as $n, p \to \infty$ at the same pace, that*

1. $\|\mathbf{K}\| = O(1)$ *if* $\mathbb{E}_{\xi \sim \mathcal{N}(0,1)}[f(\xi)] = 0$*; and*

2. $\|\mathbf{K}\| = O(\sqrt{p})$ *with* $\|\mathbf{K} - \mathbb{E}[f(\xi)]\mathbf{1}_n \mathbf{1}_n^\top / \sqrt{p}\| = O(1)$ *otherwise.*

**Lemma 3** (Matrix norm controls)*. We have the following facts on the operator norm of matrices and Hadamard product between matrices.*

1. *For $\mathbf{A} \in \mathbb{R}^{n \times n}$, we have $\|\mathbf{A}\|_{\max} \leq \|\mathbf{A}\| \leq n\|\mathbf{A}\|_{\max}$ with $\|\mathbf{A}\|_{\max} \equiv \max_{i,j} |A_{ij}|$.*

2. *For $\mathbf{A}, \mathbf{B} \in \mathbb{R}^{N \times n}$, we have $\|\mathbf{A} \odot \mathbf{B}\| \leq \max(\sqrt{n}, \sqrt{N})\|\mathbf{A}\|_{\max} \cdot \|\mathbf{B}\|$.*

3. *If $\mathbf{A} \in \mathbb{R}^{N \times n}$ is of rank one with $\mathbf{A} = \mathbf{u}\mathbf{v}^\top$, $\mathbf{u} \in \mathbb{R}^N$, $\mathbf{v} \in \mathbb{R}^n$, we have $\mathbf{A} \odot \mathbf{B} = \mathrm{diag}(\mathbf{u})\mathbf{B}\,\mathrm{diag}(\mathbf{v})$ so that*

$$\|\mathbf{A} \odot \mathbf{B}\| \leq \|\mathbf{u}\|_\infty \cdot \|\mathbf{v}\|_\infty \cdot \|\mathbf{B}\|, \tag{23}$$

*see (Ba et al., 2022, Fact 13). More generally, if $\mathbf{A}$ is of rank $K$ with $\mathbf{A} = \sum_{k=1}^K \mathbf{u}_k \mathbf{w}_K^\top$, we similarly have*

$$\|\mathbf{A} \odot \mathbf{B}\| = \|(\sum_{k=1}^K \mathbf{u}_k \mathbf{w}_k^\top) \odot \mathbf{B}\| = \|\sum_{k=1}^K (\mathbf{u}_k \mathbf{w}_k^\top) \odot \mathbf{B}\|$$

$$\leq \sum_{k=1}^K \|(\mathbf{u}_k \mathbf{w}_k^\top) \odot \mathbf{B}\| \leq \sum_{k=1}^K \|\mathbf{u}_k\|_\infty \cdot \|\mathbf{w}_k\|_\infty \cdot \|\mathbf{B}\|.$$

## C.2 PROOF OF LEMMA 1

Here, we present the proof of Lemma 1 by "linearizing" the nonlinear kernel matrix

$$\mathbf{K_X} = f(\mathbf{X}^\top \mathbf{W}_K^\top \mathbf{W}_Q \mathbf{X} / \sqrt{p}) / \sqrt{p} \in \mathbb{R}^{n \times n}, \tag{24}$$

defined in (1) of Definition 1.

To start, note that for the binary mixture model in Definition 2 and under Assumptions 1 and 3, we have $\mathbf{x}_i = y_i \boldsymbol{\mu} + \mathbf{z}_i$ and $\mathbf{W}_K^\top \mathbf{W}_Q = \mathbf{I}_p + \mathbf{w}_K \mathbf{w}_Q^\top$, so that for $i \neq j$,

$$\mathbf{x}_i^\top \mathbf{W}_K^\top \mathbf{W}_Q \mathbf{x}_j = \mathbf{x}_i^\top \mathbf{x}_j + \mathbf{x}_i^\top \mathbf{w}_K \mathbf{w}_Q^\top \mathbf{x}_j$$

$$= \underbrace{\mathbf{z}_i^\top \mathbf{z}_j}_{O(\sqrt{p})} + \underbrace{y_i y_j \|\boldsymbol{\mu}\|^2 + (y_j \mathbf{z}_i + y_i \mathbf{z}_j)^\top \boldsymbol{\mu} + y_j \mathbf{z}_i^\top \mathbf{w}_K \mathbf{w}_Q^\top \boldsymbol{\mu} + y_i \boldsymbol{\mu}^\top \mathbf{w}_K \mathbf{w}_Q^\top \mathbf{z}_j + \mathbf{z}_i^\top \mathbf{w}_K \mathbf{w}_Q^\top \mathbf{z}_j + y_i y_j \boldsymbol{\mu}^\top \mathbf{w}_K \mathbf{w}_Q^\top \boldsymbol{\mu}}_{O(1)},$$

for $y_i, y_j \in \{\pm 1\}$ and independent $\mathbf{z}_i, \mathbf{z}_j \sim \mathcal{N}(\mathbf{0}, \mathbf{I}_p)$, where we used the fact that $\max\{\|\boldsymbol{\mu}\|, \|\mathbf{w}_K\|, \|\mathbf{w}_Q\|\} = O(1)$ under Assumption 3. Similarly, for $i = j$,

$$\mathbf{x}_i^\top \mathbf{W}_K^\top \mathbf{W}_Q \mathbf{x}_i = \|\mathbf{x}_i\|^2 + \mathbf{x}_i^\top \mathbf{w}_K \mathbf{w}_Q^\top \mathbf{x}_i$$

$$= \underbrace{\|\mathbf{z}_i\|^2}_{O(p)} + \underbrace{\|\boldsymbol{\mu}\|^2 + 2y_i \mathbf{z}_i^\top \boldsymbol{\mu} + y_i \mathbf{z}_i^\top \mathbf{w}_K \mathbf{w}_Q^\top \boldsymbol{\mu} + y_i \boldsymbol{\mu}^\top \mathbf{w}_K \mathbf{w}_Q^\top \mathbf{z}_i + \mathbf{z}_i^\top \mathbf{w}_K \mathbf{w}_Q^\top \mathbf{z}_i + \boldsymbol{\mu}^\top \mathbf{w}_K \mathbf{w}_Q^\top \boldsymbol{\mu}}_{O(1)},$$

where we used the fact that for any deterministic vector $\mathbf{w} \in \mathbb{R}^p$ of bounded norm, one has $\mathbf{z}_i^\top \mathbf{w} \sim \mathcal{N}(0, \|\mathbf{w}\|^2)$. As a consequence, we can Taylor-expand the smooth function $f$ in $\mathbf{K_X}$ defined in (1) of Definition 1. We first treat its non-diagonal entry $(i, j)$, for $i \neq j$, as

$$\sqrt{p}[\mathbf{K_X}]_{ij} = f(\mathbf{z}_i^\top \mathbf{z}_j / \sqrt{p}) + f'(\mathbf{z}_i^\top \mathbf{z}_j / \sqrt{p})(y_i y_j \|\boldsymbol{\mu}\|^2 + (y_j \mathbf{z}_i + y_i \mathbf{z}_j)^\top \boldsymbol{\mu} + y_j \mathbf{z}_i^\top \mathbf{w}_K \mathbf{w}_Q^\top \boldsymbol{\mu} + y_i \boldsymbol{\mu}^\top \mathbf{w}_K \mathbf{w}_Q^\top \mathbf{z}_j$$

$$+ \mathbf{z}_i^\top \mathbf{w}_K \mathbf{w}_Q^\top \mathbf{z}_j + y_i y_j \boldsymbol{\mu}^\top \mathbf{w}_K \mathbf{w}_Q^\top \boldsymbol{\mu}) / \sqrt{p} + O(p^{-1}),$$

and for its diagonal entries as

$$\sqrt{p}[\mathbf{K_X}]_{ii} = f(\|\mathbf{z}_i\|^2 / \sqrt{p}) + f'(\|\mathbf{z}_i\|^2 / \sqrt{p})(\|\boldsymbol{\mu}\|^2 + 2y_i \mathbf{z}_i^\top \boldsymbol{\mu} + y_i \mathbf{z}_i^\top \mathbf{w}_K \mathbf{w}_Q^\top \boldsymbol{\mu} + y_i \boldsymbol{\mu}^\top \mathbf{w}_K \mathbf{w}_Q^\top \mathbf{z}_i + \mathbf{z}_i^\top \mathbf{w}_K \mathbf{w}_Q^\top \mathbf{z}_i + \boldsymbol{\mu}^\top \mathbf{w}_K \mathbf{w}_Q^\top \boldsymbol{\mu}) / \sqrt{p}$$

$$+ O(p^{-1}).$$

Note that under Assumption 2, one has $\lim_{t\to\infty} f(t) < \infty$ so that as $n, p \to \infty$,

$$\sqrt{p}[\mathbf{K_X}]_{ii} = O(1).$$

This leads to the following spectral norm approximation of $\mathbf{K_X}$ as

$$\sqrt{p}\mathbf{K_X} = \underbrace{f(\mathbf{Z}^\top\mathbf{Z}/\sqrt{p}) - \mathrm{diag}(\cdot)}_{O_{\|\cdot\|}(\sqrt{p})} + \underbrace{f'(\mathbf{Z}^\top\mathbf{Z}/\sqrt{p}) \odot (\|\boldsymbol{\mu}\|^2\mathbf{y}\mathbf{y}^\top + \mathbf{y}\boldsymbol{\mu}^\top\mathbf{Z} + \mathbf{Z}^\top\boldsymbol{\mu}\mathbf{y}^\top + \boldsymbol{\mu}^\top\mathbf{w}_Q \cdot \mathbf{Z}^\top\mathbf{w}_K\mathbf{y}^\top + \boldsymbol{\mu}^\top\mathbf{w}_K \cdot \mathbf{y}\mathbf{w}_Q^\top\mathbf{Z})/\sqrt{p}}_{O_{\|\cdot\|}(\sqrt{p})}$$

$$+ \underbrace{\boldsymbol{\mu}^\top\mathbf{w}_K\mathbf{w}_Q^\top\boldsymbol{\mu} \cdot f'(\mathbf{Z}^\top\mathbf{Z}/\sqrt{p}) \odot (\mathbf{y}\mathbf{y}^\top)/\sqrt{p}}_{O_{\|\cdot\|}(\sqrt{p})} + \underbrace{\mathbf{Z}^\top\mathbf{w}_K\mathbf{w}_Q^\top\mathbf{Z} \odot f'(\mathbf{Z}^\top\mathbf{Z}/\sqrt{p})/\sqrt{p}}_{O_{\|\cdot\|}(\sqrt{p})} - \mathrm{diag}(\cdot) + O_{\|\cdot\|}(1)$$

$$= \underbrace{f(\mathbf{Z}^\top\mathbf{Z}/\sqrt{p}) - \mathrm{diag}(\cdot)}_{O_{\|\cdot\|}(\sqrt{p})} + \underbrace{a_1((\|\boldsymbol{\mu}\|^2 + \boldsymbol{\mu}^\top\mathbf{w}_K\mathbf{w}_Q^\top\boldsymbol{\mu})\mathbf{y}\mathbf{y}^\top + \mathbf{y}\boldsymbol{\mu}^\top\mathbf{Z} + \mathbf{Z}^\top\boldsymbol{\mu}\mathbf{y}^\top + \boldsymbol{\mu}^\top\mathbf{w}_Q \cdot \mathbf{Z}^\top\mathbf{w}_K\mathbf{y}^\top + \boldsymbol{\mu}^\top\mathbf{w}_K \cdot \mathbf{y}\mathbf{w}_Q^\top\mathbf{Z})/\sqrt{p}}_{O_{\|\cdot\|}(\sqrt{p})}$$

$$+ \underbrace{a_1\mathbf{Z}^\top\mathbf{w}_K\mathbf{w}_Q^\top\mathbf{Z}/\sqrt{p}}_{O_{\|\cdot\|}(\sqrt{p})} - \mathrm{diag}(\cdot) + O_{\|\cdot\|}(1),$$

where we used the fact that under Assumption 2 for $\mathbb{E}[f(\xi)] = 0$ and $\mathbb{E}[f'(\xi)] = a_1 \neq 0$, it follows from Lemma 2 that $f(\mathbf{Z}^\top\mathbf{Z}/\sqrt{p}) - \mathrm{diag}(\cdot) = O_{\|\cdot\|}(\sqrt{p})$ and $f'(\mathbf{Z}^\top\mathbf{Z}/\sqrt{p}) = \mathbb{E}[f'(\xi)]\mathbf{1}_n\mathbf{1}_n^\top + O_{\|\cdot\|}(\sqrt{p})$, and then Item 3 of Lemma 3.

Let $\mathbf{U}_K = [\mathbf{y}, \mathbf{Z}^\top\boldsymbol{\mu}, \mathbf{Z}^\top\mathbf{w}_K]/\sqrt{p} \in \mathbb{R}^{n\times 3}$, $\mathbf{V}_Q = [\mathbf{y}, \mathbf{Z}^\top\boldsymbol{\mu}, \mathbf{Z}^\top\mathbf{w}_Q]/\sqrt{p} \in \mathbb{R}^{n\times 3}$, and

$$\Sigma_{\mathbf{K}} = a_1 \begin{bmatrix} \|\boldsymbol{\mu}\|^2 + \boldsymbol{\mu}^\top\mathbf{w}_K\mathbf{w}_Q^\top\boldsymbol{\mu} & 1 & \boldsymbol{\mu}^\top\mathbf{w}_K \\ 1 & 0 & 0 \\ \boldsymbol{\mu}^\top\mathbf{w}_Q & 0 & 1 \end{bmatrix} \in \mathbb{R}^{3\times 3}. \tag{25}$$

Putting everything in matrix form, we conclude the proof of Lemma 1.

## C.3 PROOF OF PROPOSITION 1

For the sake of presentation, we provide here the derivation of the Deterministic Equivalent for the resolvent[2]

$$\mathbf{Q}(\gamma) = \left(\frac{1}{p}\mathbf{K}\mathbf{Z}^\top\mathbf{Z}\mathbf{K} + \frac{\gamma}{c}\mathbf{I}_n\right)^{-1}, \quad \gamma > 0, \tag{26}$$

where we denote, with a slight abuse of notation that $\mathbf{K} = \mathbf{K}_N = f(\mathbf{Z}^\top\mathbf{Z}/\sqrt{p})/\sqrt{p} - \mathrm{diag}(\cdot)$ for the noise-only kernel matrix $\mathbf{K}_N$ defined in Lemma 1. The result in Proposition 1 can be obtained with a simple scaling.

Consider the following normalized traces involving $\mathbf{Q}(\gamma)$ defined in (26):

$$\frac{1}{n}\mathrm{tr}\,\mathbf{Q}(\gamma), \quad \frac{1}{p}\mathrm{tr}(\mathbf{Q}(\gamma)\mathbf{K}), \quad \frac{1}{p}\mathrm{tr}(\mathbf{Q}(\gamma)\mathbf{K}\cdot\mathbf{Z}^\top\mathbf{Z}/p) \quad \frac{1}{p}\mathrm{tr}(\mathbf{K}\mathbf{Q}(\gamma)\mathbf{K}), \quad \frac{1}{p}\mathrm{tr}(\mathbf{Z}^\top\mathbf{Z}/p\cdot\mathbf{K}\mathbf{Q}(\gamma)\mathbf{K}\cdot\mathbf{Z}^\top\mathbf{Z}/p),$$

for which we shall subsequently prove that

$$\frac{1}{n}\mathrm{tr}\,\mathbf{Q}(\gamma) - m(\gamma) \to 0, \quad \frac{1}{p}\mathrm{tr}(\mathbf{Q}(\gamma)\mathbf{K}) - \delta_1(\gamma) \to 0, \quad \frac{1}{p}\mathrm{tr}(\mathbf{Q}(\gamma)\mathbf{K}\mathbf{Z}^\top\mathbf{Z}/p) - \delta_2(\gamma) \to 0,$$

$$\frac{1}{p}\mathrm{tr}(\mathbf{K}\mathbf{Q}(\gamma)\mathbf{K}) - \delta_3(\gamma) \to 0, \quad \frac{1}{p}\mathrm{tr}(\mathbf{Z}^\top\mathbf{Z}/p \cdot \mathbf{K}\mathbf{Q}(\gamma)\mathbf{K} \cdot \mathbf{Z}^\top\mathbf{Z}/p) - \delta_4(\gamma) \to 0,$$

$$\tag{27}$$

in probability as $n, p \to \infty$, where $m(\gamma)$ and $\delta_1(\gamma), \delta_2(\gamma), \delta_3(\gamma), \delta_4(\gamma)$ are Stieltjes transforms satisfying the following self-consistent system of equations

$$\begin{cases} m(\gamma) &= \left(\frac{\gamma}{c} + \frac{v}{c} + \frac{a_1^2}{c^2} - \mathbf{v}^\top\mathbf{T}(\gamma)\mathbf{v}\right)^{-1} \\ c\delta_1(\gamma) &= -m(\gamma)\mathbf{v}^\top\mathbf{T}(\gamma)\mathbf{v}_1 \\ c\delta_2(\gamma) &= \mathbf{v}_2^\top\mathbf{T}(\gamma)\mathbf{v}_1 + c\delta_1(\gamma)\left(1 - \mathbf{v}_2^\top\mathbf{T}(\gamma)\mathbf{v}\right) \\ c\delta_3(\gamma) &= \mathbf{v}_1^\top\mathbf{T}(\gamma)\mathbf{v}_1 + \frac{c^2\delta_1^2(\gamma)}{m(\gamma)} \\ c\delta_4(\gamma) &= \mathbf{v}_4^\top\mathbf{T}(\gamma)\mathbf{v}_4 + m(\gamma)\left(\mathbf{v}_4^\top\mathbf{T}(\gamma)\mathbf{v} - \frac{a_1}{c}\right)^2 \end{cases} \tag{28}$$

---

[2]Note that this is not the same $\mathbf{Q}(\gamma)$ as in (3) of Definition 3. It is used here for the sake of notational convenience and only within the proof of Proposition 1.

where we denote

$$\mathbf{T}(\gamma) \equiv \Delta_0(\gamma)(\mathbf{I}_6 + \Lambda_0 \Delta_0(\gamma))^{-1} \in \mathbb{R}^{6\times6}, \tag{29}$$

that is symmetric, for

$$\Delta_0(\gamma) \equiv \begin{bmatrix} \frac{m(\gamma)}{c} & \frac{a_1}{c}m(\gamma) & \delta_1(\gamma) & a_1\delta_1(\gamma) & \delta_2(\gamma) & a_1\delta_2(\gamma) \\ \frac{a_1}{c}m(\gamma) & \frac{v}{c}m(\gamma) & a_1\delta_1(\gamma) & v\delta_1(\gamma) & a_1\delta_2(\gamma) & v\delta_2(\gamma) \\ \delta_1(\gamma) & a_1\delta_1(\gamma) & \delta_3(\gamma) & a_1\delta_3(\gamma) & \frac{1}{c}(1-\frac{\gamma}{c}m(\gamma)) & \frac{a_1}{c}(1-\frac{\gamma}{c}m(\gamma)) \\ a_1\delta_1(\gamma) & v\delta_1(\gamma) & a_1\delta_3(\gamma) & v\delta_3(\gamma) & \frac{a_1}{c}(1-\frac{\gamma}{c}m(\gamma)) & \frac{v}{c}(1-\frac{\gamma}{c}m(\gamma)) \\ \delta_2(\gamma) & a_1\delta_2(\gamma) & \frac{1}{c}(1-\frac{\gamma}{c}m(\gamma)) & \frac{a_1}{c}(1-\frac{\gamma}{c}m(\gamma)) & \delta_4(\gamma) & a_1\delta_4(\gamma) \\ a_1\delta_2(\gamma) & v\delta_2(\gamma) & \frac{a_1}{c}(1-\frac{\gamma}{c}m(\gamma)) & \frac{v}{c}(1-\frac{\gamma}{c}m(\gamma)) & a_1\delta_4(\gamma) & v\delta_4(\gamma) \end{bmatrix} \in \mathbb{R}^{6\times6}, \tag{30}$$

and

$$\Lambda_0 = \begin{bmatrix} \frac{a_1^2}{c^2}(c+1) & a_1/c & a_1/c & 0 & a_1 & 0 \\ a_1/c & 1 & 1 & 0 & 0 & 0 \\ a_1/c & 1 & 1 & 0 & 0 & 0 \\ 0 & 0 & 0 & 0 & 0 & 0 \\ a_1 & 0 & 0 & 0 & 0 & 0 \\ 0 & 0 & 0 & 0 & 0 & 0 \end{bmatrix} \in \mathbb{R}^{6\times6}, \tag{31}$$

that are both symmetric, as well as

$$\begin{cases} \mathbf{v} = \begin{bmatrix} \frac{a_1^2}{c^2}(1+c) & \frac{a_1}{c} & \frac{a_1}{c} & 0 & 0 & 1 \end{bmatrix}^\top \in \mathbb{R}^6 \\ \mathbf{v}_1 = \begin{bmatrix} 0 & 1 & 0 & 0 & 0 & 0 \end{bmatrix}^\top \in \mathbb{R}^6 \\ \mathbf{v}_2 = \begin{bmatrix} 1 & 0 & 0 & 0 & 0 & 0 \end{bmatrix}^\top \in \mathbb{R}^6 \\ \mathbf{v}_4 = \begin{bmatrix} \frac{a_1}{c} & 1 & 1 & 0 & 0 & 0 \end{bmatrix}^\top \in \mathbb{R}^6. \end{cases} \tag{32}$$

### C.3.1 PRELIMINARIES

First, let us introduce some notations and preliminary results that will be used in the proof of Proposition 1.

Following (Couillet & Liao, 2022, Section 4.3.3), we can decompose, up to permutation, the nonlinear kernel matrix $\mathbf{K}$ as

$$\mathbf{K} = \begin{bmatrix} \mathbf{K}_{-i} & f(\mathbf{Z}_{-i}^\top \mathbf{z}_i/\sqrt{p})/\sqrt{p} \\ f(\mathbf{z}_i^\top \mathbf{Z}_{-i}/\sqrt{p})/\sqrt{p} & 0 \end{bmatrix} = \begin{bmatrix} \mathbf{K}_{-i} & f(\boldsymbol{\alpha}_{-i})/\sqrt{p} \\ f(\boldsymbol{\alpha}_{-i})^\top/\sqrt{p} & 0 \end{bmatrix}, \tag{33}$$

where we denote $\mathbf{K}_{-i} \equiv f(\mathbf{Z}_{-i}^\top \mathbf{Z}_{-i}/\sqrt{p})/\sqrt{p} - \mathrm{diag}(\cdot) \in \mathbb{R}^{(n-1)\times(n-1)}$,

$$\boldsymbol{\alpha}_{-i} \equiv \mathbf{Z}_{-i}^\top \mathbf{z}_i/\sqrt{p} \in \mathbb{R}^{n-1}, \tag{34}$$

for $\mathbf{Z}_{-i} \in \mathbb{R}^{p\times(n-1)}$ the Gaussian random matrix $\mathbf{Z}$ with its $i$th column removed, and $\mathbf{z}_i \in \mathbb{R}^p$ is the $i$th column of $\mathbf{Z}$. Note that in the large $p$ limit, the random vector $\boldsymbol{\alpha}_{-i}$ is standard Gaussian $\mathcal{N}(\mathbf{0}, \mathbf{I}_{n-1})$.

Denote the shortcut $\check{\mathbf{Z}} = \frac{1}{p}\mathbf{Z}^\top\mathbf{Z}$, we can similarly decompose $\check{\mathbf{Z}}$ as

$$\check{\mathbf{Z}} = \frac{1}{p}\mathbf{Z}^\top\mathbf{Z} = \begin{bmatrix} \mathbf{Z}_{-i}^\top\mathbf{Z}_{-i}/p & \boldsymbol{\alpha}_{-i}/\sqrt{p} \\ \boldsymbol{\alpha}_{-i}^\top/\sqrt{p} & 1 \end{bmatrix} + O_{\|\cdot\|}(p^{-1/2}) = \begin{bmatrix} \check{\mathbf{Z}}_{-i} & \boldsymbol{\alpha}_{-i}/\sqrt{p} \\ \boldsymbol{\alpha}_{-i}^\top/\sqrt{p} & 1 \end{bmatrix} + O_{\|\cdot\|}(p^{-1/2}), \tag{35}$$

where the $O_{\|\cdot\|}(p^{-1/2})$ error term is due to the approximation $\|\mathbf{z}_i\|^2/p = 1 + O(p^{-1/2})$ with a CLT argument.

Note that, by the decomposition $f(x) = a_1 x + f_{>1}(x)$ for $a_1$ the first Hermite polynomial of $f$ as defined in Assumption 2, we have

$$f(\boldsymbol{\alpha}_{-i}) = a_1\boldsymbol{\alpha}_{-i} + f_{>1}(\boldsymbol{\alpha}_{-i}), \tag{36}$$

and, for any $\mathbf{A} \in \mathbb{R}^{(n-1)\times(n-1)}$ independent of $\boldsymbol{\alpha}_{-i}$,

$$\frac{1}{p}f(\boldsymbol{\alpha}_{-i})^\top\mathbf{A}\boldsymbol{\alpha}_{-i} = a_1 \cdot \frac{1}{p}\mathrm{tr}\,\mathbf{A} + o(1), \quad \frac{1}{p}f(\boldsymbol{\alpha}_{-i})^\top\mathbf{A}f(\boldsymbol{\alpha}_{-i}) = v \cdot \frac{1}{p}\mathrm{tr}\,\mathbf{A} + o(1). \tag{37}$$

In particular, for $\mathbf{A} = \mathbf{I}_{n-1}$, we get $\frac{1}{p} f(\boldsymbol{\alpha}_{-i})^\top \mathbf{A} \boldsymbol{\alpha}_{-i} = \frac{a_1}{c} + o(1)$ and $\frac{1}{p} f(\boldsymbol{\alpha}_{-i})^\top \mathbf{A} f(\boldsymbol{\alpha}_{-i}) = \frac{\nu}{c} + o(1)$, where $c = \lim p/n$ as in Assumption 3.

Further denote

$$\mathbf{U}_0 = \begin{bmatrix} \boldsymbol{\alpha}_{-i} & f(\boldsymbol{\alpha}_{-i}) & \mathbf{K}_{-i}^\perp \boldsymbol{\alpha}_{-i} & \mathbf{K}_{-i}^\perp f(\boldsymbol{\alpha}_{-i}) & \mathbf{K}_{-i}^\perp \check{\mathbf{Z}}_{-i}^\perp \boldsymbol{\alpha}_{-i} & \mathbf{K}_{-i}^\perp \check{\mathbf{Z}}_{-i}^\perp f(\boldsymbol{\alpha}_{-i}) \end{bmatrix} / \sqrt{p} \in \mathbb{R}^{(n-1)\times 6}, \quad (38)$$

as well as

$$\mathbf{K}_{-i}^\perp = \{f((\mathbf{z}_j^\perp)^\top \mathbf{z}_k^\perp / \sqrt{p})\}_{j,k\neq i} / \sqrt{p} - \mathrm{diag}(\cdot) \in \mathbb{R}^{(n-1)\times(n-1)}, \quad \check{\mathbf{Z}}_{-i}^\perp \equiv \{(\mathbf{z}_j^\perp)^\top \mathbf{z}_k^\perp / p\}_{j,k\neq i} \in \mathbb{R}^{(n-1)\times(n-1)}, \quad (39)$$

where, for given $i$,

$$\mathbf{z}_j^\perp \equiv \mathbf{z}_j - \frac{\mathbf{z}_i^\top \mathbf{z}_j}{\|\mathbf{z}_i\|} \frac{\mathbf{z}_i}{\|\mathbf{z}_i\|}, \quad (40)$$

that is orthogonal to and asymptotically independent of $\mathbf{z}_i^\top \mathbf{z}_j / \|\mathbf{z}_i\| \approx [\boldsymbol{\alpha}_{-i}]_j$ in (34). For the two "leave-one-out" matrices $\mathbf{K}_{-i}$ and $\check{\mathbf{Z}}_{-i}$ defined in (33) and (35), we have the following result.

**Lemma 4** (Approximations of $\mathbf{K}_{-i}$ and $\check{\mathbf{Z}}_{-i}$, (Couillet & Liao, 2022, Section 4.3.3)). *For $\mathbf{K}_{-i}$ and $\check{\mathbf{Z}}_{-i}$ defined respectively in (33) and (35), we have the following approximations in spectral norm:*

1. $\mathbf{K}_{-i} = \mathbf{K}_{-i}^\perp + \frac{a_1}{p} \boldsymbol{\alpha}_{-i} \boldsymbol{\alpha}_{-i}^\top + o_{\|\cdot\|}(1)$*; and*

2. $\check{\mathbf{Z}}_{-i} = \check{\mathbf{Z}}_{-i}^\perp + \frac{1}{p} \boldsymbol{\alpha}_{-i} \boldsymbol{\alpha}_{-i}^\top + o_{\|\cdot\|}(1)$*;*

*for $\mathbf{K}_{-i}^\perp$ and $\check{\mathbf{Z}}_{-i}^\perp$ defined in (39) that is asymptotically independent of $\boldsymbol{\alpha}_{-i}$ in (34).*

With these preliminary results at hand, we are ready to derive a Deterministic Equivalent for $\mathbf{Q}(\gamma)$ defined in (26).

### C.3.2 SELF-CONSISTENT EQUATION FOR $m(z)$

Here we present the derivation for the Deterministic Equivalent (of the diagonal entries) of $\mathbf{Q}(z)$. With the block decomposition of $\mathbf{K}$ and $\check{\mathbf{Z}}$ in (33) and (35), we obtain for $\mathbf{Q}(\gamma) \equiv \mathbf{Q}$ in (26) (where we drop the argument $\gamma$) that

$$\mathbf{Q}^{-1} \equiv \mathbf{K}\check{\mathbf{Z}}\mathbf{K} + \frac{\gamma}{c} \mathbf{I}_n = \begin{bmatrix} [\mathbf{Q}^{-1}]_{11} & [\mathbf{Q}^{-1}]_{12} \equiv [\mathbf{Q}^{-1}]_{21}^\top \\ [\mathbf{Q}^{-1}]_{21} & [\mathbf{Q}^{-1}]_{22} \end{bmatrix} \quad (41)$$

with

$$[\mathbf{Q}^{-1}]_{11} \equiv \mathbf{K}_{-i}\check{\mathbf{Z}}_{-i}\mathbf{K}_{-i} + \frac{1}{p} f(\boldsymbol{\alpha}_{-i})\boldsymbol{\alpha}_{-i}^\top \mathbf{K}_{-i} + \frac{1}{p} \mathbf{K}_{-i}\boldsymbol{\alpha}_{-i} f(\boldsymbol{\alpha}_{-i})^\top + \frac{1}{p} f(\boldsymbol{\alpha}_{-i}) f(\boldsymbol{\alpha}_{-i})^\top + \frac{\gamma}{c} \mathbf{I}_{n-1},$$

$$[\mathbf{Q}^{-1}]_{21} \equiv \frac{1}{\sqrt{p}} f(\boldsymbol{\alpha}_{-i})^\top \check{\mathbf{Z}}_{-i} \mathbf{K}_{-i} + \frac{a_1}{c} \frac{f(\boldsymbol{\alpha}_{-i})^\top}{\sqrt{p}},$$

$$[\mathbf{Q}^{-1}]_{22} \equiv \frac{1}{p} f(\boldsymbol{\alpha}_{-i})^\top \check{\mathbf{Z}}_{-i} f(\boldsymbol{\alpha}_{-i}) + \frac{\gamma}{c},$$

for which we have, per Lemma 4,

1. $\mathbf{K}_{-i} = \mathbf{K}_{-i}^\perp + \frac{a_1}{p} \boldsymbol{\alpha}_{-i} \boldsymbol{\alpha}_{-i}^\top + o_{\|\cdot\|}(1)$; and

2. $\check{\mathbf{Z}}_{-i} = \check{\mathbf{Z}}_{-i}^\perp + \frac{1}{p} \boldsymbol{\alpha}_{-i} \boldsymbol{\alpha}_{-i}^\top + o_{\|\cdot\|}(1)$;

where $\mathbf{K}_{-i}^\perp$ and $\check{\mathbf{Z}}_{-i}^\perp$ as defined in (39) that are both asymptotically independent of $\boldsymbol{\alpha}_{-i}$. This allows for the first approximation of $[\mathbf{Q}^{-1}]_{22}$ as

$$[\mathbf{Q}^{-1}]_{22} = \frac{1}{p} f(\boldsymbol{\alpha}_{-i})^\top \check{\mathbf{Z}}_{-i} f(\boldsymbol{\alpha}_{-i}) + \frac{\gamma}{c} = \frac{1}{p} f(\boldsymbol{\alpha}_{-i})^\top \left( \frac{1}{p} \boldsymbol{\alpha}_{-i} \boldsymbol{\alpha}_{-i}^\top + \check{\mathbf{Z}}_{-i}^\perp \right) f(\boldsymbol{\alpha}_{-i}) + \frac{\gamma}{c} + o(1)$$

$$= \frac{a_1^2}{c^2} + \frac{\nu}{c} + \frac{\gamma}{c} + o(1). \quad (42)$$

Let

$$\mathbf{Q}_{-i}^{\perp} = \left(\mathbf{K}_{-i}^{\perp}\check{\mathbf{Z}}_{-i}^{\perp}\mathbf{K}_{-i}^{\perp} + \frac{\gamma}{c}\mathbf{I}_{n-1}\right)^{-1} \in \mathbb{R}^{(n-1)\times(n-1)}, \tag{43}$$

for $\mathbf{K}_{-i}^{\perp}, \check{\mathbf{Z}}_{-i}^{\perp}$ defined in (39), so that $\mathbf{Q}_{-i}^{\perp}$ is asymptotically independent of $\boldsymbol{\alpha}_{-i}$ satisfying $\operatorname{tr}(\mathbf{Q} - \mathbf{Q}_{-i}^{\perp}) = O(1)$. We have the following approximation.

**Lemma 5** (Approximation of $\mathbf{U}_0^{\top}\mathbf{Q}_{-i}^{\perp}\mathbf{U}_0$). *For $\mathbf{Q}_{-i}^{\perp} \in \mathbb{R}^{(n-1)\times(n-1)}$ as in (43) and $\mathbf{U}_0 \in \mathbb{R}^{(n-1)\times 6}$ in (38), we have*

$$\mathbf{U}_0^{\top}\mathbf{Q}_{-i}^{\perp}\mathbf{U}_0 = \Delta_0(\gamma) + o_{\|\cdot\|}(1), \tag{44}$$

*with*

$$\Delta_0(z) \equiv \begin{bmatrix} \frac{m(\gamma)}{c} & \frac{a_1}{c}m(\gamma) & \delta_1(\gamma) & a_1\delta_1(\gamma) & \delta_2(\gamma) & a_1\delta_2(\gamma) \\ \frac{a_1}{c}m(\gamma) & \frac{v}{c}m(\gamma) & a_1\delta_1(\gamma) & v\delta_1(\gamma) & a_1\delta_2(\gamma) & v\delta_2(\gamma) \\ \delta_1(\gamma) & a_1\delta_1(\gamma) & \delta_3(\gamma) & a_1\delta_3(\gamma) & \frac{1}{c}(1-\frac{\gamma}{c}m(\gamma)) & \frac{a_1}{c}(1-\frac{\gamma}{c}m(\gamma)) \\ a_1\delta_1(\gamma) & v\delta_1(\gamma) & a_1\delta_3(\gamma) & v\delta_3(\gamma) & \frac{a_1}{c}(1-\frac{\gamma}{c}m(\gamma)) & \frac{v}{c}(1-\frac{\gamma}{c}m(\gamma)) \\ \delta_2(\gamma) & a_1\delta_2(\gamma) & \frac{1}{c}(1-\frac{\gamma}{c}m(\gamma)) & \frac{a_1}{c}(1-\frac{\gamma}{c}m(\gamma)) & \delta_4(\gamma) & a_1\delta_4(\gamma) \\ a_1\delta_2(\gamma) & v\delta_2(\gamma) & \frac{a_1}{c}(1-\frac{\gamma}{c}m(\gamma)) & \frac{v}{c}(1-\frac{\gamma}{c}m(\gamma)) & a_1\delta_4(\gamma) & v\delta_4(\gamma) \end{bmatrix} \in \mathbb{R}^{6\times 6},$$

$$\tag{45}$$

*as in (30), for $m(\gamma), \delta_1(\gamma), \delta_2(\gamma), \delta_3(\gamma), \delta_4(\gamma)$ as defined in (27) and (28).*

*Proof of Lemma 5.* Since $\mathbf{Q}_{-i}^{\perp}$ is asymptotically independent of

$$\mathbf{U}_0 = \begin{bmatrix} \boldsymbol{\alpha}_{-i} & f(\boldsymbol{\alpha}_{-i}) & \mathbf{K}_{-i}^{\perp}\boldsymbol{\alpha}_{-i} & \mathbf{K}_{-i}^{\perp}f(\boldsymbol{\alpha}_{-i}) & \mathbf{K}_{-i}^{\perp}\check{\mathbf{Z}}_{-i}^{\perp}\boldsymbol{\alpha}_{-i} & \mathbf{K}_{-i}^{\perp}\check{\mathbf{Z}}_{-i}^{\perp}f(\boldsymbol{\alpha}_{-i}) \end{bmatrix} / \sqrt{p} \in \mathbb{R}^{(n-1)\times 6}, \tag{46}$$

we obtain

$$\mathbf{U}_0^{\top}\mathbf{Q}_{-i}^{\perp}\mathbf{U}_0 = \begin{bmatrix} \frac{1}{p}\operatorname{tr}\mathbf{Q} & a_1\frac{1}{p}\operatorname{tr}\mathbf{Q} & \frac{1}{p}\operatorname{tr}(\mathbf{QK}) & a_1\frac{1}{p}\operatorname{tr}(\mathbf{QK}) & \frac{1}{p}\operatorname{tr}(\mathbf{QK}\check{\mathbf{Z}}) & a_1\frac{1}{p}\operatorname{tr}(\mathbf{QK}\check{\mathbf{Z}}) \\ a_1\frac{1}{p}\operatorname{tr}\mathbf{Q} & v\frac{1}{p}\operatorname{tr}\mathbf{Q} & a_1\frac{1}{p}\operatorname{tr}(\mathbf{QK}) & v\frac{1}{p}\operatorname{tr}(\mathbf{QK}) & a_1\frac{1}{p}\operatorname{tr}(\mathbf{QK}\check{\mathbf{Z}}) & v\frac{1}{p}\operatorname{tr}(\mathbf{QK}\check{\mathbf{Z}}) \\ \frac{1}{p}\operatorname{tr}(\mathbf{QK}) & a_1\frac{1}{p}\operatorname{tr}(\mathbf{QK}) & \frac{1}{p}\operatorname{tr}(\mathbf{KQK}) & a_1\operatorname{tr}(\mathbf{KQK}) & \frac{1}{p}\operatorname{tr}(\mathbf{KQK}\check{\mathbf{Z}}) & a_1\frac{1}{p}\operatorname{tr}(\mathbf{KQK}\check{\mathbf{Z}}) \\ a_1\frac{1}{p}\operatorname{tr}(\mathbf{QK}) & v\frac{1}{p}\operatorname{tr}(\mathbf{QK}) & a_1\frac{1}{p}\operatorname{tr}(\mathbf{KQK}) & v\operatorname{tr}(\mathbf{KQK}) & a_1\frac{1}{p}\operatorname{tr}(\mathbf{KQK}\check{\mathbf{Z}}) & v\frac{1}{p}\operatorname{tr}(\mathbf{KQK}\check{\mathbf{Z}}) \\ \frac{1}{p}\operatorname{tr}(\check{\mathbf{Z}}\mathbf{KQ}) & a_1\frac{1}{p}\operatorname{tr}(\check{\mathbf{Z}}\mathbf{KQ}) & \frac{1}{p}\operatorname{tr}(\mathbf{KQK}\check{\mathbf{Z}}) & a_1\frac{1}{p}\operatorname{tr}(\mathbf{KQK}\check{\mathbf{Z}}) & \frac{1}{p}\operatorname{tr}(\check{\mathbf{Z}}\mathbf{KQK}\check{\mathbf{Z}}) & a_1\frac{1}{p}\operatorname{tr}(\check{\mathbf{Z}}\mathbf{KQK}\check{\mathbf{Z}}) \\ a_1\frac{1}{p}\operatorname{tr}(\check{\mathbf{Z}}\mathbf{KQ}) & v\frac{1}{p}\operatorname{tr}(\check{\mathbf{Z}}\mathbf{KQ}) & a_1\frac{1}{p}\operatorname{tr}(\mathbf{KQK}\check{\mathbf{Z}}) & v\frac{1}{p}\operatorname{tr}(\mathbf{KQK}\check{\mathbf{Z}}) & a_1\frac{1}{p}\operatorname{tr}(\check{\mathbf{Z}}\mathbf{KQK}\check{\mathbf{Z}}) & v\frac{1}{p}\operatorname{tr}(\check{\mathbf{Z}}\mathbf{KQK}\check{\mathbf{Z}}) \end{bmatrix} + o_{\|\cdot\|}(1)$$

$$= \Delta_0(\gamma) + o_{\|\cdot\|}(1),$$

where we recall from (27) that

$$m(\gamma) = \frac{1}{n}\operatorname{tr}\mathbf{Q}(\gamma) + o(1) = \frac{1}{n}\operatorname{tr}\mathbf{Q}_{-i}^{\perp}(\gamma) + o(1)$$

$$\delta_1(\gamma) = \frac{1}{p}\operatorname{tr}(\mathbf{Q}(\gamma)\mathbf{K}) + o(1) = \frac{1}{p}\operatorname{tr}\left(\mathbf{Q}_{-i}^{\perp}(\gamma)\mathbf{K}_{-i}^{\perp}\right) + o(1)$$

$$\delta_2(\gamma) = \frac{1}{p}\operatorname{tr}(\mathbf{Q}(\gamma)\mathbf{K}\check{\mathbf{Z}}) + o(1) = \frac{1}{p}\operatorname{tr}(\mathbf{Q}_{-i}^{\perp}(\gamma)\mathbf{K}_{-i}^{\perp}\check{\mathbf{Z}}_{-i}^{\perp}) + o(1)$$

$$\delta_3(\gamma) = \frac{1}{p}\operatorname{tr}(\mathbf{KQ}(\gamma)\mathbf{K}) + o(1) = \frac{1}{p}\operatorname{tr}(\mathbf{K}_{-i}^{\perp}\mathbf{Q}_{-i}^{\perp}(\gamma)\mathbf{K}_{-i}^{\perp}) + o(1)$$

$$\delta_4(\gamma) = \frac{1}{p}\operatorname{tr}(\check{\mathbf{Z}}\mathbf{KQ}(\gamma)\mathbf{K}\check{\mathbf{Z}}) + o(1) = \frac{1}{p}\operatorname{tr}(\check{\mathbf{Z}}_{-i}^{\perp}\mathbf{K}_{-i}^{\perp}\mathbf{Q}_{-i}^{\perp}(\gamma)\mathbf{K}_{-i}^{\perp}\check{\mathbf{Z}}_{-i}^{\perp}) + o(1),$$

and we use the fact that by (Silverstein & Bai, 1995, Lemma 2.6) and (Couillet & Liao, 2022, Lemma 2.9), when evaluating normalized traces forms as in (27) for $n, p$ large, we can ignore terms of finite rank inside the trace, by adding an error term $o(1)$ with high probability, as well as $\operatorname{tr}(\tilde{\mathbf{K}}\mathbf{Q}\tilde{\mathbf{K}}\check{\mathbf{Z}}\tilde{\mathbf{K}}) = \operatorname{tr}(\mathbf{Q}\tilde{\mathbf{K}}\check{\mathbf{Z}}\tilde{\mathbf{K}}) = \operatorname{tr}[\mathbf{Q}(\mathbf{Q}^{-1} - \gamma\mathbf{I}_n)] = \operatorname{tr}[\mathbf{I}_n - \gamma\mathbf{Q}] = \operatorname{tr}(\mathbf{I}_n) - \gamma\operatorname{tr}(\mathbf{Q}) = n - n\gamma m(\gamma) = n(1 - \gamma m(\gamma))$, This concludes the proof of Lemma 5. $\square$

Our objective is to compute the $(i, i)$th diagonal entries of the inverse $\mathbf{Q} = (\mathbf{K}\check{\mathbf{Z}}\mathbf{K} + \gamma\mathbf{I}_n)^{-1}$. Using the block inversion lemma, we get

$$[\mathbf{Q}]_{ii} = \left([\mathbf{Q}^{-1}]_{22} - [\mathbf{Q}^{-1}]_{21}([\mathbf{Q}^{-1}]_{11})^{-1}[\mathbf{Q}^{-1}]_{12}\right)^{-1}, \tag{47}$$

and

$$\mathbf{Q} = \begin{bmatrix} ([\mathbf{Q}^{-1}]_{11} - [\mathbf{Q}^{-1}]_{12}([\mathbf{Q}^{-1}]_{22})^{-1}[\mathbf{Q}^{-1}]_{21})^{-1} & -([\mathbf{Q}^{-1}]_{11})^{-1}[\mathbf{Q}^{-1}]_{12}[\mathbf{Q}]_{ii} \\ -[\mathbf{Q}]_{ii}[\mathbf{Q}^{-1}]_{21}([\mathbf{Q}^{-1}]_{11})^{-1} & [\mathbf{Q}]_{ii} \end{bmatrix}. \tag{48}$$

We start with the inverse $([\mathbf{Q}^{-1}]_{11})^{-1}$, for which we have the following result.

**Lemma 6** (Approximation of $([\mathbf{Q}^{-1}]_{11})^{-1}$). *For $[\mathbf{Q}^{-1}]_{11} \in \mathbb{R}^{(n-1)\times(n-1)}$ defined in* (41), *we have*

$$([\mathbf{Q}^{-1}]_{11})^{-1} = \mathbf{Q}_{-i}^{\perp} - \mathbf{Q}_{-i}^{\perp}\mathbf{U}_0 \left(\mathbf{I}_6 + \mathbf{\Lambda}_0\mathbf{\Delta}_0(\gamma)\right)^{-1}\mathbf{\Lambda}_0\mathbf{U}_0^{\top}\mathbf{Q}_{-i}^{\perp} + o_{\|\cdot\|}(1), \tag{49}$$

*where we recall $\mathbf{\Delta}_0(\gamma) \in \mathbb{R}^{6\times6}$ as in* (30), $\mathbf{U}_0 \in \mathbb{R}^{(n-1)\times6}$ *as defined in* (38), *and*

$$\mathbf{\Lambda}_0 = \begin{bmatrix} \frac{a_1^2}{c^2}(c+1) & a_1/c & a_1/c & 0 & a_1 & 0 \\ a_1/c & 1 & 1 & 0 & 0 & 0 \\ a_1/c & 1 & 1 & 0 & 0 & 0 \\ 0 & 0 & 0 & 0 & 0 & 0 \\ a_1 & 0 & 0 & 0 & 0 & 0 \\ 0 & 0 & 0 & 0 & 0 & 0 \end{bmatrix} \in \mathbb{R}^{6\times6}, \tag{50}$$

*as in* (31).

*Proof of Lemma 6.* Per its definition in (41), we have

$$[\mathbf{Q}^{-1}]_{11} \equiv \mathbf{K}_{-i}\check{\mathbf{Z}}_{-i}\mathbf{K}_{-i} + \frac{1}{p}f(\boldsymbol{\alpha}_{-i})\boldsymbol{\alpha}_{-i}^{\top}\mathbf{K}_{-i} + \frac{1}{p}\mathbf{K}_{-i}\boldsymbol{\alpha}_{-i}f(\boldsymbol{\alpha}_{-i})^{\top} + \frac{1}{p}f(\boldsymbol{\alpha}_{-i})f(\boldsymbol{\alpha}_{-i})^{\top} + \frac{\gamma}{c}\mathbf{I}_{n-1}$$

$$= \mathbf{K}_{-i}^{\perp}\check{\mathbf{Z}}_{-i}^{\perp}\mathbf{K}_{-i}^{\perp} + \mathbf{U}_0\mathbf{\Lambda}_0\mathbf{U}_0^{\top} + \frac{\gamma}{c}\mathbf{I}_{n-1} + o_{\|\cdot\|}(1),$$

for $\mathbf{U}_0, \mathbf{\Lambda}$ defined in (31), $\mathbf{K}_{-i}^{\perp}, \check{\mathbf{Z}}_{-i}^{\perp}$ defined in (39) such that $\mathbf{K}_{-i} = \mathbf{K}_{-i}^{\perp} + \frac{a_1}{p}\boldsymbol{\alpha}_{-i}\boldsymbol{\alpha}_{-i}^{\top} + o_{\|\cdot\|}(1)$ and $\check{\mathbf{Z}}_{-i} = \check{\mathbf{Z}}_{-i}^{\perp} + \frac{1}{p}\boldsymbol{\alpha}_{-i}\boldsymbol{\alpha}_{-i}^{\top} + o_{\|\cdot\|}(1)$ by Lemma 4. As such, by Woodbury identity,

$$([\mathbf{Q}^{-1}]_{11})^{-1} = \left(\mathbf{K}_{-i}^{\perp}\check{\mathbf{Z}}_{-i}^{\perp}\mathbf{K}_{-i}^{\perp} + \frac{\gamma}{c}\mathbf{I}_{n-1} + \mathbf{U}_0\mathbf{\Lambda}\mathbf{U}_0^{\top}\right)^{-1} + o_{\|\cdot\|}(1)$$

$$= \mathbf{Q}_{-i}^{\perp} - \mathbf{Q}_{-i}^{\perp}\mathbf{U}_0 \left(\mathbf{I}_6 + \mathbf{\Lambda}_0\mathbf{U}_0^{\top}\mathbf{Q}_{-i}^{\perp}\mathbf{U}_0\right)^{-1}\mathbf{\Lambda}_0\mathbf{U}_0^{\top}\mathbf{Q}_{-i}^{\perp} + o_{\|\cdot\|}(1),$$

for $\mathbf{Q}_{-i}^{\perp}$ defined in (43). Using Lemma 5 to approximate $\mathbf{U}_0^{\top}\mathbf{Q}_{-i}^{\perp}\mathbf{U}_0 = \mathbf{\Delta}_0(\gamma) + o_{\|\cdot\|}(1)$, we conclude the proof of Lemma 6. □

With Lemmas 5 and 6, we get the following (block-wise) approximation for $\mathbf{Q}$.

**Lemma 7** (Block approximation of $\mathbf{Q}$). *We have*

$$\mathbf{Q} = \begin{bmatrix} \mathbf{Q}_{-i}^{\perp} - \mathbf{Q}_{-i}^{\perp}\mathbf{U}_0(\mathbf{I}_6 + \mathbf{\Lambda}_1(\gamma)\mathbf{\Delta}_0(\gamma))^{-1}\mathbf{\Lambda}_1(\gamma)\mathbf{U}_0^{\top}\mathbf{Q}_{-i}^{\perp} & -m(\gamma)\mathbf{Q}_{-i}^{\perp}\mathbf{U}_0 \left(\mathbf{I}_6 + \mathbf{\Lambda}_0\mathbf{\Delta}_0(\gamma)\right)^{-1}\mathbf{v} \\ -m(\gamma)\mathbf{v}^{\top} \left(\mathbf{I}_6 + \mathbf{\Delta}_0(\gamma)\mathbf{\Lambda}_0\right)^{-1}\mathbf{U}_0^{\top}\mathbf{Q}_{-i}^{\perp} & [\mathbf{Q}]_{ii} \end{bmatrix} + o_{\|\cdot\|}(1), \tag{51}$$

*where we recall $\mathbf{\Delta}_0(\gamma) \in \mathbb{C}^{6\times6}$ as in* (30), $\mathbf{\Lambda}_0$ *as in* (31), $\mathbf{U}_0 \in \mathbb{R}^{(n-1)\times6}$ *as defined in* (38), *and*

$$\mathbf{\Lambda}_1(\gamma) = \mathbf{\Lambda}_0 - \left(\frac{a_1^2}{c^2} + \frac{v}{c} + \frac{\gamma}{c}\right)^{-1}\mathbf{v}\mathbf{v}^{\top} \in \mathbb{R}^{6\times6}, \quad \mathbf{v}^{\top} = \begin{bmatrix} \frac{a_1^2}{c^2}(1+c) & \frac{a_1}{c} & \frac{a_1}{c} & 0 & 0 & 1 \end{bmatrix} \in \mathbb{R}^6, \tag{52}$$

*as in* (32). *We also have, by* (52) *and Sherman–Morrison identity that*

$$\mathbf{\Delta}_0(\gamma)(\mathbf{I}_6 + \mathbf{\Lambda}_1(\gamma)\mathbf{\Delta}_0(\gamma))^{-1} = \mathbf{\Delta}_0(\gamma)\left(\mathbf{I}_6 + \mathbf{\Lambda}_0\mathbf{\Delta}_0(\gamma) - \left(\frac{a_1^2}{c^2} + \frac{v}{c} + \frac{\gamma}{c}\right)^{-1}\mathbf{v}\mathbf{v}^{\top}\mathbf{\Delta}_0(\gamma)\right)^{-1}$$

$$= \mathbf{\Delta}_0(\gamma)(\mathbf{I}_6 + \mathbf{\Lambda}_0\mathbf{\Delta}_0(\gamma))^{-1} + \frac{\mathbf{\Delta}_0(\gamma)(\mathbf{I}_6 + \mathbf{\Lambda}_0\mathbf{\Delta}_0(\gamma))^{-1}\mathbf{v}\mathbf{v}^{\top}\mathbf{\Delta}_0(\gamma)(\mathbf{I}_6 + \mathbf{\Lambda}_0\mathbf{\Delta}_0(\gamma))^{-1}}{\frac{a_1^2}{c^2} + \frac{v}{c} + \frac{\gamma}{c} - \mathbf{v}^{\top}\mathbf{\Delta}_0(\gamma)(\mathbf{I}_6 + \mathbf{\Lambda}_0\mathbf{\Delta}_0(\gamma))^{-1}\mathbf{v}}, \tag{53}$$

*and*

$$\mathbf{U}_0^{\top}\left(\mathbf{Q}_{-i}^{\perp} - \mathbf{Q}_{-i}^{\perp}\mathbf{U}_0(\mathbf{I}_6 + \mathbf{\Lambda}_1(\gamma)\mathbf{\Delta}_0(\gamma))^{-1}\mathbf{\Lambda}_1(\gamma)\mathbf{U}_0^{\top}\mathbf{Q}_{-i}^{\perp}\right)\mathbf{U}_0 = \mathbf{\Delta}_0(\gamma)(\mathbf{I}_6 + \mathbf{\Lambda}_1(\gamma)\mathbf{\Delta}_0(\gamma))^{-1} + o_{\|\cdot\|}(1). \tag{54}$$

*Proof of Lemma 7.* We first work on $[\mathbf{Q}^{-1}]_{21}$ by expanding the term $\frac{1}{\sqrt{p}} f(\boldsymbol{\alpha}_{-i})^\top \check{\mathbf{Z}}_{-i}\mathbf{K}_{-i}$ as

$$\frac{f(\boldsymbol{\alpha}_{-i})^\top}{\sqrt{p}}\check{\mathbf{Z}}_{-i}\mathbf{K}_{-i} = \frac{f(\boldsymbol{\alpha}_{-i})^\top}{\sqrt{p}}(\check{\mathbf{Z}}_{-i}^\perp + \frac{1}{p}\boldsymbol{\alpha}_{-i}\boldsymbol{\alpha}_{-i}^\top)(\mathbf{K}_{-i}^\perp + \frac{a_1}{p}\boldsymbol{\alpha}_{-i}\boldsymbol{\alpha}_{-i}^\top) + o_{\|\cdot\|}(1)$$

$$= \frac{f(\boldsymbol{\alpha}_{-i})^\top}{\sqrt{p}}\check{\mathbf{Z}}_{-i}^\perp\mathbf{K}_{-i}^\perp + a_1\frac{n}{p}\frac{1}{\sqrt{p}}\boldsymbol{\alpha}_{-i}^\top\mathbf{K}_{-i}^\perp + a_1^2\frac{n}{p}\left(1+\frac{n}{p}\right)\frac{\boldsymbol{\alpha}_{-i}^\top}{\sqrt{p}} + o_{\|\cdot\|}(1),$$

so that

$$[\mathbf{Q}^{-1}]_{21} = \frac{f(\boldsymbol{\alpha}_{-i})^\top}{\sqrt{p}}\check{\mathbf{Z}}_{-i}\mathbf{K}_{-i} + a_1\frac{n}{p}\frac{f(\boldsymbol{\alpha}_{-i})^\top}{\sqrt{p}} + o_{\|\cdot\|}(1) = \mathbf{v}^\top\mathbf{U}_0^\top + o_{\|\cdot\|}(1), \tag{55}$$

with $\mathbf{v}\in\mathbb{R}^6$ defined in (52).

So that

$$[\mathbf{Q}]_{ii}[\mathbf{Q}^{-1}]_{21}([\mathbf{Q}^{-1}]_{11})^{-1} = m(\gamma)\mathbf{v}^\top\mathbf{U}_0^\top\left(\mathbf{Q}_{-i}^\perp - \mathbf{Q}_{-i}^\perp\mathbf{U}_0(\mathbf{I}_6+\boldsymbol{\Lambda}_0\boldsymbol{\Delta}_0(\gamma))^{-1}\boldsymbol{\Lambda}_0\mathbf{U}_0^\top\mathbf{Q}_{-i}^\perp\right) + o_{\|\cdot\|}(1)$$

$$= m(\gamma)\mathbf{v}^\top(\mathbf{I}_6+\boldsymbol{\Delta}_0(\gamma)\boldsymbol{\Lambda}_0)^{-1}\mathbf{U}_0^\top\mathbf{Q}_{-i}^\perp + o_{\|\cdot\|}(1).$$

Then, with the approximation of the inverse $([\mathbf{Q}^{-1}]_{11})^{-1}$ in Lemma 6 and that of $[\mathbf{Q}^{-1}]_{21}$ above, we obtain

$$([\mathbf{Q}^{-1}]_{11} - [\mathbf{Q}^{-1}]_{12}([\mathbf{Q}^{-1}]_{22})^{-1}[\mathbf{Q}^{-1}]_{21})^{-1} = (\mathbf{K}_{-i}^\perp\check{\mathbf{Z}}_{-i}^\perp\mathbf{K}_{-i}^\perp + \mathbf{U}_0\boldsymbol{\Lambda}_0\mathbf{U}_0^\top + \frac{\gamma}{c}\mathbf{I}_{n-1} - ([\mathbf{Q}^{-1}]_{22})^{-1}\mathbf{U}_0\mathbf{v}\mathbf{v}^\top\mathbf{U}_0^\top)^{-1} + o_{\|\cdot\|}(1)$$

$$= (\mathbf{K}_{-i}^\perp\check{\mathbf{Z}}_{-i}^\perp\mathbf{K}_{-i}^\perp + \mathbf{U}_0\boldsymbol{\Lambda}_1\mathbf{U}_0^\top + \frac{\gamma}{c}\mathbf{I}_{n-1})^{-1} + o_{\|\cdot\|}(1)$$

$$= \mathbf{Q}_{-i}^\perp - \mathbf{Q}_{-i}^\perp\mathbf{U}_0(\mathbf{I}_6+\boldsymbol{\Lambda}_1(\gamma)\boldsymbol{\Delta}_0(\gamma))^{-1}\boldsymbol{\Lambda}_1(\gamma)\mathbf{U}_0^\top\mathbf{Q}_{-i}^\perp + o_{\|\cdot\|}(1),$$

by (42) and Woodbury identity, for

$$\boldsymbol{\Lambda}_1(\gamma) = \boldsymbol{\Lambda}_0 - ([\mathbf{Q}^{-1}]_{22})^{-1}\mathbf{v}\mathbf{v}^\top + o_{\|\cdot\|}(1) = \boldsymbol{\Lambda}_0 - \left(\frac{a_1^2}{c^2}+\frac{v}{c}+\frac{\gamma}{c}\right)^{-1}\mathbf{v}\mathbf{v}^\top + o_{\|\cdot\|}(1), \tag{56}$$

as defined in (52). This concludes the proof of Lemma 7. $\square$

Following the same idea, we expand the quadratic form $[\mathbf{Q}^{-1}]_{21}([\mathbf{Q}^{-1}]_{11})^{-1}[\mathbf{Q}^{-1}]_{12}$ in (47) as

$$[\mathbf{Q}^{-1}]_{21}([\mathbf{Q}^{-1}]_{11})^{-1}[\mathbf{Q}^{-1}]_{12} = \mathbf{v}^\top\boldsymbol{\Delta}_0(\gamma)(\mathbf{I}_6+\boldsymbol{\Lambda}_0\boldsymbol{\Delta}_0(\gamma))^{-1}\mathbf{v} + o(1), \tag{57}$$

for $\mathbf{v}\in\mathbb{R}^6$ defined in (52). Plugging this approximation back to (47) and ignoring the terms in $o(1)$, we obtain the following self-consistent equation on $m(\gamma)$,

$$\frac{1}{m(\gamma)} = \frac{\gamma}{c} + \frac{v}{c} + \frac{a_1^2}{c^2} - \mathbf{v}^\top\mathbf{T}(\gamma)\mathbf{v}, \quad \mathbf{T}(\gamma) = \boldsymbol{\Delta}_0(\gamma)(\mathbf{I}_6+\boldsymbol{\Lambda}_0\boldsymbol{\Delta}_0(\gamma))^{-1}. \tag{58}$$

In the following, we determine the (self-consistent) equations for $\delta_1(\gamma), \delta_2(\gamma), \delta_3(\gamma)$ and $\delta_4(\gamma)$ in $\mathbf{T}(\gamma)$, so as to retrieve the final self-consistent equations in (27).

### C.3.3 ESTABLISHING SELF-CONSISTENT EQUATIONS FOR $\delta(\gamma)$S

Following the same idea above in Appendix C.3.2, we now establish self-consistent equations for the intermediate variables $\delta_1(\gamma), \delta_2(\gamma), \delta_3(\gamma), \delta_4(\gamma)$ defined in (27).

**Self-consistent equation for $\delta_1(\gamma)$.** We start with $\delta_1(\gamma) = \frac{1}{p}\operatorname{tr}(\mathbf{Q}(\gamma)\mathbf{K}) + o(1)$ by writing

$$\delta_1(\gamma) = \frac{1}{p}\operatorname{tr}(\mathbf{QK}) + o(1) = \frac{1}{p}\sum_{i=1}^n[\mathbf{QK}]_{ii} + o(1) = \frac{1}{c}[\mathbf{QK}]_{ii} + o(1)$$

$$= -[\mathbf{Q}]_{ii}\frac{1}{c}[\mathbf{Q}^{-1}]_{21}([\mathbf{Q}^{-1}]_{11})^{-1}f(\boldsymbol{\alpha}_{-i})/\sqrt{p} + o(1)$$

$$= -\frac{m(\gamma)}{c}\mathbf{v}^\top\mathbf{U}_0^\top([\mathbf{Q}^{-1}]_{11})^{-1}\mathbf{U}_0\mathbf{v}_1 + o(1) = -\frac{m(\gamma)}{c}\mathbf{v}^\top\boldsymbol{\Delta}_0(\gamma)(\mathbf{I}_6+\boldsymbol{\Lambda}_0\boldsymbol{\Delta}_0(\gamma))^{-1}\mathbf{v}_1 + o(1)$$

$$= -\frac{m(\gamma)}{c}\mathbf{v}^\top\mathbf{T}(\gamma)\mathbf{v}_1 + o(1),$$

for

$$\mathbf{v}_1^\top = \begin{bmatrix} 0 & 1 & 0 & 0 & 0 & 0 \end{bmatrix} \in \mathbb{R}^6. \tag{59}$$

where we used the fact that $f(\boldsymbol{\alpha}_{-i})/\sqrt{p} = \mathbf{U}_0\mathbf{v}_1$, (55), and Lemma 6.

**Self-consistent equation for $\delta_2(\gamma)$.** We consider now $\delta_2(\gamma) = \frac{1}{p}\operatorname{tr}(\mathbf{Q}(\gamma)\mathbf{K}\check{\mathbf{Z}}) + o(1)$ and write

$$\delta_2(\gamma) = \frac{1}{p}\operatorname{tr}(\mathbf{QK}\check{\mathbf{Z}}) + o(1) = \frac{1}{c}[\check{\mathbf{Z}}\mathbf{QK}]_{ii} + o(1)$$

$$= \frac{1}{c}\left[\boldsymbol{\alpha}_{-i}^{\top}/\sqrt{p} \quad 1\right]\begin{bmatrix} \mathbf{Q}_{-i}^{\perp} - \mathbf{Q}_{-i}^{\perp}\mathbf{U}_0(\mathbf{I}_6 + \boldsymbol{\Lambda}_1(\gamma)\boldsymbol{\Delta}_0(\gamma))^{-1}\boldsymbol{\Lambda}_1(\gamma)\mathbf{U}_0^{\top}\mathbf{Q}_{-i}^{\perp} & -m(\gamma)\mathbf{Q}_{-i}^{\perp}\mathbf{U}_0\,(\mathbf{I}_6 + \boldsymbol{\Lambda}_0\boldsymbol{\Delta}_0(\gamma))^{-1}\,\mathbf{v} \\ -m(\gamma)\mathbf{v}^{\top}\,(\mathbf{I}_6 + \boldsymbol{\Delta}_0(\gamma)\boldsymbol{\Lambda}_0)^{-1}\,\mathbf{U}_0^{\top}\mathbf{Q}_{-i}^{\perp} & [\mathbf{Q}]_{ii} \end{bmatrix}\begin{bmatrix} f(\boldsymbol{\alpha}_{-i})/\sqrt{p} \\ 0 \end{bmatrix} + o(1)$$

$$= \frac{1}{c}\left(\mathbf{v}_2^{\top}\boldsymbol{\Delta}_0(\gamma)(\mathbf{I}_6 + \boldsymbol{\Lambda}_1(\gamma)\boldsymbol{\Delta}_0(\gamma))^{-1}\mathbf{v}_1 - m(\gamma)\mathbf{v}^{\top}\boldsymbol{\Delta}_0(\gamma)(\mathbf{I}_6 + \boldsymbol{\Lambda}_0\boldsymbol{\Delta}_0(\gamma))^{-1}\mathbf{v}_1\right) + o(1)$$

$$= \frac{1}{c}\left((\mathbf{v}_2 - m(\gamma)\mathbf{v})^{\top}\boldsymbol{\Delta}_0(\gamma)(\mathbf{I}_6 + \boldsymbol{\Lambda}_0\boldsymbol{\Delta}_0(\gamma))^{-1}\mathbf{v}_1 + \frac{\mathbf{v}_2^{\top}\boldsymbol{\Delta}_0(\gamma)(\mathbf{I}_6 + \boldsymbol{\Lambda}_0\boldsymbol{\Delta}_0(\gamma))^{-1}\mathbf{v} \times \mathbf{v}^{\top}\boldsymbol{\Delta}_0(\gamma)(\mathbf{I}_6 + \boldsymbol{\Lambda}_0\boldsymbol{\Delta}_0(\gamma))^{-1}\mathbf{v}_1}{\frac{a_1^2}{c^2} + \frac{v}{c} + \gamma - \mathbf{v}^{\top}\boldsymbol{\Delta}_0(\gamma)(\mathbf{I}_6 + \boldsymbol{\Lambda}_0\boldsymbol{\Delta}_0(\gamma))^{-1}\mathbf{v}}\right) + o(1)$$

$$= \frac{1}{c}\left((\mathbf{v}_2 - m(\gamma)\mathbf{v})^{\top}\boldsymbol{\Delta}_0(\gamma)(\mathbf{I}_6 + \boldsymbol{\Lambda}_0\boldsymbol{\Delta}_0(\gamma))^{-1}\mathbf{v}_1 + m(\gamma)\mathbf{v}_2^{\top}\boldsymbol{\Delta}_0(\gamma)(\mathbf{I}_6 + \boldsymbol{\Lambda}_0\boldsymbol{\Delta}_0(\gamma))^{-1}\mathbf{v} \times \mathbf{v}^{\top}\boldsymbol{\Delta}_0(\gamma)(\mathbf{I}_6 + \boldsymbol{\Lambda}_0\boldsymbol{\Delta}_0(\gamma))^{-1}\mathbf{v}_1\right) + o(1)$$

$$= \frac{1}{c}\left(\mathbf{v}_2^{\top}\mathbf{T}(\gamma)\mathbf{v}_1 + c\delta_1(\gamma)\left(1 - \mathbf{v}_2^{\top}\mathbf{T}(\gamma)\mathbf{v}\right)\right) + o(1),$$

for $\mathbf{v}_1^{\top} = \begin{bmatrix} 0 & 1 & 0 & 0 & 0 & 0 \end{bmatrix} \in \mathbb{R}^6, \mathbf{v}_2^{\top} = \begin{bmatrix} 1 & 0 & 0 & 0 & 0 & 0 \end{bmatrix} \in \mathbb{R}^6$, where we used the fact that $\boldsymbol{\alpha}_{-i}/\sqrt{p} = \mathbf{U}_0\mathbf{v}_2$, $f(\boldsymbol{\alpha}_{-i})/\sqrt{p} = \mathbf{U}_0\mathbf{v}_1$, Lemma 7, and the relation in (53).

**Self-consistent equation for $\delta_3(\gamma)$.** We consider now $\delta_3(\gamma) = \frac{1}{p}\operatorname{tr}(\mathbf{K}\mathbf{Q}(\gamma)\mathbf{K}) + o(1)$ and write

$$\delta_3(\gamma) = \frac{1}{p}\operatorname{tr}(\mathbf{KQK}) + o(1) = \frac{1}{c}[\mathbf{KQK}]_{ii} + o(1)$$

$$= \frac{1}{c}\left[f(\boldsymbol{\alpha}_{-i})^{\top}/\sqrt{p} \quad 0\right]\begin{bmatrix} \mathbf{Q}_{-i}^{\perp} - \mathbf{Q}_{-i}^{\perp}\mathbf{U}_0(\mathbf{I}_6 + \boldsymbol{\Lambda}_1(\gamma)\boldsymbol{\Delta}_0(\gamma))^{-1}\boldsymbol{\Lambda}_1(\gamma)\mathbf{U}_0^{\top}\mathbf{Q}_{-i}^{\perp} & -m(\gamma)\mathbf{Q}_{-i}^{\perp}\mathbf{U}_0\,(\mathbf{I}_6 + \boldsymbol{\Lambda}_0\boldsymbol{\Delta}_0(\gamma))^{-1}\,\mathbf{v} \\ -m(\gamma)\mathbf{v}^{\top}\,(\mathbf{I}_6 + \boldsymbol{\Delta}_0(\gamma)\boldsymbol{\Lambda}_0)^{-1}\,\mathbf{U}_0^{\top}\mathbf{Q}_{-i}^{\perp} & [\mathbf{Q}]_{ii} \end{bmatrix}$$

$$\times \begin{bmatrix} f(\boldsymbol{\alpha}_{-i})/\sqrt{p} \\ 0 \end{bmatrix} + o(1) = \frac{1}{c}\mathbf{v}_1^{\top}\boldsymbol{\Delta}_0(\gamma)(\mathbf{I}_6 + \boldsymbol{\Lambda}_1(\gamma)\boldsymbol{\Delta}_0(\gamma))^{-1}\mathbf{v}_1 + o(1)$$

$$= \frac{1}{c}\left(\mathbf{v}_1^{\top}\boldsymbol{\Delta}_0(\gamma)(\mathbf{I}_6 + \boldsymbol{\Lambda}_0\boldsymbol{\Delta}_0(\gamma))^{-1}\mathbf{v}_1 + \frac{(\mathbf{v}_1^{\top}\boldsymbol{\Delta}_0(\gamma)(\mathbf{I}_6 + \boldsymbol{\Lambda}_0\boldsymbol{\Delta}_0(\gamma))^{-1}\mathbf{v})^2}{\frac{a_1^2}{c^2} + \frac{v}{c} + \gamma - \mathbf{v}^{\top}\boldsymbol{\Delta}_0(\gamma)(\mathbf{I}_6 + \boldsymbol{\Lambda}_0\boldsymbol{\Delta}_0(\gamma))^{-1}\mathbf{v}}\right) + o(1)$$

$$= \frac{1}{c}\left(\mathbf{v}_1^{\top}\boldsymbol{\Delta}_0(\gamma)(\mathbf{I}_6 + \boldsymbol{\Lambda}_0\boldsymbol{\Delta}_0(\gamma))^{-1}\mathbf{v}_1 + m(\gamma)(\mathbf{v}_1^{\top}\boldsymbol{\Delta}_0(\gamma)(\mathbf{I}_6 + \boldsymbol{\Lambda}_0\boldsymbol{\Delta}_0(\gamma))^{-1}\mathbf{v})^2\right) + o(1)$$

$$= \frac{1}{c}\left(\mathbf{v}_1^{\top}\mathbf{T}(\gamma)\mathbf{v}_1 + m(\gamma)(\mathbf{v}_1^{\top}\mathbf{T}(\gamma)\mathbf{v})^2\right) + o(1) = \frac{1}{c}\left(\mathbf{v}_1^{\top}\mathbf{T}(\gamma)\mathbf{v}_1 + \frac{c^2\delta_1^2(\gamma)}{m(\gamma)}\right) + o(1).$$

**Self-consistent equation for $\delta_4(\gamma)$.** We consider now $\delta_4(\gamma) = \frac{1}{p}\operatorname{tr}(\check{\mathbf{Z}}\mathbf{K}\mathbf{Q}(\gamma)\mathbf{K}\check{\mathbf{Z}}) + o(1)$ and write

$$\delta_4(\gamma) = \frac{1}{p}\operatorname{tr}(\check{\mathbf{Z}}\mathbf{KQK}\check{\mathbf{Z}}) + o(1) = \frac{1}{c}[\check{\mathbf{Z}}\mathbf{KQK}\check{\mathbf{Z}}]_{ii} + o(1)$$

$$= \frac{1}{c}\left[(\mathbf{K}_{-i}^{\perp}\boldsymbol{\alpha}_{-i} + \frac{a_1}{c}\boldsymbol{\alpha}_{-i} + f(\boldsymbol{\alpha}_{-i}))^{\top}/\sqrt{p} \quad \frac{a_1}{c}\right]$$

$$\times \begin{bmatrix} \mathbf{Q}_{-i}^{\perp} - \mathbf{Q}_{-i}^{\perp}\mathbf{U}_0(\mathbf{I}_6 + \boldsymbol{\Lambda}_1(\gamma)\boldsymbol{\Delta}_0(\gamma))^{-1}\boldsymbol{\Lambda}_1(\gamma)\mathbf{U}_0^{\top}\mathbf{Q}_{-i}^{\perp} & -m(\gamma)\mathbf{Q}_{-i}^{\perp}\mathbf{U}_0\,(\mathbf{I}_6 + \boldsymbol{\Lambda}_0\boldsymbol{\Delta}_0(\gamma))^{-1}\,\mathbf{v} \\ -m(\gamma)\mathbf{v}^{\top}\,(\mathbf{I}_6 + \boldsymbol{\Delta}_0(\gamma)\boldsymbol{\Lambda}_0)^{-1}\,\mathbf{U}_0^{\top}\mathbf{Q}_{-i}^{\perp} & [\mathbf{Q}]_{ii} \end{bmatrix}\begin{bmatrix} * \\ * \end{bmatrix} + o(1)$$

$$= \frac{1}{c}\left(\mathbf{v}_4^{\top}\boldsymbol{\Delta}_0(\gamma)(\mathbf{I}_6 + \boldsymbol{\Lambda}_1(\gamma)\boldsymbol{\Delta}_0(\gamma))^{-1}\mathbf{v}_4 - \frac{2a_1m(\gamma)}{c}\mathbf{v}_4^{\top}\boldsymbol{\Delta}_0(\gamma)(\mathbf{I}_6 + \boldsymbol{\Lambda}_0(\gamma)\boldsymbol{\Delta}_0(\gamma))^{-1}\mathbf{v} + \frac{a_1^2}{c^2}m(\gamma)\right) + o(1)$$

$$= \frac{1}{c}\left(\mathbf{v}_4^{\top}\boldsymbol{\Delta}_0(\gamma)(\mathbf{I}_6 + \boldsymbol{\Lambda}_0\boldsymbol{\Delta}_0(\gamma))^{-1}\mathbf{v}_4 + \frac{(\mathbf{v}_4^{\top}\boldsymbol{\Delta}_0(\gamma)(\mathbf{I}_6 + \boldsymbol{\Lambda}_0\boldsymbol{\Delta}_0(\gamma))^{-1}\mathbf{v})^2}{\frac{a_1^2}{c^2} + \frac{v}{c} + \gamma - \mathbf{v}^{\top}\boldsymbol{\Delta}_0(\gamma)(\mathbf{I}_6 + \boldsymbol{\Lambda}_0\boldsymbol{\Delta}_0(\gamma))^{-1}\mathbf{v}} - \frac{2a_1m(\gamma)}{c}\mathbf{v}_4^{\top}\boldsymbol{\Delta}_0(\gamma)(\mathbf{I}_6 + \boldsymbol{\Lambda}_0(\gamma)\boldsymbol{\Delta}_0(\gamma))^{-1}\mathbf{v}\right)$$

$$+ \frac{a_1^2}{c^2 \times c}m(\gamma) + o(1)$$

$$= \frac{1}{c}\left(\mathbf{v}_4^{\top}\boldsymbol{\Delta}_0(\gamma)(\mathbf{I}_6 + \boldsymbol{\Lambda}_0\boldsymbol{\Delta}_0(\gamma))^{-1}\mathbf{v}_4 + m(\gamma)\left(\mathbf{v}_4^{\top}\boldsymbol{\Delta}_0(\gamma)(\mathbf{I}_6 + \boldsymbol{\Lambda}_0\boldsymbol{\Delta}_0(\gamma))^{-1}\mathbf{v} - \frac{a_1}{c}\right)^2\right) + o(1)$$

$$= \frac{1}{c}\left(\mathbf{v}_4^{\top}\mathbf{T}(\gamma)\mathbf{v}_4 + m(\gamma)\left(\mathbf{v}_4^{\top}\mathbf{T}(\gamma)\mathbf{v} - \frac{a_1}{c}\right)^2\right) + o(1),$$

for

$$\mathbf{v}_4^\top = \begin{bmatrix} \frac{a_1}{c} & 1 & 1 & 0 & 0 & 0 \end{bmatrix} \in \mathbb{R}^6. \tag{60}$$

Putting these together, we obtain the system of equations as in (28).

We thus conclude the proof of Proposition 1.

## C.4  PROOF OF THEOREM 1

Here, we provide detailed derivations of Theorem 1 on the Deterministic Equivalent of the in-context memorization error $E$ in (6) Definition 3. To do this, recall the following structured nonlinear resolvent

$$\mathbf{Q}(\gamma) = \left( \frac{1}{n} \mathbf{K}_{\mathbf{X}}^\top \mathbf{X}^\top \mathbf{X} \mathbf{K}_{\mathbf{X}} + \gamma \mathbf{I}_n \right)^{-1}, \tag{61}$$

in (7) of Definition 3.

First note that by Lemma 1, we have

$$\mathbf{K}_{\mathbf{X}} = \mathbf{K}_N + \mathbf{U}_K \Sigma_{\mathbf{K}} \mathbf{V}_Q^\top + O_{\|\cdot\|}(n^{-1/2}), \tag{62}$$

for $\Sigma_{\mathbf{K}} \in \mathbb{R}^{3 \times 3}$ defined in (9). Similarly, under Assumption 3, we have

$$\frac{1}{n} \mathbf{X}^\top \mathbf{X} = \frac{1}{n} \mathbf{Z}^\top \mathbf{Z} + \mathbf{U}_K \Sigma_{\mathbf{X}} \mathbf{U}_K^\top = c\check{\mathbf{Z}} + \mathbf{U}_K \Sigma_{\mathbf{X}} \mathbf{U}_K^\top, \quad \Sigma_{\mathbf{X}} \equiv c \begin{bmatrix} \|\boldsymbol{\mu}\|^2 & 1 & 0 \\ 1 & 0 & 0 \\ 0 & 0 & 0 \end{bmatrix}, \tag{63}$$

that is of bounded norm with probability one as $n, p \to \infty$ at the same rate. As such, we have

$$\mathbf{Q}(\gamma) = \left( \frac{1}{n} \mathbf{K}_N \mathbf{Z}^\top \mathbf{Z} \mathbf{K}_N + \mathbf{U}\Sigma\mathbf{U}^\top + \gamma \mathbf{I}_n \right)^{-1} + O_{\|\cdot\|}(n^{-\frac{1}{2}})$$

$$= \left( \frac{1}{n} \mathbf{K}_N \mathbf{Z}^\top \mathbf{Z} \mathbf{K}_N + \gamma \mathbf{I}_n \right)^{-1} + O_{\|\cdot\|}(n^{-\frac{1}{2}})$$

$$- \left( \frac{1}{n} \mathbf{K}_N \mathbf{Z}^\top \mathbf{Z} \mathbf{K}_N + \gamma \mathbf{I}_n \right)^{-1} \mathbf{U} \left( \Sigma^{-1} + \mathbf{U}^\top \left( \frac{1}{n} \mathbf{K}_N \mathbf{Z}^\top \mathbf{Z} \mathbf{K}_N + \gamma \mathbf{I}_n \right)^{-1} \mathbf{U} \right)^{-1} \mathbf{U}^\top \left( \frac{1}{n} \mathbf{K}_N \mathbf{Z}^\top \mathbf{Z} \mathbf{K}_N + \gamma \mathbf{I}_n \right)^{-1},$$

by Woodbury identity, for

$$\mathbf{U} = \begin{bmatrix} \frac{1}{n} \mathbf{K}_N \mathbf{Z}^\top \mathbf{Z} \mathbf{U}_K & \mathbf{K}_N \mathbf{U}_K & \mathbf{V}_Q \end{bmatrix} \in \mathbb{R}^{n \times 9}, \tag{64}$$

with $\mathbf{U}_K \in \mathbb{R}^{n \times 3}$ and $\mathbf{V}_Q \in \mathbb{R}^{n \times 3}$ defined in Lemma 1, and

$$\Sigma = \begin{bmatrix} \mathbf{0}_3 & \mathbf{0}_3 & \Sigma_{\mathbf{K}} \\ \mathbf{0}_3 & \Sigma_{\mathbf{X}} & \Sigma_{\mathbf{X}} \mathbf{U}_K^\top \mathbf{U}_K \Sigma_{\mathbf{K}} \\ \Sigma_{\mathbf{K}}^\top & \Sigma_{\mathbf{K}}^\top \mathbf{U}_K^\top \mathbf{U}_K \Sigma_{\mathbf{X}} & \Sigma_{\mathbf{K}}^\top (\mathbf{U}_K^\top \frac{1}{n} \mathbf{Z}^\top \mathbf{Z} \mathbf{U}_K + \mathbf{U}_K^\top \mathbf{U}_K \Sigma_{\mathbf{X}} \mathbf{U}_K^\top \mathbf{U}_K) \Sigma_{\mathbf{K}} \end{bmatrix} \in \mathbb{R}^{9 \times 9}, \tag{65}$$

Our objective of interest is the the memorization error $E$ defined in (7) of Definition 3 as

$$E = -\frac{\gamma^2}{n} \frac{\partial \mathbf{y}^\top \mathbf{Q}(\gamma) \mathbf{y}}{\partial \gamma}. \tag{66}$$

Note that $\mathbf{y}/\sqrt{p}$ is the first column of $\mathbf{V}_Q$ and thus the seventh column of $\mathbf{U}$ defined in (64), so that

$$\frac{1}{n} \mathbf{y}^\top \mathbf{Q}(\gamma) \mathbf{y} = c \cdot \mathbf{e}_7^\top \mathbf{U}^\top \mathbf{Q}(\gamma) \mathbf{U} \mathbf{e}_7$$

$$= c \cdot \mathbf{e}_7^\top \mathbf{U}^\top \left( \frac{1}{n} \mathbf{K}_N \mathbf{Z}^\top \mathbf{Z} \mathbf{K}_N + \gamma \mathbf{I}_n \right)^{-1} \mathbf{U} \cdot \left( \mathbf{I}_9 + \Sigma \mathbf{U}^\top \left( \frac{1}{n} \mathbf{K}_N \mathbf{Z}^\top \mathbf{Z} \mathbf{K}_N + \gamma \mathbf{I}_n \right)^{-1} \mathbf{U} \right)^{-1} \mathbf{e}_7 + O(n^{-\frac{1}{2}}), \tag{67}$$

where $\mathbf{e}_7 \in \mathbb{R}^9$ is the canonical vector at location seven.

We have the following approximation for the above objective of interest.

**Lemma 8** (Further approximations). *For $\Sigma$ defined in (65) and $\mathbf{U}$ in (64), we have the following approximations in spectral norm holds with high probability as $n, p \to \infty$ with $p/n \to c \in (0, \infty)$,*

$$\Sigma = \Lambda + O_{\|\cdot\|}(n^{-\frac{1}{2}}) \tag{68}$$

$$\mathbf{U}^\top \left( \frac{1}{p} \mathbf{K}_N \mathbf{Z}^\top \mathbf{Z} \mathbf{K}_N + \frac{\gamma}{c} \mathbf{I}_n \right)^{-1} \mathbf{U} = \Delta(\gamma) + O_{\|\cdot\|}(n^{-\frac{1}{2}}), \tag{69}$$

*with* $\Lambda = \begin{bmatrix} \mathbf{0}_3 & \mathbf{0}_3 & \Sigma_{\mathbf{K}} \\ \mathbf{0}_3 & \Sigma_{\mathbf{X}} & [\Lambda]_{2,3} \\ \Sigma_{\mathbf{K}}^\top & [\Lambda]_{2,3}^\top & [\Lambda]_{3,3} \end{bmatrix} \in \mathbb{R}^{9 \times 9}$ *and* $\Delta(\gamma) = \begin{bmatrix} [\Delta(\gamma)]_{1,1} & [\Delta(\gamma)]_{1,2} & [\Delta(\gamma)]_{1,3} \\ [\Delta(\gamma)]_{1,2}^\top & [\Delta(\gamma)]_{2,2} & [\Delta(\gamma)]_{2,3} \\ [\Delta(\gamma)]_{1,3}^\top & [\Delta(\gamma)]_{2,3}^\top & [\Delta(\gamma)]_{3,3} \end{bmatrix} \in \mathbb{C}^{9 \times 9}$ *both three-by-three block symmetric matrices with corresponding blocks given by*

$$[\Lambda]_{2,3} = a_1 \begin{bmatrix} (\|\boldsymbol{\mu}\|^2 + 1) T_1 & \|\boldsymbol{\mu}\|^2 & \boldsymbol{\mu}^\top \mathbf{w}_K (\|\boldsymbol{\mu}\|^2 + 1) \\ T_1 & 1 & \boldsymbol{\mu}^\top \mathbf{w}_K \\ 0 & 0 & 0 \end{bmatrix}$$

$$[\Lambda]_{3,3} = a_1^2 \begin{bmatrix} \frac{2+c+\|\boldsymbol{\mu}\|^2}{c} T_1^2 + \frac{1+c}{c} T_1 + \frac{1+c}{c} \boldsymbol{\mu}^\top \mathbf{w}_Q \left( \boldsymbol{\mu}^\top \mathbf{w}_K + \boldsymbol{\mu}^\top \mathbf{w}_Q \|\mathbf{w}_K\|^2 \right) & (*) & (*) \\ \frac{1+c+\|\boldsymbol{\mu}\|^2}{c} T_1 & 1 + \frac{\|\boldsymbol{\mu}\|^2}{c} & \frac{1+c+\|\boldsymbol{\mu}\|^2}{c} \boldsymbol{\mu}^\top \mathbf{w}_K \\ \frac{2+c+\|\boldsymbol{\mu}\|^2}{c} \boldsymbol{\mu}^\top \mathbf{w}_K T_1 + \frac{1+c}{c} (\boldsymbol{\mu}^\top \mathbf{w}_K + \boldsymbol{\mu}^\top \mathbf{w}_Q \|\mathbf{w}_K\|^2) & (*) & \frac{2+c+\|\boldsymbol{\mu}\|^2}{c} (\boldsymbol{\mu}^\top \mathbf{w}_K)^2 + \frac{1+c}{c} \|\mathbf{w}_K\|^2 \end{bmatrix}$$

*for*

$$T_1 = \|\boldsymbol{\mu}\|^2 + \boldsymbol{\mu}^\top \mathbf{w}_K \boldsymbol{\mu}^\top \mathbf{w}_Q, \tag{70}$$

*and* $\Sigma_{\mathbf{K}} \equiv a_1 \begin{bmatrix} T_1 & 1 & \boldsymbol{\mu}^\top \mathbf{w}_K \\ 1 & 0 & 0 \\ \boldsymbol{\mu}^\top \mathbf{w}_Q & 0 & 1 \end{bmatrix} \in \mathbb{R}^{3 \times 3}$ *defined in (9) of Lemma 1,* $\Sigma_{\mathbf{X}} \equiv c \begin{bmatrix} \|\boldsymbol{\mu}\|^2 & 1 & 0 \\ 1 & 0 & 0 \\ 0 & 0 & 0 \end{bmatrix} \in \mathbb{R}^{3 \times 3}$, *as well as*

$$[\Delta(\gamma)]_{1,1} = \begin{bmatrix} c^2 \delta_4(\gamma) & 0 & 0 \\ 0 & c^2 \|\boldsymbol{\mu}\|^2 \delta_7(\gamma) & c^2 \boldsymbol{\mu}^\top \mathbf{w}_K \delta_7(\gamma) \\ 0 & (*) & c^2 \|\mathbf{w}_K\|^2 \delta_7(\gamma) \end{bmatrix} \in \mathbb{R}^{3 \times 3}$$

$$[\Delta(\gamma)]_{1,2} = \begin{bmatrix} 1 - \frac{\gamma}{c} m(\gamma) & 0 & 0 \\ 0 & c \|\boldsymbol{\mu}\|^2 \delta_4(\gamma) & c \boldsymbol{\mu}^\top \mathbf{w}_K \delta_4(\gamma) \\ 0 & c \boldsymbol{\mu}^\top \mathbf{w}_K \delta_4(\gamma) & c \|\mathbf{w}_K\|^2 \delta_4(\gamma) \end{bmatrix} \in \mathbb{R}^{3 \times 3}$$

$$[\Delta(\gamma)]_{1,3} = \begin{bmatrix} c \delta_2(\gamma) & 0 & 0 \\ 0 & c \|\boldsymbol{\mu}\|^2 \delta_6(\gamma) & c \boldsymbol{\mu}^\top \mathbf{w}_Q \delta_6(\gamma) \\ 0 & c \boldsymbol{\mu}^\top \mathbf{w}_K \delta_6(\gamma) & c \mathbf{w}_K^\top \mathbf{w}_Q \delta_6(\gamma) \end{bmatrix} \in \mathbb{R}^{3 \times 3}$$

$$[\Delta(\gamma)]_{2,2} = \begin{bmatrix} \delta_3(\gamma) & 0 & 0 \\ 0 & \frac{1}{c} \|\boldsymbol{\mu}\|^2 (1 - \frac{\gamma}{c} m(\gamma)) & \frac{1}{c} \boldsymbol{\mu}^\top \mathbf{w}_K (1 - \frac{\gamma}{c} m(\gamma)) \\ 0 & (*) & \frac{1}{c} \|\mathbf{w}_K\|^2 (1 - \frac{\gamma}{c} m(\gamma)) \end{bmatrix} \in \mathbb{R}^{3 \times 3}$$

$$[\Delta(\gamma)]_{2,3} = \begin{bmatrix} \delta_1(\gamma) & 0 & 0 \\ 0 & \|\boldsymbol{\mu}\|^2 \delta_2(\gamma) & \boldsymbol{\mu}^\top \mathbf{w}_Q \delta_2(\gamma) \\ 0 & \boldsymbol{\mu}^\top \mathbf{w}_K \delta_2(\gamma) & \mathbf{w}_K^\top \mathbf{w}_Q \delta_2(\gamma) \end{bmatrix} \in \mathbb{R}^{3 \times 3}$$

$$[\Delta(\gamma)]_{3,3} = \begin{bmatrix} \frac{1}{c} m(\gamma) & 0 & 0 \\ 0 & \|\boldsymbol{\mu}\|^2 \delta_5(\gamma) & \boldsymbol{\mu}^\top \mathbf{w}_Q \delta_5(\gamma) \\ 0 & (*) & \|\mathbf{w}_Q\|^2 \delta_5(\gamma) \end{bmatrix} \in \mathbb{R}^{3 \times 3},$$

*for $\delta_1(\gamma), \delta_2(\gamma), \delta_3(\gamma), \delta_4(\gamma)$ as defined in (27) of the proof of Proposition 1, and*

$$\begin{cases} c \delta_5(\gamma) &= m(\gamma) \left( 1 - \mathbf{v}_2^\top \mathbf{T}(\gamma) \mathbf{v} \right) \\ c \delta_6(\gamma) &= \mathbf{v}_4^\top \mathbf{T}(\gamma) \mathbf{v}_2 + m(\gamma)(\mathbf{v}_2^\top \mathbf{T}(\gamma) \mathbf{v} - 1) \left( \mathbf{v}_4^\top \mathbf{T}(\gamma) \mathbf{v} - \frac{a_1}{c} \right) \\ c \delta_7(\gamma) &= \mathbf{v}_4^\top \mathbf{T}(\gamma) \mathbf{v}_7 + m(\gamma) \left( \mathbf{v}_4^\top \mathbf{T}(\gamma) \mathbf{v} - \frac{a_1}{c} \right) \left( \mathbf{v}_7^\top \mathbf{T}(\gamma) \mathbf{v} - \frac{a_1}{c} \left( 2 + \frac{1}{c} \right) \right) \end{cases} \tag{71}$$

*with*

$$\mathbf{v}_7^\top = \begin{bmatrix} 2\frac{a_1}{c} + \frac{a_1}{c^2} & \frac{1}{c} + 1 & \frac{1}{c} + 1 & 0 & 1 & 0 \end{bmatrix} \in \mathbb{R}^6. \tag{72}$$

*Proof of Lemma 8.* We first work on the approximation of $\Sigma$ defined in (65), for which we exploit the following concentration results:

$$\mathbf{U}_K^\top \mathbf{U}_K = \frac{1}{c} \begin{bmatrix} 1 & 0 & 0 \\ 0 & \|\boldsymbol{\mu}\|^2 & \boldsymbol{\mu}^\top \mathbf{w}_K \\ 0 & \boldsymbol{\mu}^\top \mathbf{w}_K & \|\mathbf{w}_K\|^2 \end{bmatrix} + O_{\|\cdot\|}(n^{-\frac{1}{2}}),$$

$$\frac{1}{n} \mathbf{U}_K^\top \mathbf{Z}^\top \mathbf{Z} \mathbf{U}_K = \begin{bmatrix} 1 & 0 & 0 \\ 0 & \frac{1+c}{c}\|\boldsymbol{\mu}\|^2 & \frac{1+c}{c}\boldsymbol{\mu}^\top \mathbf{w}_K \\ 0 & \frac{1+c}{c}\boldsymbol{\mu}^\top \mathbf{w}_K & \frac{1+c}{c}\|\mathbf{w}_K\|^2 \end{bmatrix} + O_{\|\cdot\|}(n^{-\frac{1}{2}}),$$

where we used the Gaussian moments, we thus get

$$\mathbf{U}_K^\top \frac{1}{n} \mathbf{Z}^\top \mathbf{Z} \mathbf{U}_K + \mathbf{U}_K^\top \mathbf{U}_K \Sigma_{\mathbf{X}} \mathbf{U}_K^\top \mathbf{U}_K = \begin{bmatrix} 1 + \|\boldsymbol{\mu}\|^2/c & \|\boldsymbol{\mu}\|^2/c & \boldsymbol{\mu}^\top \mathbf{w}_K/c \\ \|\boldsymbol{\mu}\|^2/c & \frac{1+c}{c}\|\boldsymbol{\mu}\|^2 & \frac{1+c}{c}\boldsymbol{\mu}^\top \mathbf{w}_K \\ \boldsymbol{\mu}^\top \mathbf{w}_K/c & \frac{1+c}{c}\boldsymbol{\mu}^\top \mathbf{w}_K & \frac{1+c}{c}\|\mathbf{w}_K\|^2 \end{bmatrix} + O_{\|\cdot\|}(n^{-\frac{1}{2}}), \quad (73)$$

and therefore $\Sigma = \Lambda + O_{\|\cdot\|}(n^{-\frac{1}{2}})$ with

$$\Sigma = \begin{bmatrix} \mathbf{0}_3 & \mathbf{0}_3 & \Sigma_{\mathbf{K}} \\ \mathbf{0}_3 & \Sigma_{\mathbf{X}} & [\Lambda]_{2,3} \\ \Sigma_{\mathbf{K}}^\top & [\Lambda]_{2,3}^\top & [\Lambda]_{3,3} \end{bmatrix}, \quad (74)$$

and

$$[\Lambda]_{2,3} = a_1 \begin{bmatrix} (\|\boldsymbol{\mu}\|^2 + 1)T_1 & \|\boldsymbol{\mu}\|^2 & \boldsymbol{\mu}^\top \mathbf{w}_K(\|\boldsymbol{\mu}\|^2 + 1) \\ T_1 & 1 & \boldsymbol{\mu}^\top \mathbf{w}_K \\ 0 & 0 & 0 \end{bmatrix}$$

$$[\Lambda]_{3,3} = a_1^2 \begin{bmatrix} \frac{2+c+\|\boldsymbol{\mu}\|^2}{c} T_1^2 + \frac{1+c}{c} T_1 + \frac{1+c}{c}\boldsymbol{\mu}^\top \mathbf{w}_Q \left( \boldsymbol{\mu}^\top \mathbf{w}_K + \boldsymbol{\mu}^\top \mathbf{w}_Q \|\mathbf{w}_K\|^2 \right) & (*) & (*) \\ \frac{1+c+\|\boldsymbol{\mu}\|^2}{c} T_1 & 1 + \frac{\|\boldsymbol{\mu}\|^2}{c} & \frac{1+c+\|\boldsymbol{\mu}\|^2}{c}\boldsymbol{\mu}^\top \mathbf{w}_K \\ \frac{2+c+\|\boldsymbol{\mu}\|^2}{c}\boldsymbol{\mu}^\top \mathbf{w}_K T_1 + \frac{1+c}{c}(\boldsymbol{\mu}^\top \mathbf{w}_K + \boldsymbol{\mu}^\top \mathbf{w}_Q \|\mathbf{w}_K\|^2) & (*) & \frac{2+c+\|\boldsymbol{\mu}\|^2}{c}(\boldsymbol{\mu}^\top \mathbf{w}_K)^2 + \frac{1+c}{c}\|\mathbf{w}_K\|^2 \end{bmatrix}.$$

where we denote the shortcut $T_1 = \|\boldsymbol{\mu}\|^2 + \boldsymbol{\mu}^\top \mathbf{w}_K \boldsymbol{\mu}^\top \mathbf{w}_Q$, This concludes the proof of the approximation of $\Sigma$ in Lemma 8.

We then proceed to the approximation of $\mathbf{U}^\top \left( \frac{1}{n} \mathbf{K}_N \mathbf{Z}^\top \mathbf{Z} \mathbf{K}_N + \gamma \mathbf{I}_n \right)^{-1} \mathbf{U}$. Note that for $\mathbf{U}$ defined in (64) and

$$\mathbf{Q}_0 \equiv \left( \frac{1}{p} \mathbf{K}_N \mathbf{Z}^\top \mathbf{Z} \mathbf{K}_N + \frac{\gamma}{c} \mathbf{I}_n \right)^{-1}, \quad (75)$$

we have

$$\mathbf{U}^\top \mathbf{Q}_0 \mathbf{U} = \begin{bmatrix} \mathbf{U}_K^\top \frac{1}{n} \mathbf{Z}^\top \mathbf{Z} \mathbf{K}_N \mathbf{Q}_0 \frac{1}{n} \mathbf{K}_N \mathbf{Z}^\top \mathbf{Z} \mathbf{U}_K & \mathbf{U}_K^\top \frac{1}{n} \mathbf{Z}^\top \mathbf{Z} \mathbf{K}_N \mathbf{Q}_0 \mathbf{K}_N \mathbf{U}_K & \mathbf{U}_K^\top \frac{1}{n} \mathbf{Z}^\top \mathbf{Z} \mathbf{K}_N \mathbf{Q}_0 \mathbf{V}_Q \\ \mathbf{U}_K^\top \mathbf{K}_N \mathbf{Q}_0 \frac{1}{n} \mathbf{K}_N \mathbf{Z}^\top \mathbf{Z} \mathbf{U}_K & \mathbf{U}_K^\top \mathbf{K}_N \mathbf{Q}_0 \mathbf{K}_N \mathbf{U}_K & \mathbf{U}_K^\top \mathbf{K}_N \mathbf{Q}_0 \mathbf{V}_Q \\ \mathbf{V}_Q^\top \mathbf{Q}_0 \frac{1}{n} \mathbf{K}_N \mathbf{Z}^\top \mathbf{Z} \mathbf{U}_K & \mathbf{V}_Q^\top \mathbf{Q}_0 \mathbf{K}_N \mathbf{U}_K & \mathbf{V}_Q^\top \mathbf{Q}_0 \mathbf{V}_Q \end{bmatrix} \in \mathbb{R}^{9 \times 9}, \quad (76)$$

which writes as a three-by-three block matrix, for $\mathbf{U}_K = [\mathbf{y}, \mathbf{Z}^\top \boldsymbol{\mu}, \mathbf{Z}^\top \mathbf{w}_K]/\sqrt{p} \in \mathbb{R}^{n \times 3}$, $\mathbf{V}_Q = [\mathbf{y}, \mathbf{Z}^\top \boldsymbol{\mu}, \mathbf{Z}^\top \mathbf{w}_Q]/\sqrt{p} \in \mathbb{R}^{n \times 3}$ as in Lemma 1.

In the following, we further evaluate the nine (in fact six by symmetry) blocks of $\mathbf{U}^\top \mathbf{Q}_0 \mathbf{U}$, in the limit of $n, p \to \infty$ with $p/n \to c \in (0, \infty)$. To that end, we need the following intermediate results.

**Lemma 9** (Further Deterministic Equivalents). *Under the same settings and notations as in Proposition 1, we have the following Deterministic Equivalent results (in the sense of Definition 4)*

$$\frac{1}{n^2}\mathbf{Z}^\top\mathbf{Z}\mathbf{K}_N\mathbf{Q}_0\mathbf{K}_N\mathbf{Z}^\top\mathbf{Z} \leftrightarrow c^3\delta_4(\gamma)\cdot\mathbf{I}_n,$$

$$\frac{1}{n}\mathbf{Z}^\top\mathbf{Z}\mathbf{K}_N\mathbf{Q}_0\mathbf{K}_N \leftrightarrow (c-\gamma m(\gamma))\cdot\mathbf{I}_n,$$

$$\frac{1}{n}\mathbf{Z}^\top\mathbf{Z}\mathbf{K}_N\mathbf{Q}_0 \leftrightarrow c^2\delta_2(\gamma)\cdot\mathbf{I}_n,$$

$$\mathbf{K}_N\mathbf{Q}_0\mathbf{K}_N \leftrightarrow c\delta_3(\gamma)\cdot\mathbf{I}_n,$$

$$\mathbf{K}_N\mathbf{Q}_0 \leftrightarrow c\delta_1(\gamma)\cdot\mathbf{I}_n,$$

$$\frac{1}{p}\mathbf{Z}\frac{1}{n}\mathbf{Z}^\top\mathbf{Z}\mathbf{K}_N\mathbf{Q}_0\frac{1}{n}\mathbf{K}_N\mathbf{Z}^\top\mathbf{Z}\mathbf{Z}^\top \leftrightarrow c^2\delta_7(\gamma)\cdot\mathbf{I}_p,$$

$$\frac{1}{p}\mathbf{Z}\frac{1}{n}\mathbf{Z}^\top\mathbf{Z}\mathbf{K}_N\mathbf{Q}_0\mathbf{K}_N\mathbf{Z}^\top \leftrightarrow c\delta_4(\gamma)\cdot\mathbf{I}_p,$$

$$\frac{1}{p}\mathbf{Z}\frac{1}{n}\mathbf{Z}^\top\mathbf{Z}\mathbf{K}_N\mathbf{Q}_0\mathbf{Z}^\top \leftrightarrow c\delta_6(\gamma)\cdot\mathbf{I}_p,$$

$$\frac{1}{p}\mathbf{Z}\mathbf{K}_N\mathbf{Q}_0\mathbf{K}_N\mathbf{Z}^\top \leftrightarrow \frac{1}{c}\left(1-\frac{\gamma}{c}m(\gamma)\right)\cdot\mathbf{I}_p,$$

$$\frac{1}{p}\mathbf{Z}\mathbf{K}_N\mathbf{Q}_0\mathbf{Z}^\top \leftrightarrow \delta_2(\gamma)\cdot\mathbf{I}_p,$$

$$\frac{1}{p}\mathbf{Z}\mathbf{Q}_0\mathbf{Z}^\top \leftrightarrow \delta_5(\gamma)\cdot\mathbf{I}_p.$$

*Proof of Lemma 9.* Note that for $\mathbf{Q}_0$ defined in (75), we have, by the proof of Proposition 1 in Appendix C.3., the following Deterministic Equivalent results.

$$\frac{1}{n^2}\mathbf{Z}^\top\mathbf{Z}\mathbf{K}_N\mathbf{Q}_0\mathbf{K}_N\mathbf{Z}^\top\mathbf{Z} \leftrightarrow \frac{1}{n^2}\mathbb{E}[\mathbf{Z}^\top\mathbf{Z}\mathbf{K}_N\mathbf{Q}_0\mathbf{K}_N\mathbf{Z}^\top\mathbf{Z}] \leftrightarrow \frac{p^3}{n^3}\operatorname{tr}\frac{1}{p}\left(\frac{1}{p}\mathbf{Z}^\top\mathbf{Z}\mathbf{K}_N\mathbf{Q}_0\frac{1}{p}\mathbf{Z}^\top\mathbf{Z}\right)\cdot\mathbf{I}_n \leftrightarrow c^3\delta_4(\gamma)\cdot\mathbf{I}_n,$$

$$\frac{1}{n}\mathbf{Z}^\top\mathbf{Z}\mathbf{K}_N\mathbf{Q}_0\mathbf{K}_N \leftrightarrow \frac{1}{n}\mathbb{E}[\mathbf{Z}^\top\mathbf{Z}\mathbf{K}_N\mathbf{Q}_0\mathbf{K}_N] \leftrightarrow \frac{p}{n}\frac{1}{n}\operatorname{tr}\left(\frac{1}{p}\mathbf{Z}^\top\mathbf{Z}\mathbf{K}_N\mathbf{Q}_0\mathbf{K}_N\right)\cdot\mathbf{I}_n \leftrightarrow (c-\gamma m(\gamma))\cdot\mathbf{I}_n,$$

$$\frac{1}{n}\mathbf{Z}^\top\mathbf{Z}\mathbf{K}_N\mathbf{Q}_0 \leftrightarrow \frac{1}{n}\mathbb{E}[\mathbf{Z}^\top\mathbf{Z}\mathbf{K}_N\mathbf{Q}_0] \leftrightarrow \frac{p^2}{n^2}\frac{1}{p}\operatorname{tr}\left(\frac{1}{p}\mathbf{Z}^\top\mathbf{Z}\mathbf{K}_N\mathbf{Q}_0\right)\cdot\mathbf{I}_n \leftrightarrow c^2\delta_2(\gamma)\cdot\mathbf{I}_n,$$

$$\mathbf{K}_N\mathbf{Q}_0\mathbf{K}_N \leftrightarrow \mathbb{E}[\mathbf{K}_N\mathbf{Q}_0\mathbf{K}_N] \leftrightarrow \frac{p}{n}\frac{1}{p}\operatorname{tr}(\mathbf{K}_N\mathbf{Q}_0\mathbf{K}_N)\cdot\mathbf{I}_n \leftrightarrow c\delta_3(\gamma)\cdot\mathbf{I}_n,$$

$$\mathbf{K}_N\mathbf{Q}_0 \leftrightarrow \mathbb{E}[\mathbf{K}_N\mathbf{Q}_0] \leftrightarrow \frac{p}{n}\frac{1}{p}\operatorname{tr}(\mathbf{K}_N\mathbf{Q}_0)\cdot\mathbf{I}_n \leftrightarrow c\delta_1(\gamma)\cdot\mathbf{I}_n,$$

$$\mathbf{Q}_0 \leftrightarrow m(\gamma)\cdot\mathbf{I}_n.$$

Similarly, we have

$$\frac{1}{p}\mathbf{Z}\frac{1}{n}\mathbf{Z}^\top\mathbf{Z}\mathbf{K}_N\mathbf{Q}_0\frac{1}{n}\mathbf{K}_N\mathbf{Z}^\top\mathbf{Z}\mathbf{Z}^\top \leftrightarrow \frac{p^2}{n^2}\frac{1}{p}\operatorname{tr}\left(\frac{1}{p}\mathbf{Z}^\top\mathbf{Z}\mathbf{K}_N\mathbf{Q}_0\frac{1}{p}\mathbf{K}_N\mathbf{Z}^\top\mathbf{Z}\frac{1}{p}\mathbf{Z}^\top\mathbf{Z}\right)\cdot\mathbf{I}_p \leftrightarrow c^2\delta_7(\gamma)\cdot\mathbf{I}_p,$$

$$\frac{1}{p}\mathbf{Z}\frac{1}{n}\mathbf{Z}^\top\mathbf{Z}\mathbf{K}_N\mathbf{Q}_0\mathbf{K}_N\mathbf{Z}^\top \leftrightarrow \frac{p}{n}\frac{1}{p}\operatorname{tr}\left(\frac{1}{p}\mathbf{Z}^\top\mathbf{Z}\mathbf{K}_N\mathbf{Q}_0\mathbf{K}_N\frac{1}{p}\mathbf{Z}^\top\mathbf{Z}\right)\cdot\mathbf{I}_p \leftrightarrow c\delta_4(\gamma)\cdot\mathbf{I}_p,$$

$$\frac{1}{p}\mathbf{Z}\frac{1}{n}\mathbf{Z}^\top\mathbf{Z}\mathbf{K}_N\mathbf{Q}_0\mathbf{Z}^\top \leftrightarrow \frac{p}{n}\frac{1}{p}\operatorname{tr}\left(\frac{1}{p}\mathbf{Z}^\top\mathbf{Z}\mathbf{K}_N\mathbf{Q}_0\frac{1}{p}\mathbf{Z}^\top\mathbf{Z}\right)\cdot\mathbf{I}_p \leftrightarrow c\delta_6(\gamma)\cdot\mathbf{I}_p,$$

$$\frac{1}{p}\mathbf{Z}\mathbf{K}_N\mathbf{Q}_0\mathbf{K}_N\mathbf{Z}^\top \leftrightarrow \frac{1}{p}\operatorname{tr}\left(\mathbf{K}_N\mathbf{Q}_0\mathbf{K}_N\frac{1}{p}\mathbf{Z}^\top\mathbf{Z}\right)\cdot\mathbf{I}_p \leftrightarrow \frac{1}{c}\left(1-\frac{\gamma}{c}m(\gamma)\right)\cdot\mathbf{I}_p,$$

$$\frac{1}{p}\mathbf{Z}\mathbf{K}_N\mathbf{Q}_0\mathbf{Z}^\top \leftrightarrow \frac{1}{p}\operatorname{tr}\left(\mathbf{K}_N\mathbf{Q}_0\frac{1}{p}\mathbf{Z}^\top\mathbf{Z}\right)\cdot\mathbf{I}_p \leftrightarrow \delta_2(\gamma)\cdot\mathbf{I}_p,$$

$$\frac{1}{p}\mathbf{Z}\mathbf{Q}_0\mathbf{Z}^\top \leftrightarrow \frac{1}{p}\operatorname{tr}\left(\mathbf{Q}_0\frac{1}{p}\mathbf{Z}^\top\mathbf{Z}\right)\cdot\mathbf{I}_p \leftrightarrow \delta_5(\gamma)\cdot\mathbf{I}_p,$$

for $\delta_5(\gamma), \delta_6(\gamma), \delta_7(\gamma)$ as defined in (71).

To complete the proof of Lemma 9, we establish, in the following as similar to Appendix C.3.3, self-consistent equations for $\delta_5(\gamma), \delta_6(\gamma)$ and $\delta_7(\gamma)$.

**Self-consistent equation for $\delta_5(\gamma)$.** Consider $\delta_5(\gamma) = \frac{1}{p} \mathrm{tr}(\mathbf{Q}\check{\mathbf{Z}}) + o(1)$ and write

$$\delta_5(\gamma) = \frac{1}{p} \mathrm{tr}(\mathbf{Q}\check{\mathbf{Z}}) + o(1) = \frac{1}{c}[\mathbf{Q}\check{\mathbf{Z}}]_{ii} + o(1)$$

$$= \frac{1}{c}\left[-m(\gamma)\mathbf{v}^\top (\mathbf{I}_6 + \Delta_0(\gamma)\Lambda_0)^{-1}\mathbf{U}^\top \mathbf{Q}_{-i}^\perp \quad [\mathbf{Q}]_{ii}\right]\begin{bmatrix}\boldsymbol{\alpha}_{-i}/\sqrt{p} \\ 1\end{bmatrix} + o(1)$$

$$= \frac{1}{c}\left(-m(\gamma)\mathbf{v}^\top (\mathbf{I}_6 + \Delta_0(\gamma)\Lambda_0)^{-1}\mathbf{U}^\top \mathbf{Q}_{-i}^\perp \mathbf{U}\mathbf{v}_2 + m(\gamma)\right) + o(1)$$

$$= \frac{m(\gamma)}{c}\left(-\mathbf{v}^\top (\mathbf{I}_6 + \Delta_0(\gamma)\Lambda_0)^{-1}\Delta_0(\gamma)\mathbf{v}_2 + 1\right) + o(1)$$

$$= \frac{m(\gamma)}{c}\left(-\mathbf{v}_2^\top \Delta_0(\gamma)(\mathbf{I}_6 + \Lambda_0\Delta_0(\gamma))^{-1}\mathbf{v} + 1\right) + o(1)$$

$$= \frac{m(\gamma)}{c}\left(1 - \mathbf{v}_2^\top \mathbf{T}(\gamma)\mathbf{v}\right) + o(1).$$

**Self-consistent equation for $\delta_6(\gamma)$.** Consider now $\delta_6(\gamma) = \frac{1}{p} \mathrm{tr}(\check{\mathbf{Z}}\mathbf{K}\mathbf{Q}\check{\mathbf{Z}}) + o(1)$ and write

$$\delta_6(\gamma) = \frac{1}{p} \mathrm{tr}(\check{\mathbf{Z}}\mathbf{K}\mathbf{Q}\check{\mathbf{Z}}) + o(1) = \frac{1}{c}[\check{\mathbf{Z}}\mathbf{K}\mathbf{Q}\check{\mathbf{Z}}]_{ii} + o(1)$$

$$= \frac{1}{c}\left[\mathbf{v}_4^\top \mathbf{U}^\top \quad \frac{a_1}{c}\right]\begin{bmatrix}\mathbf{Q}_{-i}^\perp - \mathbf{Q}_{-i}^\perp \mathbf{U}(\mathbf{I}_6 + \Lambda_1(\gamma)\Delta_0(\gamma))^{-1}\Lambda_1(\gamma)\mathbf{U}^\top \mathbf{Q}_{-i}^\perp & -m(\gamma)\mathbf{Q}_{-i}^\perp \mathbf{U}(\mathbf{I}_6 + \Lambda_0\Delta_0(\gamma))^{-1}\mathbf{v} \\ -m(\gamma)\mathbf{v}^\top (\mathbf{I}_6 + \Delta_0(\gamma)\Lambda_0)^{-1}\mathbf{U}^\top \mathbf{Q}_{-i}^\perp & [\mathbf{Q}]_{ii}\end{bmatrix}\begin{bmatrix}\mathbf{U}\mathbf{v}_2 \\ 1\end{bmatrix} + o(1)$$

$$= \frac{1}{c}\left(\mathbf{v}_4^\top \Delta_0(\gamma)(\mathbf{I}_6 + \Lambda_1(\gamma)\Delta_0(\gamma))^{-1}\mathbf{v}_2 - m(\gamma)\mathbf{v}_4^\top \Delta_0(\gamma)(\mathbf{I}_6 + \Lambda_0\Delta_0(\gamma))^{-1}\mathbf{v}\right.$$

$$\left.- \frac{a_1}{c}m(\gamma)\mathbf{v}^\top (\mathbf{I}_6 + \Delta_0(\gamma)\Lambda_0)^{-1}\Delta_0(\gamma)\mathbf{v}_2 + \frac{a_1}{c}m(\gamma)\right) + o(1)$$

$$= \frac{1}{c}\left(\mathbf{v}_4^\top \Delta_0(\gamma)(\mathbf{I}_6 + \Lambda_0\Delta_0(\gamma))^{-1}\mathbf{v}_2 + m(\gamma)\mathbf{v}_4^\top \Delta_0(\gamma)(\mathbf{I}_6 + \Lambda_0\Delta_0(\gamma))^{-1}\mathbf{v} \times \mathbf{v}^\top \Delta_0(\gamma)(\mathbf{I}_6 + \Lambda_0\Delta_0(\gamma))^{-1}\mathbf{v}_2\right.$$

$$\left.-m(\gamma)\left(\frac{a_1}{c}\mathbf{v}_2 + \mathbf{v}_4\right)^\top \Delta_0(\gamma)(\mathbf{I}_6 + \Lambda_0\Delta_0(\gamma))^{-1}\mathbf{v} + \frac{a_1}{c}m(\gamma)\right) + o(1)$$

$$= \frac{1}{c}\left(\mathbf{v}_4^\top \mathbf{T}(\gamma)\mathbf{v}_2 + m(\gamma)(\mathbf{v}_2^\top \mathbf{T}(\gamma)\mathbf{v} - 1)\left(\mathbf{v}_4^\top \mathbf{T}(\gamma)\mathbf{v} - \frac{a_1}{c}\right)\right) + o(1).$$

**Self-consistent equation for $\delta_7(\gamma)$.** Consider $\delta_7(\gamma) = \frac{1}{p} \mathrm{tr}(\check{\mathbf{Z}}\mathbf{K}\mathbf{Q}\mathbf{K}\check{\mathbf{Z}}\check{\mathbf{Z}}) + o(1)$ and write

$$\delta_7(\gamma) = \frac{1}{p} \mathrm{tr}(\check{\mathbf{Z}}\mathbf{K}\mathbf{Q}\mathbf{K}\check{\mathbf{Z}}\check{\mathbf{Z}}) + o(1) = \frac{1}{c}[\check{\mathbf{Z}}\mathbf{K}\mathbf{Q}\mathbf{K}\check{\mathbf{Z}}\check{\mathbf{Z}}]_{ii} + o(1)$$

$$= \frac{1}{c}\left[\mathbf{v}_4^\top \mathbf{U}^\top \quad \frac{a_1}{c}\right]\begin{bmatrix}\mathbf{Q}_{-i}^\perp - \mathbf{Q}_{-i}^\perp \mathbf{U}(\mathbf{I}_6 + \Lambda_1(\gamma)\Delta(\gamma))^{-1}\Lambda_1(\gamma)\mathbf{U}^\top \mathbf{Q}_{-i}^\perp & -m(\gamma)\mathbf{Q}_{-i}^\perp \mathbf{U}(\mathbf{I}_6 + \Lambda_0\Delta(\gamma))^{-1}\mathbf{v} \\ -m(\gamma)\mathbf{v}^\top (\mathbf{I}_6 + \Delta(\gamma)\Lambda_0)^{-1}\mathbf{U}^\top \mathbf{Q}_{-i}^\perp & [\mathbf{Q}]_{ii}\end{bmatrix}\begin{bmatrix}\mathbf{U}\mathbf{v}_7 \\ 2\frac{a_1}{c} + \frac{a_1}{c^2}\end{bmatrix} + o(1)$$

$$= \frac{1}{c}\left(\mathbf{v}_4^\top \Delta(\gamma)(\mathbf{I}_6 + \Lambda_1(\gamma)\Delta(\gamma))^{-1}\mathbf{v}_7 - \left(2\frac{a_1}{c} + \frac{a_1}{c^2}\right)m(\gamma)\mathbf{v}_4^\top \Delta(\gamma)(\mathbf{I}_6 + \Lambda_0\Delta(\gamma))^{-1}\mathbf{v}\right.$$

$$\left.- \frac{a_1}{c}m(\gamma)\left(\mathbf{v}^\top (\mathbf{I}_6 + \Delta(\gamma)\Lambda_0)^{-1}\Delta(\gamma)\mathbf{v}_7 - \left(2\frac{a_1}{c} + \frac{a_1}{c^2}\right)\right)\right) + o(1)$$

$$= \frac{1}{c}\left(\mathbf{v}_4^\top \Delta(\mathbf{I}_6 + \Lambda_0\Delta(\gamma))^{-1}\mathbf{v}_7 + m(\gamma)\mathbf{v}_4^\top \Delta(\mathbf{I}_6 + \Lambda_0\Delta(\gamma))^{-1}\mathbf{v}\mathbf{v}^\top \Delta(\mathbf{I}_6 + \Lambda_0\Delta(\gamma))^{-1}\mathbf{v}_7\right.$$

$$\left.- \frac{a_1}{c}m(\gamma)\left(\left(\frac{1}{c} + 2\right)\mathbf{v}_4 + \mathbf{v}_7\right)^\top \Delta(\gamma)(\mathbf{I}_6 + \Lambda_0\Delta(\gamma))^{-1}\mathbf{v} + \frac{a_1^2}{c^2}\left(\frac{1}{c} + 2\right)m(\gamma)\right) + o(1).$$

With these self-consistent equations for $\delta_5(\gamma), \delta_6(\gamma), \delta_7(\gamma)$, we conclude the proof of Lemma 9. $\square$

With Lemma 9 at hand, we are now ready to evaluate the blocks of $\mathbf{U}^\top \mathbf{Q}_0 \mathbf{U}$ as follows.

**Approximation of the $(1, 1)$ block of $\mathbf{U}^\top \mathbf{Q}_0 \mathbf{U}$.**

$$
\mathbf{U}_K^\top \frac{1}{n} \mathbf{Z}^\top \mathbf{Z} \mathbf{K}_N \mathbf{Q}_0(\gamma) \frac{1}{n} \mathbf{K}_N \mathbf{Z}^\top \mathbf{Z} \mathbf{U}_K = \frac{1}{p} \begin{bmatrix} \mathbf{y}^\top \\ \boldsymbol{\mu}^\top \mathbf{Z} \\ \mathbf{w}_K^\top \mathbf{Z} \end{bmatrix} \frac{1}{n} \mathbf{Z}^\top \mathbf{Z} \mathbf{K}_N \mathbf{Q}_0 \frac{1}{n} \mathbf{K}_N \mathbf{Z}^\top \mathbf{Z} \begin{bmatrix} \mathbf{y} & \mathbf{Z}^\top \boldsymbol{\mu} & \mathbf{Z}^\top \mathbf{w}_K \end{bmatrix}
$$

$$
= \frac{1}{p} \begin{bmatrix} \mathbf{y}^\top \frac{1}{n} \mathbf{Z}^\top \mathbf{Z} \mathbf{K}_N \mathbf{Q}_0 \frac{1}{n} \mathbf{K}_N \mathbf{Z}^\top \mathbf{Z} \mathbf{y} & \mathbf{y}^\top \frac{1}{n} \mathbf{Z}^\top \mathbf{Z} \mathbf{K}_N \mathbf{Q}_0 \frac{1}{n} \mathbf{K}_N \mathbf{Z}^\top \mathbf{Z} \mathbf{Z}^\top \boldsymbol{\mu} & \mathbf{y}^\top \frac{1}{n} \mathbf{Z}^\top \mathbf{Z} \mathbf{K}_N \mathbf{Q}_0 \frac{1}{n} \mathbf{K}_N \mathbf{Z}^\top \mathbf{Z} \mathbf{Z}^\top \mathbf{w}_K \\ \boldsymbol{\mu}^\top \mathbf{Z} \frac{1}{n} \mathbf{Z}^\top \mathbf{Z} \mathbf{K}_N \mathbf{Q}_0 \frac{1}{n} \mathbf{K}_N \mathbf{Z}^\top \mathbf{Z} \mathbf{y} & \boldsymbol{\mu}^\top \mathbf{Z} \frac{1}{n} \mathbf{Z}^\top \mathbf{Z} \mathbf{K}_N \mathbf{Q}_0 \frac{1}{n} \mathbf{K}_N \mathbf{Z}^\top \mathbf{Z} \mathbf{Z}^\top \boldsymbol{\mu} & \boldsymbol{\mu}^\top \mathbf{Z} \frac{1}{n} \mathbf{Z}^\top \mathbf{Z} \mathbf{K}_N \mathbf{Q}_0 \frac{1}{n} \mathbf{K}_N \mathbf{Z}^\top \mathbf{Z} \mathbf{Z}^\top \mathbf{w}_K \\ \mathbf{w}_K^\top \mathbf{Z} \frac{1}{n} \mathbf{Z}^\top \mathbf{Z} \mathbf{K}_N \mathbf{Q}_0 \frac{1}{n} \mathbf{K}_N \mathbf{Z}^\top \mathbf{Z} \mathbf{y} & \mathbf{w}_K^\top \mathbf{Z} \frac{1}{n} \mathbf{Z}^\top \mathbf{Z} \mathbf{K}_N \mathbf{Q}_0 \frac{1}{n} \mathbf{K}_N \mathbf{Z}^\top \mathbf{Z} \mathbf{Z}^\top \boldsymbol{\mu} & \mathbf{w}_K^\top \mathbf{Z} \frac{1}{n} \mathbf{Z}^\top \mathbf{Z} \mathbf{K}_N \mathbf{Q}_0 \frac{1}{n} \mathbf{K}_N \mathbf{Z}^\top \mathbf{Z} \mathbf{Z}^\top \mathbf{w}_K \end{bmatrix}
$$

$$
= c^2 \begin{bmatrix} \delta_4(\gamma) & 0 & 0 \\ 0 & \|\boldsymbol{\mu}\|^2 \delta_7(\gamma) & \boldsymbol{\mu}^\top \mathbf{w}_K \delta_7(\gamma) \\ 0 & \boldsymbol{\mu}^\top \mathbf{w}_K \delta_7(\gamma) & \|\mathbf{w}_K\|^2 \delta_7(\gamma) \end{bmatrix} + O_{\|\cdot\|}(n^{-1/2}),
$$

where we used Lemma 9 for the approximation in the last line.

**Approximation of the $(1, 2)$ block of $\mathbf{U}^\top \mathbf{Q}_0 \mathbf{U}$.**

$$
\mathbf{U}_K^\top \frac{1}{n} \mathbf{Z}^\top \mathbf{Z} \mathbf{K}_N \mathbf{Q}_0(\gamma) \mathbf{K}_N \mathbf{U}_K = \frac{1}{p} \begin{bmatrix} \mathbf{y}^\top \\ \boldsymbol{\mu}^\top \mathbf{Z} \\ \mathbf{w}_K^\top \mathbf{Z} \end{bmatrix} \frac{1}{n} \mathbf{Z}^\top \mathbf{Z} \mathbf{K}_N \mathbf{Q}_0 \mathbf{K}_N \begin{bmatrix} \mathbf{y} & \mathbf{Z}^\top \boldsymbol{\mu} & \mathbf{Z}^\top \mathbf{w}_K \end{bmatrix}
$$

$$
= \frac{1}{p} \begin{bmatrix} \mathbf{y}^\top \frac{1}{n} \mathbf{Z}^\top \mathbf{Z} \mathbf{K}_N \mathbf{Q}_0 \mathbf{K}_N \mathbf{y} & \mathbf{y}^\top \frac{1}{n} \mathbf{Z}^\top \mathbf{Z} \mathbf{K}_N \mathbf{Q}_0 \mathbf{K}_N \mathbf{Z}^\top \boldsymbol{\mu} & \mathbf{y}^\top \frac{1}{n} \mathbf{Z}^\top \mathbf{Z} \mathbf{K}_N \mathbf{Q}_0 \mathbf{K}_N \mathbf{Z}^\top \mathbf{w}_K \\ \boldsymbol{\mu}^\top \mathbf{Z} \frac{1}{n} \mathbf{Z}^\top \mathbf{Z} \mathbf{K}_N \mathbf{Q}_0 \mathbf{K}_N \mathbf{y} & \boldsymbol{\mu}^\top \mathbf{Z} \frac{1}{n} \mathbf{Z}^\top \mathbf{Z} \mathbf{K}_N \mathbf{Q}_0 \mathbf{K}_N \mathbf{Z}^\top \boldsymbol{\mu} & \boldsymbol{\mu}^\top \mathbf{Z} \frac{1}{n} \mathbf{Z}^\top \mathbf{Z} \mathbf{K}_N \mathbf{Q}_0 \mathbf{K}_N \mathbf{Z}^\top \mathbf{w}_K \\ \mathbf{w}_K^\top \mathbf{Z} \frac{1}{n} \mathbf{Z}^\top \mathbf{Z} \mathbf{K}_N \mathbf{Q}_0 \mathbf{K}_N \mathbf{y} & \mathbf{w}_K^\top \mathbf{Z} \frac{1}{n} \mathbf{Z}^\top \mathbf{Z} \mathbf{K}_N \mathbf{Q}_0 \mathbf{K}_N \mathbf{Z}^\top \boldsymbol{\mu} & \mathbf{w}_K^\top \mathbf{Z} \frac{1}{n} \mathbf{Z}^\top \mathbf{Z} \mathbf{K}_N \mathbf{Q}_0 \mathbf{K}_N \mathbf{Z}^\top \mathbf{w}_K \end{bmatrix}
$$

$$
= \begin{bmatrix} 1 - \frac{\gamma}{c} m(\gamma) & 0 & 0 \\ 0 & c\|\boldsymbol{\mu}\|^2 \delta_4(\gamma) & c\boldsymbol{\mu}^\top \mathbf{w}_K \delta_4(\gamma) \\ 0 & c\boldsymbol{\mu}^\top \mathbf{w}_K \delta_4(\gamma) & c\|\mathbf{w}_K\|^2 \delta_4(\gamma) \end{bmatrix} + O_{\|\cdot\|}(n^{-1/2}),
$$

where we used Lemma 9 for the approximation in the last line.

**Approximation of the $(1, 3)$ block of $\mathbf{U}^\top \mathbf{Q}_0 \mathbf{U}$.**

$$
\mathbf{U}_K^\top \frac{1}{n} \mathbf{Z}^\top \mathbf{Z} \mathbf{K}_N \mathbf{Q}_0(\gamma) \mathbf{V}_Q = \frac{1}{p} \begin{bmatrix} \mathbf{y}^\top \\ \boldsymbol{\mu}^\top \mathbf{Z} \\ \mathbf{w}_K^\top \mathbf{Z} \end{bmatrix} \frac{1}{n} \mathbf{Z}^\top \mathbf{Z} \mathbf{K}_N \mathbf{Q}_0 \begin{bmatrix} \mathbf{y} & \mathbf{Z}^\top \boldsymbol{\mu} & \mathbf{Z}^\top \mathbf{w}_Q \end{bmatrix}
$$

$$
= \frac{1}{p} \begin{bmatrix} \mathbf{y}^\top \frac{1}{n} \mathbf{Z}^\top \mathbf{Z} \mathbf{K}_N \mathbf{Q}_0 \mathbf{y} & \mathbf{y}^\top \frac{1}{n} \mathbf{Z}^\top \mathbf{Z} \mathbf{K}_N \mathbf{Q}_0 \mathbf{Z}^\top \boldsymbol{\mu} & \mathbf{y}^\top \frac{1}{n} \mathbf{Z}^\top \mathbf{Z} \mathbf{K}_N \mathbf{Q}_0 \mathbf{Z}^\top \mathbf{w}_Q \\ \boldsymbol{\mu}^\top \mathbf{Z} \frac{1}{n} \mathbf{Z}^\top \mathbf{Z} \mathbf{K}_N \mathbf{Q}_0 \mathbf{y} & \boldsymbol{\mu}^\top \mathbf{Z} \frac{1}{n} \mathbf{Z}^\top \mathbf{Z} \mathbf{K}_N \mathbf{Q}_0 \mathbf{Z}^\top \boldsymbol{\mu} & \boldsymbol{\mu}^\top \mathbf{Z} \frac{1}{n} \mathbf{Z}^\top \mathbf{Z} \mathbf{K}_N \mathbf{Q}_0 \mathbf{Z}^\top \mathbf{w}_Q \\ \mathbf{w}_K^\top \mathbf{Z} \frac{1}{n} \mathbf{Z}^\top \mathbf{Z} \mathbf{K}_N \mathbf{Q}_0 \mathbf{y} & \mathbf{w}_K^\top \mathbf{Z} \frac{1}{n} \mathbf{Z}^\top \mathbf{Z} \mathbf{K}_N \mathbf{Q}_0 \mathbf{Z}^\top \boldsymbol{\mu} & \mathbf{w}_K^\top \mathbf{Z} \frac{1}{n} \mathbf{Z}^\top \mathbf{Z} \mathbf{K}_N \mathbf{Q}_0 \mathbf{Z}^\top \mathbf{w}_Q \end{bmatrix}
$$

$$
= c \begin{bmatrix} \delta_2(\gamma) & 0 & 0 \\ 0 & \|\boldsymbol{\mu}\|^2 \delta_6(\gamma) & \boldsymbol{\mu}^\top \mathbf{w}_Q \delta_6(\gamma) \\ 0 & \boldsymbol{\mu}^\top \mathbf{w}_K \delta_6(\gamma) & \mathbf{w}_K^\top \mathbf{w}_Q \delta_6(\gamma) \end{bmatrix} + O_{\|\cdot\|}(n^{-1/2}),
$$

where we used Lemma 9 for the approximation in the last line.

**Approximation of the $(2, 2)$ block of $\mathbf{U}^\top \mathbf{Q}_0 \mathbf{U}$.**

$$
\mathbf{U}_K^\top \mathbf{K}_N \mathbf{Q}_0(\gamma) \mathbf{K}_N \mathbf{U}_K = \frac{1}{p} \begin{bmatrix} \mathbf{y}^\top \\ \boldsymbol{\mu}^\top \mathbf{Z} \\ \mathbf{w}_K^\top \mathbf{Z} \end{bmatrix} \mathbf{K}_N \mathbf{Q}_0 \mathbf{K}_N \begin{bmatrix} \mathbf{y} & \mathbf{Z}^\top \boldsymbol{\mu} & \mathbf{Z}^\top \mathbf{w}_K \end{bmatrix}
$$

$$
= \frac{1}{p} \begin{bmatrix} \mathbf{y}^\top \mathbf{K}_N \mathbf{Q}_0 \mathbf{K}_N \mathbf{y} & \mathbf{y}^\top \mathbf{K}_N \mathbf{Q}_0 \mathbf{K}_N \mathbf{Z}^\top \boldsymbol{\mu} & \mathbf{y}^\top \mathbf{K}_N \mathbf{Q}_0 \mathbf{K}_N \mathbf{Z}^\top \mathbf{w}_K \\ \boldsymbol{\mu}^\top \mathbf{Z} \mathbf{K}_N \mathbf{Q}_0 \mathbf{K}_N \mathbf{y} & \boldsymbol{\mu}^\top \mathbf{Z} \mathbf{K}_N \mathbf{Q}_0 \mathbf{K}_N \mathbf{Z}^\top \boldsymbol{\mu} & \boldsymbol{\mu}^\top \mathbf{Z} \mathbf{K}_N \mathbf{Q}_0 \mathbf{K}_N \mathbf{Z}^\top \mathbf{w}_K \\ \mathbf{w}_K^\top \mathbf{Z} \mathbf{K}_N \mathbf{Q}_0 \mathbf{K}_N \mathbf{y} & \mathbf{w}_K^\top \mathbf{Z} \mathbf{K}_N \mathbf{Q}_0 \mathbf{K}_N \mathbf{Z}^\top \boldsymbol{\mu} & \mathbf{w}_K^\top \mathbf{Z} \mathbf{K}_N \mathbf{Q}_0 \mathbf{K}_N \mathbf{Z}^\top \mathbf{w}_K \end{bmatrix}
$$

$$
= \frac{1}{c} \begin{bmatrix} c\delta_3(\gamma) & 0 & 0 \\ 0 & \|\boldsymbol{\mu}\|^2 (1 - \frac{\gamma}{c} m(\gamma)) & \boldsymbol{\mu}^\top \mathbf{w}_K (1 - \frac{\gamma}{c} m(\gamma)) \\ 0 & \boldsymbol{\mu}^\top \mathbf{w}_K (1 - \frac{\gamma}{c} m(\gamma)) & \|\mathbf{w}_K\|^2 (1 - \frac{\gamma}{c} m(\gamma)) \end{bmatrix} + O_{\|\cdot\|}(n^{-1/2}),
$$

where we used Lemma 9 for the approximation in the last line.

**Approximation of the** $(2, 3)$ **block of** $\mathbf{U}^\top \mathbf{Q}_0 \mathbf{U}$.

$$\mathbf{U}_K^\top \mathbf{K}_N \mathbf{Q}_0(\gamma) \mathbf{V}_Q = \frac{1}{p} \begin{bmatrix} \mathbf{y}^\top \\ \boldsymbol{\mu}^\top \mathbf{Z} \\ \mathbf{w}_K^\top \mathbf{Z} \end{bmatrix} \mathbf{K}_N \mathbf{Q}_0 \begin{bmatrix} \mathbf{y} & \mathbf{Z}^\top \boldsymbol{\mu} & \mathbf{Z}^\top \mathbf{w}_Q \end{bmatrix}$$

$$= \frac{1}{p} \begin{bmatrix} \mathbf{y}^\top \mathbf{K}_N \mathbf{Q}_0 \mathbf{y} & \mathbf{y}^\top \mathbf{K}_N \mathbf{Q}_0 \mathbf{Z}^\top \boldsymbol{\mu} & \mathbf{y}^\top \mathbf{K}_N \mathbf{Q}_0 \mathbf{Z}^\top \mathbf{w}_Q \\ \boldsymbol{\mu}^\top \mathbf{Z} \mathbf{K}_N \mathbf{Q}_0 \mathbf{y} & \boldsymbol{\mu}^\top \mathbf{Z} \mathbf{K}_N \mathbf{Q}_0 \mathbf{Z}^\top \boldsymbol{\mu} & \boldsymbol{\mu}^\top \mathbf{Z} \mathbf{K}_N \mathbf{Q}_0 \mathbf{Z}^\top \mathbf{w}_Q \\ \mathbf{w}_K^\top \mathbf{Z} \mathbf{K}_N \mathbf{Q}_0 \mathbf{y} & \mathbf{w}_K^\top \mathbf{Z} \mathbf{K}_N \mathbf{Q}_0 \mathbf{Z}^\top \boldsymbol{\mu} & \mathbf{w}_K^\top \mathbf{Z} \mathbf{K}_N \mathbf{Q}_0 \mathbf{Z}^\top \mathbf{w}_Q \end{bmatrix}$$

$$= \begin{bmatrix} \delta_1(\gamma) & 0 & 0 \\ 0 & \|\boldsymbol{\mu}\|^2 \delta_2(\gamma) & \boldsymbol{\mu}^\top \mathbf{w}_Q \delta_2(\gamma) \\ 0 & \boldsymbol{\mu}^\top \mathbf{w}_K \delta_2(\gamma) & \mathbf{w}_K^\top \mathbf{w}_Q \delta_2(\gamma) \end{bmatrix} + O_{\|\cdot\|}(n^{-1/2}),$$

where we again use Lemma 9 for the approximation in the last line.

**Approximation of the** $(3, 3)$ **block of** $\mathbf{U}^\top \mathbf{Q}_0 \mathbf{U}$.

$$\mathbf{V}_Q^\top \mathbf{Q}_0(\gamma) \mathbf{V}_Q = \frac{1}{p} \begin{bmatrix} \mathbf{y}^\top \\ \boldsymbol{\mu}^\top \mathbf{Z} \\ \mathbf{w}_Q^\top \mathbf{Z} \end{bmatrix} \mathbf{K}_N \mathbf{Q}_0 \mathbf{K}_N \begin{bmatrix} \mathbf{y} & \mathbf{Z}^\top \boldsymbol{\mu} & \mathbf{Z}^\top \mathbf{w}_Q \end{bmatrix}$$

$$= \frac{1}{p} \begin{bmatrix} \mathbf{y}^\top \mathbf{Q}_0 \mathbf{y} & \mathbf{y}^\top \mathbf{Q}_0 \mathbf{Z}^\top \boldsymbol{\mu} & \mathbf{y}^\top \mathbf{Q}_0 \mathbf{Z}^\top \mathbf{w}_Q \\ \boldsymbol{\mu}^\top \mathbf{Z} \mathbf{Q}_0 \mathbf{y} & \boldsymbol{\mu}^\top \mathbf{Z} \mathbf{Q}_0 \mathbf{Z}^\top \boldsymbol{\mu} & \boldsymbol{\mu}^\top \mathbf{Z} \mathbf{Q}_0 \mathbf{Z}^\top \mathbf{w}_Q \\ \mathbf{w}_Q^\top \mathbf{Z} \mathbf{Q}_0 \mathbf{y} & \mathbf{w}_Q^\top \mathbf{Z} \mathbf{Q}_0 \mathbf{Z}^\top \boldsymbol{\mu} & \mathbf{w}_Q^\top \mathbf{Z} \mathbf{Q}_0 \mathbf{Z}^\top \mathbf{w}_Q \end{bmatrix}$$

$$= \begin{bmatrix} \frac{n}{p} m(\gamma) & 0 & 0 \\ 0 & \|\boldsymbol{\mu}\|^2 \delta_5(\gamma) & \boldsymbol{\mu}^\top \mathbf{w}_Q \delta_5(\gamma) \\ 0 & \boldsymbol{\mu}^\top \mathbf{w}_Q \delta_5(\gamma) & \|\mathbf{w}_Q\|^2 \delta_5(\gamma) \end{bmatrix} + O_{\|\cdot\|}(n^{-1/2}),$$

where we used Lemma 9 for the approximation in the last line. This concludes the proof of the approximation of the quadratic form $\mathbf{U}^\top \left( \frac{1}{n} \mathbf{K}_N \mathbf{Z}^\top \mathbf{Z} \mathbf{K}_N + \gamma \mathbf{I}_n \right)^{-1} \mathbf{U}$ in Lemma 8. $\qquad \square$

With Lemma 8 at hand, it follows from (67) that

$$\frac{1}{n} \mathbf{y}^\top \mathbf{Q}(\gamma) \mathbf{y} = c \cdot \mathbf{e}_7^\top \mathbf{U}^\top \left( \frac{1}{n} \mathbf{K}_N \mathbf{Z}^\top \mathbf{Z} \mathbf{K}_N + \gamma \mathbf{I}_n \right)^{-1} \mathbf{U} \cdot \left( \mathbf{I}_9 + \boldsymbol{\Sigma} \mathbf{U}^\top \left( \frac{1}{n} \mathbf{K}_N \mathbf{Z}^\top \mathbf{Z} \mathbf{K}_N + \gamma \mathbf{I}_n \right)^{-1} \mathbf{U} \right)^{-1} \mathbf{e}_7 + O(n^{-\frac{1}{2}})$$

$$= c \cdot \mathbf{e}_7^\top \boldsymbol{\Delta}(\gamma) \cdot (\mathbf{I}_9 + \boldsymbol{\Lambda} \boldsymbol{\Delta}(\gamma))^{-1} \mathbf{e}_7 + O(n^{-\frac{1}{2}}).$$

To assess the high-dimensional behavior of the memorization error $E$ defined in (7) of Definition 3, it thus remains to evaluate the following derivative (with respective to $\gamma$) as

$$E = -\frac{\gamma^2}{n} \frac{\partial \mathbf{y}^\top \mathbf{Q}(\gamma) \mathbf{y}}{\partial \gamma} = -\gamma^2 c^2 \cdot \mathbf{e}_7^\top (c \mathbf{I}_9 + \boldsymbol{\Delta}(\gamma) \boldsymbol{\Lambda})^{-1} \boldsymbol{\Delta}'(\gamma) (c \mathbf{I}_9 + \boldsymbol{\Lambda} \boldsymbol{\Delta}(\gamma))^{-1} \mathbf{e}_7 + O(n^{-\frac{1}{2}}),$$

where we denote $\boldsymbol{\Delta}'(\gamma)$ the derivative (with respect to $\gamma$) of $\boldsymbol{\Delta}(\gamma)$ defined in (69).

To evaluate $\boldsymbol{\Delta}'(\gamma)$, we need the following result on the derivatives of $m'(\gamma)$ and $\delta_{(\gamma)}$s.

**Lemma 10** (Derivatives of the $\delta(\gamma)$s). *Under the settings and notations of Theorem 1, we have that $m'(\gamma), \delta_1'(\gamma), \delta_2'(\gamma), \delta_3'(\gamma), \delta_4'(\gamma)$ satisfy the following system of equations*

$$\begin{cases} m'(\gamma) & = \left( \mathbf{v}^\top \mathbf{T}'(\gamma) \mathbf{v} - \frac{1}{c} \right) m^2(\gamma) \\ c \delta_1'(\gamma) & = -m'(\gamma) \mathbf{v}^\top \mathbf{T}(\gamma) \mathbf{v}_1 - m(\gamma) \mathbf{v}^\top \mathbf{T}'(\gamma) \mathbf{v}_1 \\ c \delta_2'(\gamma) & = \mathbf{v}_2^\top \mathbf{T}'(\gamma) (\mathbf{v}_1 - c \delta_1(\gamma) \mathbf{v}) + c \delta_1'(\gamma) (1 - \mathbf{v}_2^\top \mathbf{T}(\gamma) \mathbf{v}) \\ c \delta_3'(\gamma) & = \mathbf{v}_1^\top \mathbf{T}'(\gamma) \mathbf{v}_1 + \frac{c^2 \delta_1(\gamma) \left( 2 \delta_1'(\gamma) m(\gamma) - \delta_1(\gamma) m'(\gamma) \right)}{m^2(\gamma)} \\ c \delta_4'(\gamma) & = \mathbf{v}_4^\top \mathbf{T}'(\gamma) \mathbf{v}_4 + m'(\gamma) \left( \mathbf{v}_4^\top \mathbf{T}(\gamma) \mathbf{v} - \frac{a_1}{c} \right)^2 + 2m(\gamma) \left( \mathbf{v}_4^\top \mathbf{T}(\gamma) \mathbf{v} - \frac{a_1}{c} \right) \mathbf{v}_4^\top \mathbf{T}'(\gamma) \mathbf{v} \end{cases} \tag{77}$$

*for $\mathbf{T}(\gamma)$ and $\boldsymbol{\Delta}_0(\gamma)$ defined in (29) and (30), receptively, so that their derivatives (with respective to $\gamma$) satisfy*

$$\mathbf{T}'(\gamma) = (\mathbf{I}_6 + \boldsymbol{\Delta}_0(\gamma) \boldsymbol{\Lambda}_0)^{-1} \boldsymbol{\Delta}_0'(\gamma) (\mathbf{I}_6 + \boldsymbol{\Lambda}_0 \boldsymbol{\Delta}_0(\gamma))^{-1},$$

*and*

$$\Delta_0'(\gamma) \equiv \begin{bmatrix} \frac{m'(\gamma)}{c} & \frac{a_1}{c}m'(\gamma) & \delta_1'(\gamma) & a_1\delta_1'(\gamma) & \delta_2'(\gamma) & a_1\delta_2'(\gamma) \\ \frac{a_1}{c}m'(\gamma) & \frac{v}{c}m'(\gamma) & a_1\delta_1'(\gamma) & v\delta_1'(\gamma) & a_1\delta_2'(\gamma) & v\delta_2'(\gamma) \\ \delta_1'(\gamma) & a_1\delta_1'(\gamma) & \delta_3'(\gamma) & a_1\delta_3'(\gamma) & -\frac{1}{c^2}(m(\gamma)+\gamma m'(\gamma)) & -\frac{a_1}{c^2}(m(\gamma)+\gamma m'(\gamma)) \\ a_1\delta_1'(\gamma) & v\delta_1'(\gamma) & a_1\delta_3'(\gamma) & v\delta_3'(\gamma) & -\frac{a_1}{c^2}(m(\gamma)+\gamma m'(\gamma)) & -\frac{v}{c^2}(m(\gamma)+\gamma m'(\gamma)) \\ \delta_2'(\gamma) & a_1\delta_2'(\gamma) & -\frac{1}{c^2}(m(\gamma)+\gamma m'(\gamma)) & -\frac{a_1}{c^2}(m(\gamma)+\gamma m'(\gamma)) & \delta_4'(\gamma) & a_1\delta_4'(\gamma) \\ a_1\delta_2'(\gamma) & v\delta_2'(\gamma) & -\frac{a_1}{c^2}(m(\gamma)+\gamma m'(\gamma)) & -\frac{v}{c^2}(m(\gamma)+\gamma m'(\gamma)) & a_1\delta_4'(\gamma) & v\delta_4'(\gamma) \end{bmatrix}$$

*Also, the derivatives $\delta_5'(\gamma), \delta_6'(\gamma), \delta_7'(\gamma)$ are given by*

$$c\delta_5'(\gamma) = m'(\gamma)\left(-\mathbf{v}_2^\top\mathbf{T}(\gamma)\mathbf{v}+1\right) - m(\gamma)\mathbf{v}_2^\top\mathbf{T}'(\gamma)\mathbf{v}$$

$$c\delta_6'(\gamma) = \mathbf{v}_4^\top\mathbf{T}'(\gamma)\mathbf{v}_2 + m'(\gamma)\left(\mathbf{v}_2^\top\mathbf{T}(\gamma)\mathbf{v}-1\right)\left(\mathbf{v}_4^\top\mathbf{T}(\gamma)\mathbf{v}-\frac{a_1}{c}\right) + m(\gamma)\mathbf{v}_2^\top\mathbf{T}'(\gamma)\mathbf{v}\left(\mathbf{v}_4^\top\mathbf{T}(\gamma)\mathbf{v}-\frac{a_1}{c}\right)$$

$$+ m(\gamma)\mathbf{v}_4^\top\mathbf{T}'(\gamma)\mathbf{v}\left(\mathbf{v}_2^\top\mathbf{T}(\gamma)\mathbf{v}-1\right)$$

$$c\delta_7'(\gamma) = \mathbf{v}_4^\top\mathbf{T}'(\gamma)\mathbf{v}_7 + m'(\gamma)\left(\mathbf{v}_4^\top\mathbf{T}(\gamma)\mathbf{v}-\frac{a_1}{c}\right)\left(\mathbf{v}_7^\top\mathbf{T}(\gamma)\mathbf{v}-\frac{a_1}{c}\left(1+\frac{1}{c}\right)\right) + m(\gamma)\mathbf{v}_4^\top\mathbf{T}'(\gamma)\mathbf{v}\left(\mathbf{v}_7^\top\mathbf{T}(\gamma)\mathbf{v}-\frac{a_1}{c}\left(1+\frac{1}{c}\right)\right)$$

$$+ m(\gamma)\mathbf{v}_7^\top\mathbf{T}'(\gamma)\mathbf{v}\left(\mathbf{v}_4^\top\mathbf{T}(\gamma)\mathbf{v}-\frac{a_1}{c}\right).$$

*Proof of Lemma 10.* By their definitions in Proposition 1 and Lemma 8, we have

$$m'(\gamma) = -\frac{\frac{1}{c}-\mathbf{v}^\top\mathbf{T}'(\gamma)\mathbf{v}}{\left(\frac{\gamma}{c}+\frac{v}{c}+\frac{a_1^2}{c^2}-\mathbf{v}^\top\mathbf{T}(\gamma)\mathbf{v}\right)^2} = \left(\mathbf{v}^\top\mathbf{T}'(\gamma)\mathbf{v}-\frac{1}{c}\right)m^2(\gamma)$$

$$c\delta_1'(\gamma) = -\left(m'(\gamma)\mathbf{v}^\top\mathbf{T}(\gamma)\mathbf{v}_1 + m(\gamma)\mathbf{v}^\top\mathbf{T}'(\gamma)\mathbf{v}_1\right)$$

$$c\delta_2'(\gamma) = \mathbf{v}_2^\top\mathbf{T}'(\gamma)\mathbf{v}_1 + c\delta_1'(\gamma)(1-\mathbf{v}_2^\top\mathbf{T}(\gamma)\mathbf{v}) - c\delta_1(\gamma)\mathbf{v}_2^\top\mathbf{T}'(\gamma)\mathbf{v} = \mathbf{v}_2^\top\mathbf{T}'(\gamma)(\mathbf{v}_1-c\delta_1(\gamma)\mathbf{v}) + c\delta_1'(\gamma)(1-\mathbf{v}_2^\top\mathbf{T}(\gamma)\mathbf{v})$$

$$c\delta_3'(\gamma) = \mathbf{v}_1^\top\mathbf{T}'(\gamma)\mathbf{v}_1 + \frac{c^2\delta_1(\gamma)\left(2\delta_1'(\gamma)m(\gamma)-\delta_1(\gamma)m'(\gamma)\right)}{m^2(\gamma)}$$

$$c\delta_4'(\gamma) = \mathbf{v}_4^\top\mathbf{T}'(\gamma)\mathbf{v}_4 + m'(\gamma)\left(\mathbf{v}_4^\top\mathbf{T}(\gamma)\mathbf{v}-\frac{a_1}{c}\right)^2 + 2m(\gamma)\left(\mathbf{v}_4^\top\mathbf{T}(\gamma)\mathbf{v}-\frac{a_1}{c}\right)\mathbf{v}_4^\top\mathbf{T}'(\gamma)\mathbf{v},$$

with

$$\mathbf{T}'(\gamma) = \Delta_0'(\gamma)(\mathbf{I}_6+\Lambda_0\Delta_0(\gamma))^{-1} - \Delta_0(\gamma)(\mathbf{I}_6+\Lambda_0\Delta_0(\gamma))^{-1}\Lambda_0\Delta_0'(\gamma)(\mathbf{I}_6+\Lambda_0\Delta_0(\gamma))^{-1}$$

$$= (\mathbf{I}_6+\Delta_0(\gamma)\Lambda_0)^{-1}\Delta_0'(\gamma)(\mathbf{I}_6+\Lambda_0\Delta_0(\gamma))^{-1},$$

and $\Delta_0'(\gamma)$ as in the statement of Lemma 10.

Similarly, by their definition in Lemma 9, we obtain the derivatives of $\delta_5, \delta_6$ and $\delta_7$ as

$$\delta_5'(\gamma) = \frac{1}{c}\left(m'(\gamma)\left(-\mathbf{v}_2^\top\mathbf{T}(\gamma)\mathbf{v}+1\right) - m(\gamma)\mathbf{v}_2^\top\mathbf{T}'(\gamma)\mathbf{v}\right)$$

$$\delta_6'(\gamma) = \frac{1}{c}\left(\mathbf{v}_4^\top\mathbf{T}'(\gamma)\mathbf{v}_2 + m'(\gamma)\mathbf{v}_4^\top\mathbf{T}(\gamma)\mathbf{v}\mathbf{v}^\top\mathbf{T}(\gamma)\mathbf{v}_2 + m(\gamma)\mathbf{v}_4^\top\mathbf{T}'(\gamma)\mathbf{v}\mathbf{v}^\top\mathbf{T}(\gamma)\mathbf{v}_2\right.$$

$$\left.+m(\gamma)\mathbf{v}_4^\top\mathbf{T}(\gamma)\mathbf{v}\mathbf{v}^\top\mathbf{T}'(\gamma)\mathbf{v}_2 - m'(\gamma)\left(\frac{a_1}{c}\mathbf{v}_2+\mathbf{v}_4\right)^\top\mathbf{T}(\gamma)\mathbf{v} - m(\gamma)\left(\frac{a_1}{c}\mathbf{v}_2+\mathbf{v}_4\right)^\top\mathbf{T}'(\gamma)\mathbf{v} + \frac{a_1}{c}m'(\gamma)\right)$$

$$\delta_7'(\gamma) = \frac{1}{c}\left(\mathbf{v}_4^\top\mathbf{T}'(\gamma)\mathbf{v}_7 + m'(\gamma)\mathbf{v}_4^\top\mathbf{T}(\gamma)\mathbf{v}\mathbf{v}^\top\mathbf{T}(\gamma)\mathbf{v}_7 + m(\gamma)\mathbf{v}_4^\top\mathbf{T}'(\gamma)\mathbf{v}\mathbf{v}^\top\mathbf{T}(\gamma)\mathbf{v}_7 + m(\gamma)\mathbf{v}_4^\top\mathbf{T}(\gamma)\mathbf{v}\mathbf{v}^\top\mathbf{T}'(\gamma)\mathbf{v}_7\right.$$

$$\left.-\frac{a_1}{c}m'(\gamma)\left(\left(\frac{1}{c}+2\right)\mathbf{v}_4+\mathbf{v}_7\right)^\top\mathbf{T}(\gamma)\mathbf{v} - \frac{a_1}{c}m(\gamma)\left(\left(\frac{1}{c}+2\right)\mathbf{v}_4+\mathbf{v}_7\right)^\top\mathbf{T}'(\gamma)\mathbf{v} + \frac{a_1^2}{c^2}\left(\frac{1}{c}+2\right)m'(\gamma)\right).$$

This concludes the proof of Lemma 10. $\qquad\square$

Putting these together, we conclude the proof of Theorem 1.

## C.5 PROOF OF PROPOSITION 2

Here, we provide the proof of Proposition 2. By the definition of linear regression (in-context) memorization error $E_{\mathrm{LR}}$ in (14) of Definition 5, it suffices to evaluate the following quadratic form

$$\frac{1}{n}\mathbf{y}^\top \left(\frac{1}{n}\mathbf{X}^\top\mathbf{X} + \gamma\mathbf{I}_n\right)^{-1} \mathbf{y}, \tag{78}$$

and its derivative with respect to $\gamma$, for $\mathbf{X} = \boldsymbol{\mu}\mathbf{y}^\top + \mathbf{Z} \in \mathbb{R}^{p\times n}$ as in Theorem 1.

By Woodbury identity, we have

$$\frac{1}{n}\mathbf{y}^\top \left(\frac{1}{n}\mathbf{X}^\top\mathbf{X} + \gamma\mathbf{I}_n\right)^{-1} \mathbf{y} = \mathbf{e}_1^\top\mathbf{U}^\top \left(\frac{1}{n}\mathbf{Z}^\top\mathbf{Z} + \gamma\mathbf{I}_n + \mathbf{U}\boldsymbol{\Lambda}\mathbf{U}^\top\right)^{-1} \mathbf{U}\mathbf{e}_1$$

$$= \mathbf{e}_1^\top\mathbf{U}^\top\mathbf{Q}_0(\gamma)\mathbf{U}\left(\mathbf{I}_2 + \boldsymbol{\Lambda}\mathbf{U}^\top\mathbf{Q}_0(\gamma)\mathbf{U}\right)^{-1}\mathbf{e}_1,$$

where $\mathbf{e}_1 = [1,\ 0]^\top$ and with a slight abuse of notations, we denote

$$\mathbf{U} = [\mathbf{y},\ \mathbf{Z}^\top\boldsymbol{\mu}]/\sqrt{n} \in \mathbb{R}^{n\times 2}, \quad \boldsymbol{\Lambda} = \begin{bmatrix} \|\boldsymbol{\mu}\|^2 & 1 \\ 1 & 0 \end{bmatrix} \in \mathbb{R}^{2\times 2}, \quad \mathbf{Q}_0(\gamma) = \left(\frac{1}{n}\mathbf{Z}^\top\mathbf{Z} + \gamma\mathbf{I}_n\right)^{-1}. \tag{79}$$

Similar to Proposition 1, we have the following Deterministic Equivalent result for the *linear* resolvent $\mathbf{Q}_0(\gamma)$.

**Lemma 11** (Deterministic Equivalent for $\mathbf{Q}_0$, (Couillet & Liao, 2022, Theorem 2.4)). *Let $\mathbf{Z} \in \mathbb{R}^{p\times n}$ have i.i.d. standard Gaussian entries. Then, as $n, p \to \infty$ at the same pace with $p/n \to c \in (0, \infty)$ and $\gamma > 0$, the following Deterministic Equivalent (see Definition 4) holds*

$$\left(\frac{1}{n}\mathbf{Z}\mathbf{Z}^\top + \gamma\mathbf{I}_p\right)^{-1} \leftrightarrow m_{\mathrm{LR}}(\gamma) \cdot \mathbf{I}_p, \quad \left(\frac{1}{n}\mathbf{Z}^\top\mathbf{Z} + \gamma\mathbf{I}_n\right)^{-1} \leftrightarrow \left(cm_{\mathrm{LR}}(\gamma) + \frac{1-c}{\gamma}\right)\mathbf{I}_n.$$

*with $m(\gamma)$ is the unique Stieltjes transform solution to the following Marčenko-Pastur equation (Marcenko & Pastur, 1967; Couillet & Liao, 2022)*

$$c\gamma m_{\mathrm{LR}}^2(\gamma) + (1 - c + \gamma)\, m_{\mathrm{LR}}(\gamma) - 1 = 0. \tag{80}$$

By Lemma 11, we have

$$\frac{1}{n}\mathbf{y}^\top \left(\frac{1}{n}\mathbf{X}^\top\mathbf{X} + \gamma\mathbf{I}_n\right)^{-1} \mathbf{y} = \mathbf{e}_1^\top\mathbf{U}^\top\mathbf{Q}_0(\gamma)\mathbf{U}\left(\mathbf{I}_2 + \boldsymbol{\Lambda}\mathbf{U}^\top\mathbf{Q}_0(\gamma)\mathbf{U}\right)^{-1}\mathbf{e}_1$$

$$= \left[\begin{bmatrix} cm_{\mathrm{LR}}(\gamma) + \frac{1-c}{\gamma} & 0 \\ 0 & \|\boldsymbol{\mu}\|^2(1 - \gamma m_{\mathrm{LR}}(\gamma)) \end{bmatrix}\begin{bmatrix} 1 + \|\boldsymbol{\mu}\|^2\left(cm_{\mathrm{LR}}(\gamma) + \frac{1-c}{\gamma}\right) & \|\boldsymbol{\mu}\|^2(1 - \gamma m_{\mathrm{LR}}(\gamma)) \\ cm_{\mathrm{LR}}(\gamma) + \frac{1-c}{\gamma} & 1 \end{bmatrix}^{-1}\right]_{1,1}$$

$$= \frac{cm_{\mathrm{LR}}(\gamma) + \frac{1-c}{\gamma}}{1 + \|\boldsymbol{\mu}\|^2(1 - \gamma m_{\mathrm{LR}}(\gamma))},$$

so that by (14), we obtain

$$E_{\mathrm{LR}} - \bar{E}_{\mathrm{LR}} \to 0, \quad \bar{E}_{\mathrm{LR}} = -\frac{c\gamma^2 m'(\gamma) + c - 1 + \|\boldsymbol{\mu}\|^2\left(\gamma^2 m'(\gamma) + (1 - c - \gamma)(\gamma m(\gamma) - 1)\right)}{(1 + \|\boldsymbol{\mu}\|^2 - \|\boldsymbol{\mu}\|^2\gamma m_{\mathrm{LR}}(\gamma))^2}, \tag{81}$$

in probability as $n, p \to \infty$, with $m_{\mathrm{LR}}(\gamma)$ the Stieltjes transform solution to the Marčenko-Pastur equation in (80), and $m'_{\mathrm{LR}}(\gamma) = -\frac{cm^2(\gamma)+m(\gamma)}{2c\gamma m(\gamma)+1-c+\gamma}$ its derivative with respect to $\gamma$.

This concludes the proof of Proposition 2.

## D ADDITIONAL NUMERICAL RESULTS AND DISCUSSIONS

In this section, we present additional numerical results.

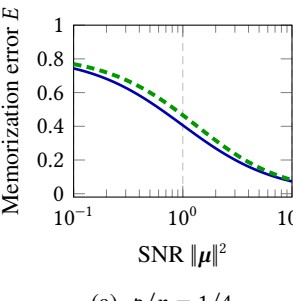 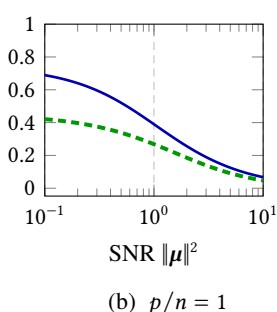 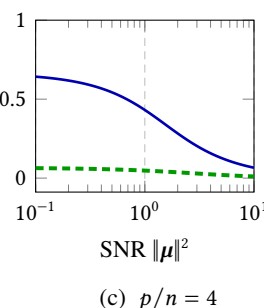

(a) $p/n = 1/4$        (b) $p/n = 1$        (c) $p/n = 4$

Figure 4: Theoretical in-context memorization error for $f(t) = \tanh(t)$ (**blue**) from Theorem 1 versus that of linear regression (**green**) from Proposition 2, as a function of SNR, for different dimension ratio $p/n$, synthetic data drawn from the Gaussian signal-plus-noise model in Definition 2 with $\mathbf{w}_K = \mathbf{w}_Q = \boldsymbol{\mu}_{\mathrm{base}} \sim \mathcal{N}(\mathbf{0}, \mathbf{1}_p/p)$, $\boldsymbol{\mu} \propto \boldsymbol{\mu}_{\mathrm{base}}$, and $\gamma = 1$.

Figure 4 compare the theoretical in-context memorization errors of nonlinear Attention (as characterized in Theorem 1) with those of linear linear regression (from Proposition 2) on synthetic Gaussian mixture data. We observe that, while linear regression generally achieves lower memorization error than nonlinear Attention in the under-determined $p > n$ regime, this advantage is reversed in the over-determined setting with $p/n < 1$. In such cases, nonlinear Attention yields *lowers* error, for structured inputs and Attention weights aligned to the data signal. Furthermore, compared to linear regression, the memorization error of nonlinear Attention exhibits remarkably less sensitivity to the dimension ratio $p/n$, especially when the Attention weights are well aligned with the underlying signal in the input data.

Figure 5 further illustrates the impact of the Attention nonlinearity, the dimension ratio $p/n$, and the regularization parameter $\gamma$ on the in-context memorization errors of nonlinear/linear Attention and linear linear regression. Reading the subfigures from left to right, we observe that the difference in memorization error between different Attention (i.e., tanh nonlinear or truncated linear) and linear regression vanishes either as the regularization strength $\gamma$ decreases *or* as the SNR increases. Moreover, the advantage of nonlinear Attention over linear regression—in terms of reduced memorization error—critically depends on both the dimension ratio $p/n$ (as already confirmed in Figure 4) *and* the choice of regularization $\gamma$, see for example Figure 5d versus Figure 5e. Reading the subfigures from top to bottom, we further observe that in the over-determined $p/n < 1$ regime, the memorization error of nonlinear Attention is considerably less sensitive to the changes in the dimension ratio $p/n$ compared to linear regression.

Figure 6 illustrates the impact of alignment between the Attention weights (the query $\mathbf{w}_Q$ and key $\mathbf{w}_K$ vectors in Assumption 1) and the input data signal $\boldsymbol{\mu}$. A consistent pattern emerges from Figure 6: when the Attention weights are aligned in direction with $\boldsymbol{\mu}$, the resulting in-context memorization error is significantly lower compared to the case where the weights are orthogonal to $\boldsymbol{\mu}$. This effect is observed across both nonlinearities considered: $f(t) = \tanh(t)$ and truncated linear function $f(t) = \max(-5, \min(5, t))$, and persists across a range of SNR values and dimension ratios $p/n$. The performance gain from the weight alignment is particularly pronounced in the over-determined $p/n < 1$ setting.

Figure 7 compares the in-context memorization error curves of nonlinear Attention using weights extracted from a pretrained GPT-2 model against our theoretical predictions from Theorem 1, across varying regularization strengths, SNR levels, and activation nonlinearities. This numerical experiment serves to empirically validate the full-plus-low-rank decomposition of Attention weights posited in Assumption 1.

To extract the Attention weights $\mathbf{W}_Q$ and $\mathbf{W}_K$, we use the first Attention head from the 1st, 7th, and 12th Transformer layers of a pretrained GPT-2 model (accessed via HuggingFace). Specifically, we extract the first and second $m$-sized column blocks from the projection matrix `model.transformer.h[l].attn.c_attn.weight` (of shape $m \times 3m$ with $m = 768$) as query and key weight matrices. The weights for a single head are then obtained by selecting the

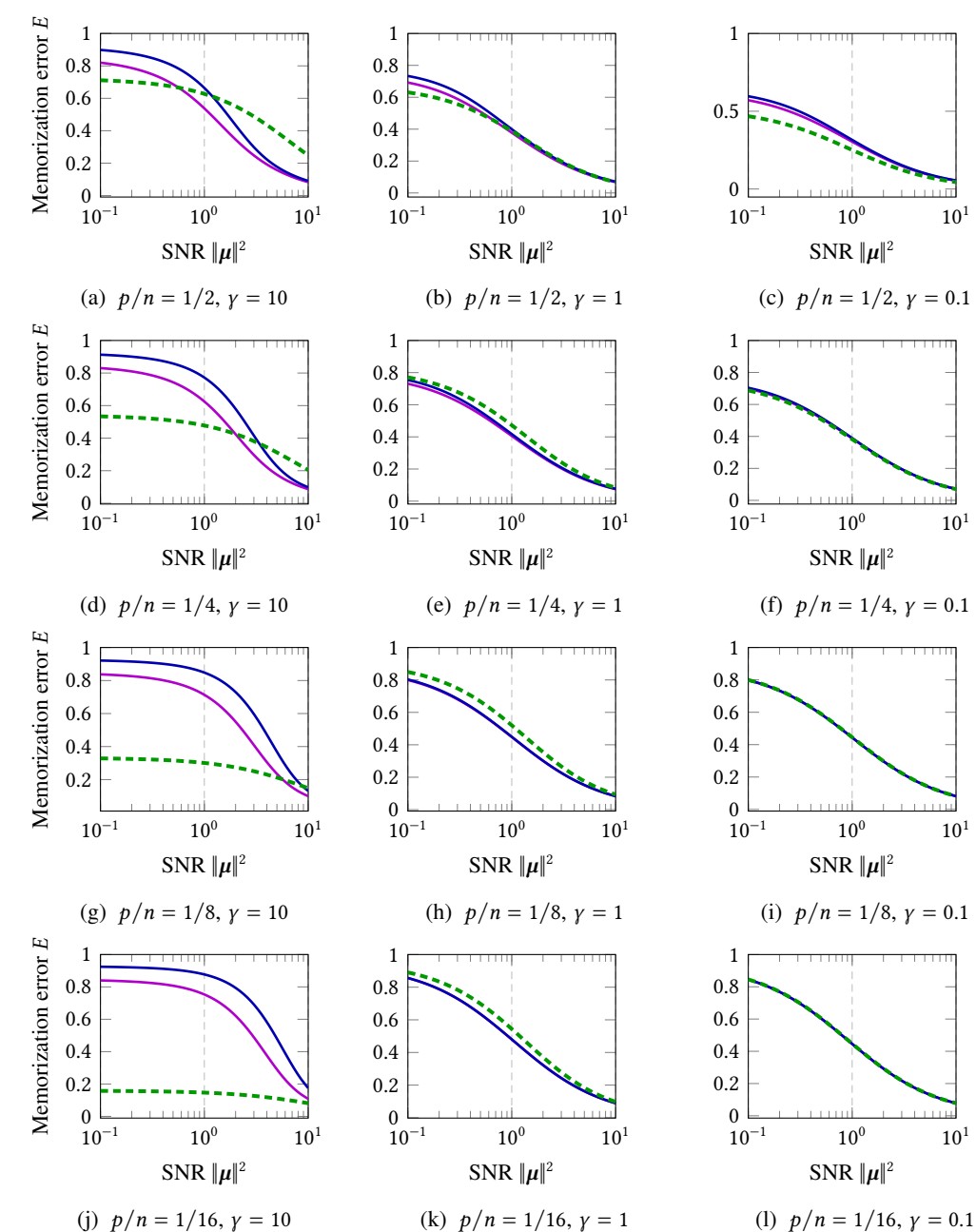

(a) $p/n = 1/2, \gamma = 10$     (b) $p/n = 1/2, \gamma = 1$     (c) $p/n = 1/2, \gamma = 0.1$

(d) $p/n = 1/4, \gamma = 10$     (e) $p/n = 1/4, \gamma = 1$     (f) $p/n = 1/4, \gamma = 0.1$

(g) $p/n = 1/8, \gamma = 10$     (h) $p/n = 1/8, \gamma = 1$     (i) $p/n = 1/8, \gamma = 0.1$

(j) $p/n = 1/16, \gamma = 10$     (k) $p/n = 1/16, \gamma = 1$     (l) $p/n = 1/16, \gamma = 0.1$

Figure 5: Theoretical in-context memorization error for $f(t) = \tanh(t)$ (**blue**) versus $f(t) = \max(-5, \min(5, t))$ (**purple**) and that of linear regression (**green**) in the over-determined regime, as a function of SNR, for different dimension ratio $p/n$ and regularization parameter $\gamma$, synthetic data drawn from the Gaussian signal-plus-noise model in Definition 2 with $\mathbf{w}_K = \mathbf{w}_Q = \boldsymbol{\mu}_{\text{base}} \sim \mathcal{N}(\mathbf{0}, \mathbf{I}_p/p)$, $\boldsymbol{\mu} \propto \boldsymbol{\mu}_{\text{base}}$.

first $m_{\text{head}} = m/n_{\text{heads}} = 64$ columns from each matrix, consistent with the model's $n_{\text{heads}} = 12$-head configuration.

As shown in Figure 7, the empirical memorization curves obtained from pretrained Attention weights closely match the theoretical trends predicted by Theorem 1, as a function of both regularization strength $\gamma$ and SNR. In particular, we observe that

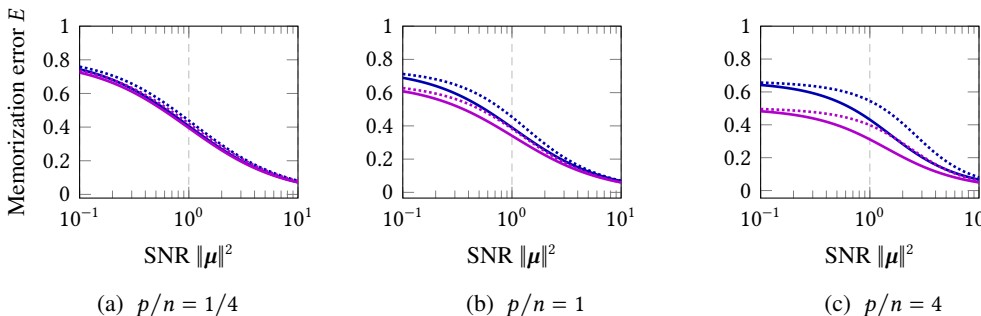

(a) $p/n = 1/4$        (b) $p/n = 1$        (c) $p/n = 4$

Figure 6: Theoretical in-context memorization error of tanh (**blue**) and truncated linear (with $f(t) = \max(-5, \min(5, t))$ in **purple**) Transformer, for key/query weights aligned with the signal direction in solid lines: $\mathbf{w}_K = \mathbf{w}_Q = \boldsymbol{\mu}_{\text{base}} \sim \mathcal{N}(\mathbf{0}, \mathbf{I}_p/p)$ and $\boldsymbol{\mu} \propto \boldsymbol{\mu}_{\text{base}}$; versus the case where both weights orthogonal to the signal in dotted lines: $\mathbf{w}_K \perp \boldsymbol{\mu}_{\text{base}}, \mathbf{w}_Q \perp \boldsymbol{\mu}_{\text{base}}, \mathbf{w}_K \perp \mathbf{w}_Q$ and $\boldsymbol{\mu} \propto \boldsymbol{\mu}_{\text{base}}$; for regularization strength $\gamma = 1$.

1. in the absence of input data signal ($\boldsymbol{\mu} = \mathbf{0}$), pretrained Attention weights yield slightly lower errors than theory; and

2. in the presence of signal, pretrained Attentions perform marginally worse than theory from (manually) aligned weights.

These discrepancies are generally modest in scale and consistent across both the tanh and truncated exponential nonlinearities. Additionally, we observe that Softmax Attention incurs substantially higher memorization error than entrywise exponential Attention, but only when meaningful input structure is present. This qualitative agreement in observed trends further supports that the simplified Attention model in Assumption 1 serves as a reasonable abstraction, even when compared to real pretrained GPT-2 weights.

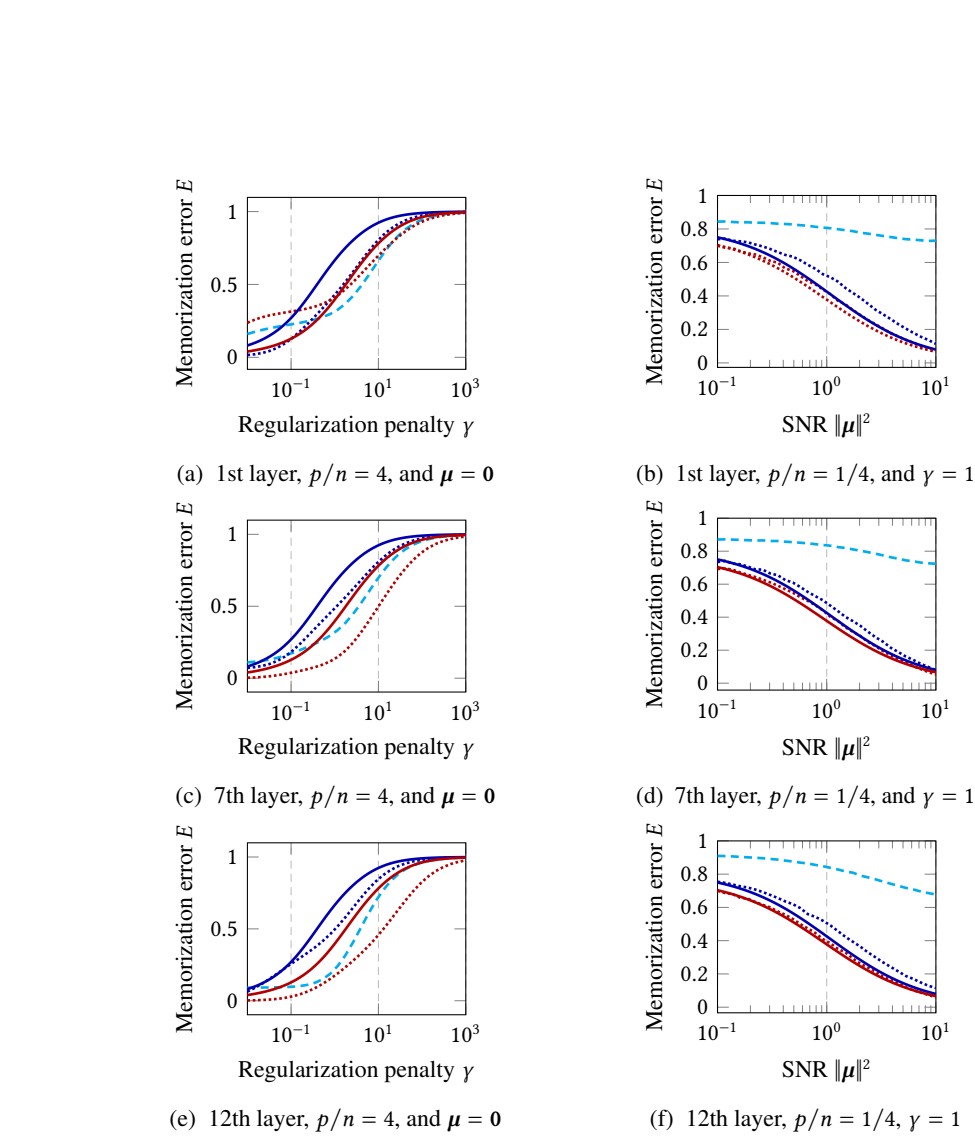

Figure 7: Theoretical in-context memorization error of Softmax (cyan) and entry-wise tanh (blue), truncated exponential ($f(t) = \min(5, \exp(t))$ in red) Attention. Theoretical predictions under Assumption 1 in solid lines and key/query weights using pretrained Attention weights in dotted lines. **Figure 7a**, **Figure 7c**, and **Figure 7e**: theoretical predictions obtained by assuming $\mathbf{w}_K = \mathbf{w}_Q = \boldsymbol{\mu} = \mathbf{0}$; **Figure 7b**, **Figure 7d**, and **Figure 7f**: theoretical predictions obtained by assuming $\mathbf{w}_K = \mathbf{w}_Q = \boldsymbol{\mu}_{\text{base}} \sim \mathcal{N}(\mathbf{0}, \mathbf{I}_p/p)$, $\boldsymbol{\mu} \propto \boldsymbol{\mu}_{\text{base}}$ and $\gamma = 1$.

