# OpenReview forum: "A Random Matrix Analysis of In-context Memorization for Nonlinear Attention"
_ICLR.cc/2026/Conference — Submitted to ICLR 2026_

### Official Review · Reviewer_3wik · 2025-10-30

**Soundness:** 3
**Presentation:** 3
**Contribution:** 2
**Rating:** 6
**Confidence:** 4

**Summary:**

The paper gives a high-dimensional characterization of the in-context
memorization error (training MSE of the best ridge linear probe fitted on
the same context) when features come from a single-head nonlinear
attention block with fixed query/key/value weights. Under a Gaussian
signal+noise model for tokens $X=\mu y^\top+Z$ and a full-plus-low-
rank structure for $W_K^\top W_Q=I+w_Kw_Q^\top$, the authors
obtain a closed deterministic $9\times 9$ system whose solution yields
the limiting in-context training error $\bar E_A$ (Theorem 1). This
enables clean comparisons against linear regression (LR) on raw $X$
(Proposition 2).
Empirically/theoretically they find: with random inputs ($\mu=0$),
nonlinear attention often underperforms LR; but with structured inputs
and alignment between attention weights and $\mu$, the gap vanishes
or reverses. They also show memorization critically requires a linear
component in $f$ (nonzero $a_1$).

**Strengths:**

The authors analyze a random feature model with a related regression task, where the feature map has structural properties shared with attention layers. They also compare linear and non-linear attention, which is a topic that may have impact on practical debates on the usefulness of non-linearities in attention.

**Weaknesses:**

In my understanding, the paper does not really study attention, as the crucial non-linearity (usually a row-wise softmax) is applied entry-wise (while Remark 5 in Appendix B argues that this is not important, it will be as soon as the attention weights are learned - see next comment - but I am happy to be disproven). Additionally, not learning the attention weights (keys and queries in particular) makes the model a random feature model (albeit structured) in disguise (as discussed in lines 139-142).

The phenomenology observed in Sections 4.2 and 4.3 is standard of random features regression: adding a non-linearity introduces additional noise. See for example Mei and Montanari '21. It is not clear to me which observations here are specific to attention layers, and which are just random feature ones.

**Questions:**

line 55: I find it a bit unfair to call "stylized models" the works Tiberi '24, Troiani '25 and Rende '24. The phrasing of this line and the next paragraph seems to suggest that this paper goes beyond such stylized models, while model line 173 is stylized as well.

line 200: why does it make sense to regularize the learned weights as || W^T_V w ||^2 instead of ||w||^2? I understand that this simplifies the expressions by virtually removing the value weights from the solution eq (5), but it is still quite a peculiar choice. Does it have some intuitive meaning?

Def 3: it is not clear to me why the error is called "in-context memorization". For me, in-context means that each input sample to the attention specifies the task that the attention needs to solve. Here I only see a classic regression task in which, independently for each sample {x,y}, one just minimizes the L2 error between y and a random feature predictor applied to x. I really fail to see the interpretation as in-context: could you try to explain this better? For comparison, Lu et al '25 truly provides an in-context learning task in my opinion.

Proposition 1: is this a direct application of (Pennington & Worah, 2017)?

Regarding the element-wise non-linearity, I wanted to mention the recent paper https://arxiv.org/pdf/2510.06685 (appeared after the submission deadline, so I am not asking for a comparison, just mentioning it for completeness), where a treatment of the true softmax seems possible.

---

> ### Author Response · Authors · 2025-11-24
>
> Thank you for your positive support as well as detailed and constructive comments.
> Below, we provide point-by-point responses to your concerns and questions.
> All modifications are marked in red of the revised paper.
>
> - **Weakness 1: difference from random-feature models**:
> We could like to clarify that the model under study (that takes the form $X f(X^\top W_K^\top W_Q X)$) differs significantly from random-feature models of the type $f(W X)$ in the following senses: (1) we consider *deterministic* weights $w_K$ and $w_Q$ here, as opposed to *random* weights $W$ in the case of random-feature models; and (2) more importantly, the nonlinear (Attention kernel) matrix $K_X = f(X^\top W_K^\top W_Q X)$ mixes the (columns of the) input $X$ in a **data dependent fashion**, since $K_X$ depends on $X$ in a highly nonlinear fashion.
> In fact, while we do not **explicitly** consider the (pre-)training procedure of model, we do so **implicitly** by considering the Attention weights (obtained from pre-training) to be *arbitrary deterministic* vectors (Assumption 1), where $w_K$ and $w_Q$ can take any values.
> And the model under study here aligns with those in the litterature of in-context learning.
>
> - **Question 1: claim of "stylized model"**:
> Thank you for pointing this out.
> We agree you that the wording of "stylized models" could be imprecise and misleading.
> In fact, we agree with you that the model under study is a **simplified yet stylized model**, see our detailed argument in the global reply.
> We have clarified that the cited previous efforts and ours use different "stylized" frameworks with different focuses.
>
> - **Question 2: regularization of $|| W_V^\top w ||^2$**:
> We regularize $|| W_v^\top w ||^2$ because the linear probing output depends on $w$ only via $W_v^\top w$.
> Regularizing this effective parameter is natural and also simplifies the derivations.
> It can also be checked that, in some cases, for example orthonormal $W_v$, the two types of regularizations yields the **same** memorization error up to a change variable on the regularization penalty $\gamma$.
>
>
> - **Question 3: "in-context memorization"**:
> We would like to clarify that the problem studied in this paper is indeed an in-context memorization problem.
> A key distinction of our analysis, however, is that we **implicitly** account for the effect of pre-training by allowing the Attention weights to be *arbitrary deterministic* vectors, as formalized in Assumption 1, where $w_K$ and $w_Q$ can take any values.
> This choice allows our theory to directly capture scenarios where the pretrained weights are aligned with, orthogonal to the signal direction $\mu$, or anywhere in between, as discussed and numerically supported in Section 4 and Appendix D.
> This is formally different from most existing analyses (e.g., [Lu et al '25]), which typically incorporate pre-training **explicitly** by modeling a particular learning algorithm (e.g., gradient descent).
> Thus, although we abstract away the pre-training dynamics, our setting still naturally fits within the standard two-stage formulation of in-context learning (i.e., pre-training and then inference).
> See also more detailed discussions in item **connections to and differences from prior work on in-context learning/memorization** of our global reply.
> On the other hand, [Lu et al '25] studied a simplified linear Attention model, which allows for **explicit** characterization of the pretrained weights, and evaluate the resulting model's in-context generalization performance on new tasks.
>
>
> - **Question 4: Difference from [Pennington & Worah, 2017]**:
> We would like to clarify that our Proposition 1 is **not** a direct application of [Pennington and Worah '17].
> The model studied by Pennington and Worah takes the form $Y^\top Y$ for $Y = f(W X)$ for independent $W$ and $X$, while the model studied in our Proposition 1 is $Y^\top Y$ for $Y = X f(X^\top X)$.
> The latter involves nonlinear dependence (between $X$ and $f(X^\top X)$) and cannot be handled, e.g., using the free probability approach adopted in [Pennington and Worah '17].
> As a consequence, we believe that the proposed analysis framework may be of independent interest for studying more involved Transformer-type architectures or token-mixing mechanisms.
> See also more detailed discussions in item **technical contribution** of our global reply.

---

> > ### Comment · Reviewer_3wik · 2025-11-25
> >
> > I thank the authors for engaging with my comments. I have some further remarks that the authors could comment upon.
> >
> > ## On the in-context issue
> >
> > Reviewer nAJP, cebM, QfQH and myself all highlighted that this task should not be classified as in-context, or at least expressed doubts on this point. In all the answers to the reviewers, and in the global one, I do not see this point addressed properly, save for a small discussion in the answer to QfQH. I invite the authors to consider adding this specific intuition to the paper, as it would start to clarify the issue. I am still conflicted whether I would call this in-context learning without a statistical model on the pre-training, or more out-of-distribution generalization. I let the area chair decide if this is semantics or an important point.
> >
> > ## Element-wise activation function
> >
> > The authors did not discuss this weakness I highlighted unless I missed it. Softmax is not an entry-wise function, it importantly mixes tokens!
> >
> > ## On fixed weights
> >
> > The authors claim in multiple places that their weights can be generic, but that is not precise: they are identity plus a deterministic rank-1 spike (assumption that they justify somehow in the paper). Many phenomena in RMT are (loosely speaking) independent on the details of the spike, e.g. the Baik-Ben Arous-Peche transition, so while the actual technical machinery of the proof may need to take this into account, from the "what do we learn" point of view I would argue that claiming that the weights are generic is a bit of a stretch. One rank-1 spike in the weights is generic, the orthogonal complement of such spike is the identity, far from a generic fixed set of weights.

---

> > > ### Author Response · Authors · 2025-11-28
> > >
> > > Thank you for your prompt reply and further constructive comments.
> > >
> > > ## On the in-context issue
> > >
> > > Thanks for this comment, which prompts us to further clarify the relation between our work and existing efforts on ICL.
> > > When we say that our setting *in spirit* aligns with in-context learning, we do mention that our analysis is different in the following perspectives:
> > >
> > > 1. We consider *deterministic* K, Q weights as a consequence of the pre-training, rather than being obtained from some specific pre-training tasks (that are assumed to follow some statistical models in most existing litterature); and
> > > 2. we study the statistical memorization/training performance instead of the (out of distribution, if one models the pre-training task) generalization performance.
> > >
> > > We have chosen to use the terminology "in-context", since we believe that this study, by investigating the behavior of a fixed Attention for in-context examples (where the fixed weights can be viewed as an abstract consequence of pre-training), aligns with the previous line of work on ICL at a conceptual level.
> > >
> > > Following the discussions with all reviewers, we notice that this terminology may cause confusion that our analysis should be **directly** compared and contrasted with existing results on in-context generalization performance, that are different from our results in the aforementioned ways.
> > >
> > > To avoid such misunderstanding, we would be happy to **reword the "in-context" in the title and throughout the manuscript**, as long as the reviewers think that this better highlights the contribution of this work.
> > > Once again, we would like to emphasize that our focus here is on the analysis of the memorization/training performance of a given Attention under structured input samples, rather than on its pre-training procedure or out-of-distribution generalization performance.
> > >
> > > ## Element-wise activation function
> > > We would like to clarify that a series of existing results interpret token-mixing **in a broad sense**: along the sequence dimension, features from different positions are combined through a (data-dependent or learnable) matrix, allowing thus each output position to depend on multiple input positions.
> > > In particular, this interpretation is *not* limited to the specific form of Softmax Attention.
> > >
> > > In this broader sense, the entry-wise Attention considered in our Definition 1 also achieves token-mixing along the sequence dimension.
> > > Further note that such entry-wise Attentions are used in practical (Vision and beyond) Transformer models, and are considered to be computationally advantageous over standard Softmax Attention, see for example (Wortsman et al., 2023; Ramapuram et al., 2024) in the list of references.
> > >
> > > From a theoretical perspective, as discussed in Remark 5 of the appendix, using the truncated exponential function $ f(t) = \min( \exp(t), C) $ can theoretically approximate (the Attention matrix) of Softmax Attention in our setting.
> > > From an empirical perspective, we compare, in Figure 7, Softmax Attention versus other entry-wise Attentions, and observe remarkably consistent trends in their memorization errors.
> > >
> > > Following the approach in the paper that you've pointed to us, a complete extension to softmax Attention is indeed possible, at least in some settings.
> > > This will be appropriately discussed in the revision.
> > > But for the clarity of our message, we believe that it would be better *not* to include such detailed results in this paper, since they are not fully compatible with our results.
> > >
> > > ## On fixed weights
> > > We agree that the term "generic weights" may not be sufficiently precise.
> > > The weights considered in this paper possess a well-defined structure and should more accurately be described as "structured deterministic weights."
> > > We have made corresponding revisions in the revised version.

---

### Official Review · Reviewer_cebM · 2025-10-31

**Soundness:** 3
**Presentation:** 3
**Contribution:** 1
**Rating:** 2
**Confidence:** 3

**Summary:**

The paper studies one layer of attention with element-wise non-linearity. Specifically, the authors consider the problem of learning a mixture of two isotropic gaussians using either a linear model or with one layer of attention by training only the values and keeping the key-query product equal to identity plus (untrained) rank one spike. The memorization error is studied through a rigorous deterministic equivalence in the proportional regime of large number of samples and large embedding dimension, which are then validated by numerical experiments.

**Strengths:**

The paper is well written and easy to follow, has a clear exposition and shows the implications of its assumptions clearly. The theoretical study of attention networks is of current interest, and results in this direction are valuable for bridging the gap between empirical performance and principled understanding. I believe the theorems to be correct and their derivations clearly written.

**Weaknesses:**

The specific setting under analysis is not particularly relevant: the authors study a single layer of attention without even training the key and query weights and without row-wise softmax activation. Additionally, I am not sure the task at hand is in fact in-context learning and I would like this aspect to be clarified by the authors. I don't think paper has enough technical novelty to compensate for the shortcomings in its settings. In particular I think that the element-wise instead of row-wise activation, as well as the strong assumption on the query-key weight matrix make the technical contributions somewhat modest. At its current stage, this manuscript does not offer much more than a random matrix theory exercise, and I would like to see the dependence on all the relevant parameters in the model fully explored.

Minor issue:
Figure 2 contains the lines already presented in Figure 1, they could be combined.

**Questions:**

1. Could you clarify in which sense is your paper studying an in-context learning task?
2. Would you give precise insights over what is the technical novelty of your work with respect to existing literature on the high-dimensional analysis of attention models?
3. Would you compare with [Cui, Behrens, Krzakala, Zdeborova 2024]?
4. Do you know how would your results change if you consider row-wise softmax attention (even just experimentally)?
5. In your analysis you never find a setting where attention outperforms a linear model. Would you comment on this, in particular in relation with not training the key-query matrix?

---

> ### Author Response · Authors · 2025-11-24
>
> We sincerely thank you for your detailed and constructive comments.
> Below, we provide point-by-point responses to your concerns and questions. All modifications are marked in red of the revised paper.
>
> - **Question 1: What "in-context" means here**:
> We would like to clarify that the setup considered in this paper conforms to standard in-context learning/memorization in the literature.
> A major difference of our analysis from existing work is that we **implicitly** account for the effect of pre-training by allowing the Attention weights to be *arbitrary deterministic* vectors (Assumption 1), where $w_K$ and $w_Q$ can take any values.
> This choice allows our theory to directly capture scenarios where the pretrained weights are aligned with, orthogonal to the signal direction $\mu$, or anywhere in between, as discussed and numerically supported in Section 4 and Appendix D.
> This is formally different from prior arts that typically
> incorporate pre-training **explicitly** by modeling a particular learning algorithm (e.g., gradient descent).
> Thus, we focus on the behavior of a pretrained Attention during inference, as a function of the in-context input, and without any parameter updates.
> This is consistent with the common definition of in-context learning.
> But here, we focus on the in-context memorization/training performance instead of in-context generalization performance.
> We have clarified this point in the revised introduction.
> See also the item **connections to and differences from prior work on in-context learning/memorization** in our global reply for further discussions.
>
> - **Question 2: Technical novelty**:
> As discussed in item **technical contribution** of our global reply above, the main technical contribution of this paper is not to (ambitiously) introduce a novel random matrix proof approach, but to connect the (existing) idea of Hermite polynomial expansion and random kernel matrix model to the nonlinear Attention model.
> We believe that this connection could be of independent interest for studying more involved Transformer-type architectures or token-mixing mechanisms.
>
> - **Question 3: Connection to [Cui, Behrens, Krzakala, Zdeborova 2024]**:
> Thank you for pointing us to this highly related paper, which we have discussed in the revised version.
> The setup of [Cui'24] is different from ours in the following perspectives: the inputs considered therein are uncorrelated (1-gram) words, while we consider a Gaussian signal-plus-noise model with statistical means (that can thus correlat with the Attention weights); the Attention weights considered in [Cui'24] are of low rank, while those considered in our paper are of high rank (being the sum of a full rank identity part and a non-symmetric rank one $w_K w_Q^\top$ part, that can be aligned with or orthogonal to the in-context data structure).
> Also, the effect of positional encoding is considered in [Cui'24], but not in our analysis.
> From a technical perspective, [Cui'24] adopts a Generalized Approximate Message Passing (GAMP) approach, while our analysis relies on random matrix techniques, in particular, the connection between random kernel matrix model and the nonlinear Attention model.
>
>
> - **Question 4: Extension to row-wise softmax Attention**:
> Thank you for this question.
> Note that Figure 7 in the appendix already includes numerical experiments on row-wise softmax Attention, where the softmax Attention memorization errors exhibit same qualitative trends as functions of regularization, SNR, and aspect ratio akin to other entrywise nonlinearities Attention studied in the paper (e.g., tanh, truncated exponential).
> But the errors are higher as a function of SNR.
> We believe that the proposed analysis extends to row-wise softmax Attention with some additional efforts, and have discussed some pistes and challenges in Remark 5 of the appendix.

---

> > ### Author Response · Authors · 2025-11-24
> >
> > -**Question 5: Attention versus linear regression**:
> > We would like to clarify that our experiments **do** include cases where **Attention outperforms linear regression baseline**.
> > In Figure 5 of the appendix, we compare the memorization errors of Attention and linear (ridge) regression across different SNR levels, aspect ratios, and regularization strengths.
> > These results consistently show that for small $ p/n $ ratios and appropriate $ \gamma $, Attention yields lower in-context memorization errors than ridge regression, when the Attention weights are aligned with the in-context signal.
> > We would also like to clarify that while we do not **explicitly** consider the (pre-)training procedure of model, we do so **implicitly** by considering the Attention weights (obtained from pre-training) to be *arbitrary deterministic* vectors (Assumption 1), where $w_K$ and $w_Q$ can take any values.
> > This is aligned with the scope of the paper, which focuses on the post-training memorization capacity of a given Attention model, and how and when it outperforms the linear regression baseline.
> > See also the item **connections to and differences from prior work on in-context learning/memorization** in our global reply for more detailed discussions.

---

### Official Review · Reviewer_nAJP · 2025-11-02

**Soundness:** 2
**Presentation:** 2
**Contribution:** 2
**Rating:** 2
**Confidence:** 4

**Summary:**

This work studies the training error of a linear ridge regression problem that uses pointwise nonlinear attention as a feature extractor for a sequence of tokens $X$. The analysis assumes the nonlinear attention matrices follow a full-rank plus low-rank structure. The work characterizes the regression error in the high-dimensional asymptotic limit, where the number of tokens $n$, their embedding dimension $p$, and the inner dimension of attention product matrices $d$ all blow up. The data is modeled using a signal-plus-noise model where each token $x_i$ consists of a signal component $y_i\mu$ and isotropic noise $z_i$. This error is then compared to that of a standard linear regression model.

**Strengths:**

The work's primary strength is its technical contribution. It builds on classical random matrix tools, often used in benign overfitting studies of linear regression, by successfully deriving a deterministic equivalent for the Gram matrix using the non-linear attention kernel in a high-dimensional asymptotic regime.

**Weaknesses:**

**Missing key related work**: The motivation to study this specific model is not well-introduced. The introduction states that a key challenge is the nonlinearity of attention, which is a fair point. However, it then proceeds to cite prior work that largely “reduces attention to generalized linear models”, while failing to mention or engage with several recent works that directly study nonlinear softmax attention [1, 2, 3, 4, 5, 6] (see [2] for a more comprehensive list of related work).


**Issues with framing the problem**: The introduction also introduces the term "in-context memorization" without sufficiently defining the problem. This points to a larger, more fundamental problem with the paper's framing. The problem setup, in its current form, does not appear to be an "in-context" task. In Def 2, each token $x_i$ has a signal component $y_i\mu$ plus noise, and the goal is to isolate signal from noise and map this to $y_i$. The signal direction $\mu$ is fixed across all contexts, hence the task remains the same across contexts. In-context learning, by contrast, typically implies that the underlying task or function changes with the context (e.g., learning different linear regressions or Markov chains from the given context). Since the task here remains consistent, the framing as "in-context" is inappropriate.


The setup is more of the flavour of classical regression task (which the authors correctly identify as a baseline in (12)) where $n$ (x, y) iid pairs are drawn from a distribution. The work is studying the training error of ridge regression on a signal-plus-noise model using attention as a feature extractor, which is more similar to the line of work in [7] than to the ICL literature.


Furthermore, the term "in-context memorization" has a specific meaning in the ICL literature, where it typically refers to a model memorizing a finite set of tasks from its pretraining data rather than generalizing to unseen ones [8, 9, 10, 11]. The paper defines it (Def 3) as the standard training error of the linear probe. Calling this "in-context memorization" is a misrepresentation of the problem being studied.

Finally, this signal-plus-noise token learning framework has been studied in other theoretical analyses of attention [1, 3] (also not discussed when introducing the data model).

**References**

[1] Deora et al. (2023) On the Optimization and Generalization of Multi-head Attention.

[2] He et al. (2025) In-Context Linear Regression Demystified: Training Dynamics and
Mechanistic Interpretability of Multi-Head Softmax Attention

[3] Vasudeva et al. (2024) Implicit Bias and Fast Convergence Rates for Self-Attention

[4] Chen et al. (2024) Unveiling Induction Heads: Provable Training Dynamics and Feature
Learning in Transformers

[5] Tarzanagh et al. (2023) Transformers as SVMs.

[6] Huang et al. (2023) In-context convergence of Transformers

[7] Boncoraglio et al. (2025) Bayes optimal learning of attention-indexed models.

[8] Singh et al. (2023) Transient nature of in-context learning

[9] Park et al. (2024) Algorithmic phases of icl

[10] Lin et al. (2024) Dual operating modes of icl

[11] Raventos et al. (2023) Preptraining task diversity and the emergence of non-bayesian icl for regression

**Questions:**

Please see weaknesses section.

---

> ### Author Response · Authors · 2025-11-24
>
> Thanks you for your constructive comments that have helped us significantly improve the paper.
> Below we provide point-by-point responses to your concerns.
> All modifications are marked in red of the revised paper.
>
> - **Missing prior work and connections to our analysis**:
> Thank you for these related prior work and they have now been appropriately discussed in the revised version.
> We would like to clarify that, as discussed in item **connections to and differences from prior work on in-context learning/memorization** in the global reply, the setting under study is indeed an instance of in-context memorization: we analyze the inference-time behavior of a fixed pretrained nonlinear Attention module memorizing some in-context examples.
> The results in [8–11] that you mentioned, on the other hand, mainly focus on how in-context learning abilities emerge during pre-training, e.g., through Bayesian retrieval, competition dynamics, dual operating modes, or task-diversity effects.
> This is different from our analysis, which does not (statistically) model the pre-training procedure, but instead characterizes the inference-time memorization behavior of a pretrained nonlinear Attention under a structured signal-plus-noise model.
> This allows us to precisely characterize key interactions between Attention nonlinearity, the structure of the pretrained weights, and their alignment (or misalignment) with the in-context task.
> From a technical perspective, we view the Attention matrix as a non-symmetric kernel matrix, and extends a classical line of work of random kernel matrices (of the form $\sigma(X^\top X)$ for i.i.d. [1] or structured data [2]) to the **novel** random matrix model $X \sigma(X^\top X)$ that naturally arises in assessing the memorization of nonlinear Attention.
> This is indeed the main technical contribution of this paper.
> See more detailed discussion in the item of **technical contribution** in our global reply.
>
> - **Problem formulation**:
> Thank you for this question.
> As clarified above, we believe that the problem studied in this paper is indeed an in-context memorization problem, aligned in spirit with the line of prior work on in-context learning and memorization.
> A key distinction of our analysis, however, is that we **implicitly** account for the effect of pre-training by allowing the Attention weights to be *arbitrary deterministic* vectors, as formalized in Assumption 1, where $w_K$ and $w_Q$ can take any values.
> This choice allows our theory to directly capture scenarios where the pretrained weights are aligned with, orthogonal to the signal direction $\mu$, or anywhere in between, as discussed and numerically supported in Section 4 and Appendix D.
> In particular, this allows for Attention weights $w_K$ and $w_Q$ (obtained from pre-training) being *arbitrarily* different from the in-context signal-plus-noise model.
> This is formally different from most existing analyses, which typically incorporate pre-training **explicitly** by modeling a particular learning algorithm (e.g., gradient descent), as you have mentioned above, and focus on the in-context generalization performance.
> Here, on the other hand, we focus on the memorization behavior of the in-context task.
> In this vein, we believe that the "in-context memorization" studied in this paper, despite being formally different from that in the existing literature (since we do not statistically model the pre-training tasks, but focus instead on the resulting Attention weights), shares the same spirit.
>
> [1] X. Cheng, A. Singer "The Spectrum of Random Inner-product Kernel Matrices", 2012.
>
> [2] Z. Liao, R. Couillet, M. W. Mahoney, "Sparse Quantized Spectral Clustering", ICLR 2021.

---

### Official Review · Reviewer_QfQH · 2025-11-11

**Soundness:** 3
**Presentation:** 3
**Contribution:** 3
**Rating:** 6
**Confidence:** 3

**Summary:**

This paper provides a theoretical characterization of in-context memorization error for (nonlinear)attention mechanisms using the tools from random matrix theory (RMT). The authors analyze attention in the high-dimensional regime where the number of tokens ($n$) and embedding dimension ($p$) are both large and comparable.

Under a signal-plus-noise input model, authors have derived the deterministic equivalents for the memorization error and compare nonlinear attention with linear regression. The key finding is that while nonlinear attention generally underperforms linear regression on random inputs, this gap vanishes or reverses when inputs has statistical structure and attention weights align with the signal direction.

**Strengths:**

1. This work addresses a crucial gap by providing the precise RMT analysis of nonlinear attention on structured inputs. The extension of deterministic equivalents to handle the generalized sample covariance matrix (Proposition 1) could have broader applications.

2. This paper tries to bridges theory and practice by explaining why certain nonlinearities  (those with strong linear components) work better in transformers.  The finding that the first Hermite coefficient $a_1$ is crucial for memorization performance
can provides actionable design guidance.

3. The comparison between nonlinear attention and linear regression across different regimes  (varying SNR, embedding dimensions, regularization) is thorough.

**Weaknesses:**

1. The analysis performed in this paper is restricted to single-head attention with rank-one perturbations (Assumption 1)  and binary classification under a Gaussian signal-plus-noise model (Definition 2).  However, modern transformers use multi-head attention with more complex weight structures, and real-world data rarely follows such clean statistical models.  These disparities between the analyzed settings and practical transformers is quite significant .

2. Assumption 2 requires centered nonlinearities with $\mathbb{E}[f(\xi)] = 0$  and specific Hermite coefficient constraints.
While the authors claim this can be achieved by subtracting constants,  such modification could affect the attention mechanism’s behavior in ways not captured by the analysis.

3. While the paper includes numerical experiments confirming theoretical predictions,  these are all on synthetic data following the exact assumed model.  There is no validation on real data or even on more realistic synthetic settings.  The mention of *pretrained GPT-2 weights* experiments appears only briefly,  without sufficient details.


##### **Missing Prior Work** #####


The role of attention and FFN nonlinearity for in-context learning task [1, 2], and the impact of LayerNorms and GELU in the model's representational dynamics [3,4]


[1]  Wang et al., How do nonlinear transformers learn and generalize in in-context learning? ICML 2024

[2] Cheng et al., Transformers implement functional gradient descent to learn non-linear functions in context, ICML 2024

[3] Brody et al., On the expressivity role of layernorm in transformers’ attention, ACL 2023

[4] Jha et al., AERO: Entropy-Guided Framework for Private LLM Inference, 2025

**Questions:**

1. How fundamental are the technical barriers to extending the analysis to multi-head attention and realistic softmax settings?
Could authors provide a discussion on whether the key qualitative insights (e.g., importance of linear components, structured input advantage)  still hold when moving beyond hardmax and single-head formulations?

2. How sensitive are the conclusions made in this paper to deviations such as sub-Gaussian or heavy-tailed distributions,  which are common in NLP taks? Would these affect the memorization dynamics or theoretical rank behavior?

3. The analysis suggests that $\cos(t)$ performs poorly when $a_1 \approx 0$,  but periodic activations sometimes succeed in practice.
Could authors clarify this discrepancy and discuss regimes where low-$a_1$ activations might help?  Additionally, can authors  provide finite-sample bounds or insights on how quickly your asymptotic results converge  for realistic transformer sizes?

4. The results are framed as  ``in-context memorization,"  but the connection to broader in-context learning phenomena in LLMs remains unclear.  How does the analyzed memorization behavior relate to few-shot learning performance observed in practice?

---

> ### Author Response · Authors · 2025-11-24
>
> Thank you for your positive support as well as detailed and constructive comments.
> Below, we provide point-by-point responses to your concerns and questions.
> All modifications are marked in red of the revised paper.
>
> - **Simplified yet stylized model**:
> We agree with you that, for the sake of analytical tractability and simplicity of presentation, our analysis is performed on a simplified model of single-head attention, rank-one perturbed weights, and Gaussian signal-plus-noise model.
> Nonetheless, we believe that the model under study is "stylized," in that it allows to characterize key interactions between Attention nonlinearity, the structure of the pretrained weights, and their alignment (or misalignment) with the in-context task.
> Also, as we have discussed in Remarks 5–8, extensions to softmax Attention, beyond rank-one perturbed weights, beyond Gaussian signal-plus-noise model, and multi-head Attention are technically possible.
> As an example, we believe that in the case of multi-task context (i.e., more that one directions to be learned from the in-context data, that are correlated with different targets), multi-head Attention should be considered for the in-context memorization to be efficient.
> For the sake of simplicity and clarity of our message here, we prefer to leave them for future work.
> See also the item of **simplified yet stylized model** in our global reply.
> Moreover, experiments with pretrained GPT-2 weights in Figure 7 show consistent qualitative trends, supporting the insights derived from the simplified model.
>
> - **Centered nonlinearity**:
> Thank you for this comment.
> A generic nonlinearity $ g $ can always be written as $g(t) = a_0 + f(t) $, for $ a_0 = \mathbb{E}[ g(\xi) ] $ and $ \mathbb{E}[ f(\xi) ] = 0 $, $\xi \sim \mathcal{N}(0,1)$.
> Our analysis is carried out for this centered component $ f $.
> Taking any $g$ with $a_0 \neq 0$ corresponds to adding a **same constant** to all entries of the Attention matrix.
> Here, we make this assumption merely for the simplicity of presentation.
> And it is rather standard in the literature of high-dimensional analysis of neural network, see for example [1, 2].
>
> - **Real-data experiments**:
> The experiments in this paper are performed on signal-plus-noise data, which allows us to have a precise control on the interplay between data signal and the Attention weights (so as to confirm the insights obtained from our asymptotic analysis).
> To the best of our knowledge, such precise characterizations, even for the fundamental signal-plus-noise model considered here, is still absent in the literature.
> The experiments using real GPT-2 weights serve as a **qualitative "sanity check"** to confirm that the predicted trends, despite obtained from the perturbed rank-one weights in Assumption 1, also appear in realistic pretrained models.
> To have a better understanding of Attention for real data, it would be necessary to go beyond the current analysis by considering more involved and realistic modeling for the (embedded) data.
> We believe that the analysis framework proposed in this paper could serve an important first step to such results.
>
> - **Question 1: extension to multi-head and/or softmax Attention**:
> We focus on single-head Attention for analytical tractability.
> Extending the framework to multi-head and/or softmax Attention is technically feasible, since the Hermite expansion and random matrix techniques used here naturally extend to more general nonlinearities and to linear combinations of Attention heads, although the resulting fixed-point system of equations is expected to become more involved.
> That being said, we expect that our main insights, such as the role of the linear component and the advantage of weights aligning to the input signals, to largely carry over.
> In particular, the experiments with pretrained GPT-2 in Figure 7 show that softmax Attention follows a behavior similar to the proposed analysis.
> See also Remark 8 in the revised version for a brief discussion on possible extension to multi-head Attention.
>
> - **Question 2: sub-gaussian or heavy-tailed inputs**:
> As noted in Remark 7, our analysis extends to (mixture of) sub-gaussian data, as in line with a large body of universality results in high-dimensional statistics and random matrix theory.
> In contrast, genuinely heavy-tailed distributions do not satisfy the same type of concentration property (as sub-gaussian distribution) and may alter the Attention memorization behavior.
> We consider this as an interesting direction for future work.

---

> > ### Author Response · Authors · 2025-11-24
> >
> > - **Question 3: periodic activations**:
> > In our signal-plus-noise model, the in-context memorization performance highly depends on the first Hermite coefficient $ a_1 $ of the nonlinear function.
> > Since $\cos(t)$ has $ a_1 = 0 $ under Gaussian inputs, it is incapable of "amplifying" the input signal, which explains its poor behavior in memorization.
> > This does not contradict the empirical success of periodic functions in practical LLMs.
> > In particular, periodic functions are mainly used for positional encoding rather than as scalar activations in Attention computation.
> > For example, RoFormer [3] applies sinusoidal rotations to queries and keys to encode relative positions and improve the performance of LLMs.
> >
> > - **Question 4: connection to in-context learning in LLMs**:
> > Our formulation is aligned with the classical definition of in-context learning, where a pretrained model solves (i.e., memorizes and/or generalizes) some task directly from the prompt without parameter updates.
> > It should be noted that we incorporate the effect of pre-training **implicitly** by allowing the learned Attention weights to be arbitrary deterministic vectors (Assumption 1), which captures different degrees of alignment with the in-context signal direction $\mu$.
> > Under this setting, we analyze how such a frozen nonlinear Attention module performs in-context memorization.
> > This is different from some existing work where both the pre-training *and* the in-context tasks are assumed to follow some statistical models, and where some special pre-training algorithms (e.g., gradient descent) are **explicitly** considered.
> > See also our discussion in item **connections to and differences from prior work on in-context learning/memorization** of the global reply.
> >
> > - **Missing prior work**: Thank you for these related prior work and they have now been appropriately discussed in the revised version.
> >
> >
> > [1] Hong Hu and Yue M. Lu, "Universality Laws for High-Dimensional Learning with Random Features" IEEE TIT, 2022.
> >
> > [2] Jimmy Ba, et al. "High-dimensional asymptotics of feature learning: How one gradient step improves the representation", NeurIPS 2022.
> >
> > [3] Jianlin Su, et al. "Roformer: Enhanced transformer with rotary position embedding", 2024

---

### Author Response · Authors · 2025-11-24

## **Global reply**:

We thank all reviewers for their careful reading of our manuscript and for their insightful and constructive comments that have helped us significantly improve the paper.

Before addressing each comment individually, we would like to first clarify several common points raised across the reviews.
All modifications are marked in red of the revised paper.

### **Connections to and differences from prior work on in-context learning/memorization**:
We would like to clarify that the problem studied in this paper is indeed an in-context memorization problem, aligned in spirit with the line of prior work on in-context learning and memorization [1-4].

A key distinction of our analysis, however, is that we **implicitly** account for the effect of pre-training by allowing the Attention weights to be *arbitrary deterministic* vectors, as formalized in Assumption 1, where $w_K$ and $w_Q$ can take any values.
This choice allows our theory to directly capture scenarios where the pretrained weights are aligned with, orthogonal to the in-context signal direction $\mu$, or anywhere in between, as discussed and numerically supported in Section 4 and Appendix D.

This is formally different from most existing analyses, which typically incorporate pre-training **explicitly** by considering a particular learning algorithm (e.g., gradient descent) and tracking the optimization trajectory.

In this vein, our contribution focuses on characterizing the **inference-time** behavior of a frozen, pretrained Attention module, specifically how it memorizes and retrieves information from the current prompt **without** any parameter updates.
This matches the commonly used definition of in-context memorization and learning in the literature.

We have added some related clarifications in the introduction of the revised version of the paper.

### **Simplified yet stylized model**:
Our goal is to characterize the in-context memorization behavior of a pretrained nonlinear Attention.
To make the analysis both tractable and faithful to the phenomena of interest, we adopt a model that is **intentionally simple** for theoretical analysis yet **sufficiently stylized** to capture key factors such as the effect of nonlinearity, and the interaction between pretrained weights and the in-context task.

Specifically, we study nonlinear Attention with **structured** pretrained weights and in-context data drawn from a Gaussian signal-plus-noise model, in which the signal direction $\mu$ is to be learned.

This is in contrast with [5], which analyzes spectral properties of random self-Attention; and with [6], which studies a solvable dot-product Attention model with a positional–semantic phase transition; both works focus on random Attention rather than structured pretrained weights aligned with a specific in-context task.

Extensions to more involved settings such as softmax Attention, multi-task and/or non-Gaussian contexts (where multiple signal directions $\mu_1, \ldots$ correlated to the corresponding target $y$ are to be learned), perturbed pretrained weights having rank larger than one, or multi-head Attention, are all technically feasible (see our Remarks 5–8).
However, for clarity of presentation and to keep the theoretical contribution focused, we leave (the details of) these extensions to future work.
Importantly, experiments using pretrained GPT-2 weights (Figure 7) exhibit the same qualitative trends predicted by our simplified model, providing empirical support for the theoretical insights.


### **Technical contribution**:

Our main technical contribution is to view the nonlinear Attention matrix as a non-symmetric kernel matrix, which enables us to expand (or "linearize") it using Hermite polynomial and then study using random matrix techniques.
This extends a classical line of work: Hermite expansions were central in the seminal analysis of random kernel matrices of the form $\sigma(X^\top X)$ for $X$ having i.i.d. entries by Cheng & Singer [7], and were later generalized to structured data by Liao, Couillet & Mahoney [8].

Here, the memorization of Attention leads to a fundamentally more complex object of the form $X \sigma(X^\top X)$ and, must be evaluated for structured data, so as to capture the interactions between pretrained weights and the in-context task.
Hence, one must go beyond the models and settings considered in [7, 8].
This paper develops a systematic framework to analyze this new class of random matrix model arising from nonlinear Attention.
We believe that the proposed analysis framework may be of independent interest for studying more involved Transformer-type architectures or token-mixing mechanisms.
See also our discussion after Definition 3.

---

> ### Author Response · Authors · 2025-11-24
>
> ### References:
>
>
> [1] Singh et al, "Transient nature of in-context learning", 2023
>
> [2] Park et al, "Algorithmic phases of ICL", 2023.
>
> [3] Lin et al, "Dual operating modes of ICL", 2024.
>
> [4] Raventos et al, "Preptraining task diversity and the emergence of non-bayesian ICL for regression", 2023.
>
> [5] Hayase T, Collins B, Karakida R, "Gaussian Equivalence for Self-Attention: Asymptotic Spectral Analysis of Attention Matrix", 2025.
>
> [6] Cui H, Behrens F, Krzakala F, et al, "A phase transition between positional and semantic learning in a solvable model of dot-product attention", 2024.
>
> [7] X. Cheng, A. Singer "The Spectrum of Random Inner-product Kernel Matrices", 2012.
>
> [8] Z. Liao, R. Couillet, M. W. Mahoney, "Sparse Quantized Spectral Clustering", ICLR 2021.

---

### Author Response · Authors · 2025-11-28
**Additional global reply**

Dear fellow reviewers,

Thank you again for your time reviewing this manuscript, and for your constructive comments that have helped us significantly improve the paper.

In the submitted version of this paper, we chose to use the term "in-context" because we believe that our study aligns conceptually with prior results on in-context learning, by examining the behavior of fixed Attentions for structured input samples (where the fixed weights can be viewed as abstracted outcomes of pre-training).

We also need to emphasize that our analysis **differs** in the following aspects:

1. We treat the deterministic K and Q weights as inherent outcomes of pre-training, not derived from specific pre-training tasks (which are assumed to follow specific statistical models in most existing literature); and
2. We investigate statistical memorization/training performance, not generalization performance (in which case one must model the pre-training tasks).

As a consequence of these differences, we notice that the use of "in-context" may cause confusion that our analysis should be directly compared and contrasted with existing results on in-context generalization analysis.

To avoid such misunderstandings, and if you believe it would more clearly highlight our contributions, we would be happy to **reword the term "in-context" in the title and throughout the text**, explicitly stating that this study focuses on the memory/training performance of fixed nonlinear Attention for structured inputs, rather than on the pre-training procedure or out-of-distribution generalization.

---

### Meta-Review · Area_Chair_dSZo · 2025-12-30

**Summary:**

This paper focuses a specific model for in-context memorization and considers the effects of nonlinear attention. The reviewers gave evaluations with high variances. Overall most reviewers question whether the setup studied in the paper can be considered as in-context memorization. Two reviewers find the model to be not compelling enough; while the other two reviewers also have some concerns on the model they do appreciate some of the technical contributions (although one of them still has concerns related to the normalization and fixed weight). The authors should reframe the paper based on the suggestions of the reviewers.

**Reviewer Concerns:**

Main concern shared by most reviewers is that the setting is not in-context. The authors provided some justifications but I agree with the reviewers that this setting should not be called in-context.
There are some technical concerns on the simplifying assumptions, some of these are addressed, but two main points from 3wik were not well addressed.

**Reviewer Scores:**

QfQH: likely to stay at 6
nAJP: things are addressed but probably not convincingly enough for the reviewer. May increase to 3.
ceBM: some of the issues addressed, the relevance of the model may or may not be convincing to the reviewer. May increase to 3
3wik: still some critical questions in reviewers follow up not well addressed. Likely to stay at 6.

---

### Decision · Program_Chairs · 2026-01-26

Reject